# Accelerating the energy transition towards photovoltaic and wind in China

Yijing Wang[1], Rong Wang[1,2,3,4,5,6,7 ✉], Katsumasa Tanaka[8,9], Philippe Ciais[8,10], Josep Penuelas[11,12], Yves Balkanski[8], Jordi Sardans[11,12], Didier Hauglustaine[8], Wang Liu[1], Xiaofan Xing[1], Jiarong Li[1], Siqing Xu[1], Yuankang Xiong[1], Ruipu Yang[1], Junji Cao[13], Jianmin Chen[1,2,3], Lin Wang[1,2,3], Xu Tang[2,3] & Renhe Zhang[2,3]

China's goal to achieve carbon (C) neutrality by 2060 requires scaling up photovoltaic (PV) and wind power from 1 to 10–15 PWh year$^{-1}$ (refs. 1–5). Following the historical rates of renewable installation[1], a recent high-resolution energy-system model[6] and forecasts based on China's 14th Five-year Energy Development (CFED)[7], however, only indicate that the capacity will reach 5–9.5 PWh year$^{-1}$ by 2060. Here we show that, by individually optimizing the deployment of 3,844 new utility-scale PV and wind power plants coordinated with ultra-high-voltage (UHV) transmission and energy storage and accounting for power-load flexibility and learning dynamics, the capacity of PV and wind power can be increased from 9 PWh year$^{-1}$ (corresponding to the CFED path) to 15 PWh year$^{-1}$, accompanied by a reduction in the average abatement cost from US\$97 to US\$6 per tonne of carbon dioxide (tCO$_2$). To achieve this, annualized investment in PV and wind power should ramp up from US\$77 billion in 2020 (current level) to US\$127 billion in the 2020s and further to US\$426 billion year$^{-1}$ in the 2050s. The large-scale deployment of PV and wind power increases income for residents in the poorest regions as co-benefits. Our results highlight the importance of upgrading power systems by building energy storage, expanding transmission capacity and adjusting power load at the demand side to reduce the economic cost of deploying PV and wind power to achieve carbon neutrality in China.

Ambitions to achieve carbon neutrality are needed in all nations to limit global warming to below 2 °C in the Paris Agreement[8,9]. Accelerating the penetration of renewables is a key pillar in climate mitigation[10]. Global decarbonization is not, however, progressing as fast as it should to meet the goals of the Paris Agreement[11–13]. The world is probably on track for 2.8 °C of warming at the end of this century on the basis of current policies[11]. To achieve the global transition towards low-C economies, the 27th Conference of the Parties to the United Nations Framework Convention on Climate Change (COP27) recommended annual investments of US\$4–6 trillion (all currency values throughout the paper are in US dollars) to accelerate the penetration of renewables[14]. However, details on how these funds should be allocated among renewables remain unclear[1,6], requiring advanced spatially explicit models to optimize the existing power systems with geospatial details and coordinating infrastructure[8,15].

The rapid increase in global carbon dioxide (CO$_2$) emissions since 2000 has been driven mainly by the growing energy demand in developing countries[8]. Decarbonization may be more challenging in developing than developed countries[16], but mitigation in developing countries is

indispensable for meeting the climate goals[8–10]. China, with 18% of the global population and 28% of the global CO$_2$ emissions, has recently strengthened its nationally determined contribution with carbon neutrality target by 2060 (ref. 2). Among renewables, PV and wind power have wider ranges of application than hydropower[6], generate less detrimental effects on food and ecosystems than bioenergy[17] and probably entail lower costs than carbon capture and storage (CCS)[18]. Achieving carbon neutrality requires scaling up PV and wind power from 1 to 10–15 PWh year$^{-1}$ during 2020–2060 in China[1–5,19]. This capacity, however, would only reach 5 PWh year$^{-1}$ assuming the annual growth rate of 100 TWh year$^{-1}$ from 2010–2020 (ref. 1) or 9 PWh year$^{-1}$ using forecasts by the governmental plans of CFED[7] or 9.5 PWh year$^{-1}$ based on a recent high-resolution energy-system model[6]. There is also a chance that the growth of PV and wind power in China slows down owing to decreasing governmental subsides[20], a lack of transmission infrastructure[6] and restrictions for protecting agricultural, industrial and urban lands[21].

A spatially explicit method is needed for performing an optimization of energy systems by coordinating the generation of power with transmission and consumption of electricity in a country as vast as

[1]Shanghai Key Laboratory of Atmospheric Particle Pollution and Prevention (LAP³), Department of Environmental Science and Engineering, Fudan University, Shanghai, China. [2]IRDR International Center of Excellence on Risk Interconnectivity and Governance on Weather/Climate Extremes Impact and Public Health, Fudan University, Shanghai, China. [3]Institute of Atmospheric Sciences, Fudan University, Shanghai, China. [4]Shanghai Frontiers Science Center of Atmosphere-Ocean Interaction, Shanghai, China. [5]MOE Laboratory for National Development and Intelligent Governance, Fudan University, Shanghai, China. [6]Institute of Eco-Chongming (IEC), Shanghai, China. [7]National Observations and Research Station for Wetland Ecosystems of the Yangtze Estuary, Fudan University, Shanghai, China. [8]Laboratoire des Sciences du Climat et de l'Environnement (LSCE), CEA/CNRS/UVSQ, Université Paris-Saclay, Gif-sur-Yvette, France. [9]Earth System Division, National Institute for Environmental Studies (NIES), Tsukuba, Japan. [10]Climate and Atmosphere Research Center (CARE-C), The Cyprus Institute, Nicosia, Cyprus. [11]CSIC, Global Ecology Unit CREAF-CSIC-UAB, Bellaterra, Spain. [12]CREAF, Cerdanyola del Vallès, Spain. [13]Institute of Atmospheric Physics, Chinese Academy of Sciences, Beijing, China. ✉e-mail: rongwang@fudan.edu.cn

China[22]. Methods accounting for the spatial heterogeneity of PV and wind resources and demand for electricity transmission and storage have been developed for Europe[23] and the USA[24], but the flexibility of power load[25] and intertemporal dynamics of learning[26] have rarely been addressed in studies for China[6,27–29]. In contrast to previous studies[2,6,27–29], we developed a unified optimization framework that accounts for the geospatial capacities of installing new PV panels and wind turbines, expansion of existing UHV transmission, storing energy, flexible power loads and dynamics of learning. Our research highlights the need for investments in upgrading power systems and infrastructure, as well as the co-benefits of increasing resident incomes.

## Optimization of PV and wind power systems

We optimized the location, capacity and construction time of new PV and wind power plants each decade during 2021–2060 by minimizing the levelized cost of electricity (LCOE)[6,27] (Extended Data Fig. 1). The LCOE is defined as the normalized present value of costs including initial investment, operation and maintenance (O&M), land acquisition, UHV transmission and energy storage that are divided by the power generated over the lifetime (25 years (ref. 30)) of power plants (see Methods). We optimized the number of pixels receiving new PV panels or wind turbines to minimize the LCOE (Extended Data Fig. 2). We identified respectively 2,767, 1,066 and 11 power plants of PV, onshore wind and offshore wind at the utility scale (>10 MW) by considering resource limitations, administrative boundaries, land suitability, restriction of land use, ground slope, land cover, latitude, longitude, terrestrial and marine ecological conservation, water depth at offshore wind stations, shipping routes, solar irradiance, wind power density and air temperature (Fig. 1a,b). The predicted location and capacity of power plants match the observed PV and wind power plants to some extent[31] (Extended Data Fig. 3).

We identified diurnal variabilities and seasonal patterns of PV and wind power generation, which are not in phase with the profile of power demand (Fig. 1c). The power generation peaks in spring owing to variations in surface air temperature, shade, solar angle and inclination of PV panels (Extended Data Fig. 4). Our model considers the flexibility of power load whereby end users adjust hourly power demand to match the supply except for heating and cooling of houses and electric cars[25] (that is, 12% in total power demand by 2060) (see Methods). The adjusted power demand shifts in the daytime to match the peak of PV and wind power generation (Fig. 1c). Expanding the capacity of transmission by 6.4 TW and building new energy storage of 1.3 TW in China improves the efficiency of power use (Fig. 1d), whereas adopting a lower rate of electrification or considering a higher capacity of other types of renewable reduces the efficiency (Extended Data Fig. 5).

Similar to a previous study[32], we estimated that the rate of learning in China during 2000–2020 could be higher than the rate measured in other regions during 1975–2020 (Supplementary Table 1). On this basis, optimizing the construction time of power plants reduces the LCOE of PV and wind power plants from $0.067 to $0.046 per kWh (Fig. 1e). This requires an increase in PV and wind investment from $127 billion year[−1] in the 2020s to $426 billion year[−1] in the 2050s. This investment profile is similar to CFED[1,7] for the 2020s and 2030s but is lower in the 2040s and 2050s. Investment in our optimal path over the period 2031–2050 ($220 billion year[−1]) is lower than a previous estimate[6] ($320 billion year[−1]) made without simulating the dynamics of learning.

We quantified the effects of optimization relative to a baseline scenario, which limits the capacity of PV and wind power plants to 10 GW without electricity transmission and energy storage and assumes that the growth rate of PV and wind power is constant during 2021–2060 without optimizing the dynamics of learning[26]. We designed five sensitivity experiments by sequentially increasing the limit of capacity for individual plants from 10 to 100 GW (case A), considering the construction of new UHV lines (case B), adding energy storage (case C), improving electrification of non-power sectors[33] from 0 to 58% (case D) and considering flexible power loads (case E). Case E becomes equivalent to our optimal path if the construction time of power plants is optimized through accounting for the dynamics of learning[26]. Our optimal path increases the capacity the most by storing energy (+6.4 PWh year[−1] as a difference between cases B and C), but it reduces the costs the most by optimizing the dynamics of learning (−$115 billion year[−1] as a difference between case E and the optimal path) (Fig. 1f).

## Costs of $CO_2$ emissions reduction

We estimated the marginal abatement cost (MAC) at the plant level, which varies from −$166 per $tCO_2$ to $106 per $tCO_2$ in 2060 in our optimal path (Fig. 2a). For example, 77% of PV and wind power could be competitive against nuclear power with a lower MAC[1]. The average abatement cost (−$4.5 per $tCO_2$) for 9.5 PWh of power generation is lower than a previous estimate ($27 per $tCO_2$) under an 80% renewable penetration in China[6]. The MAC increases as the capacity rises owing to techno-economic limits and differences in the prices of the substituted fossil fuel (Extended Data Fig. 6). Such behaviour of the MAC indicates an increase in the costs to install higher capacities of PV and wind power[34], even by considering the benefits of technological improvements[26].

The capacity of PV and wind power reaches 15 PWh in 2060 (9 PWh in the CFED plan[7]) with an average abatement cost of $6 per $tCO_2$ ($97 per $tCO_2$ in the CFED plan[1]) (Fig. 2b,c). The $CO_2$ emissions were most abated by storing energy (+3.5 Gt $CO_2$) for power plants with MAC < $100 per $tCO_2$ and by optimizing the dynamics of learning (+3.5 Gt $CO_2$) for power plants with MAC < $0 per $tCO_2$. The costs of PV and wind power would increase if we assumed international learning rates (Supplementary Table 1), high capital costs[35], a short lifetime of power plants (20 years)[30] or a high discounting rate (7%)[6], but decrease if we neglected the costs of new UHV lines or adopted low capital costs[36]. For example, the average abatement cost increased from −$2 to $14 per $tCO_2$ if we increased the discounting rate from 3% to 7% to reduce the revenue from power generation, but it decreased from $22 to $0 per $tCO_2$ if we increased the lifetime of power plants from 15 to 35 years (Supplementary Fig. 1). The cost composition shifted from transformers and O&M to modules and land acquisition as we moved from the baseline case to the optimal path (Fig. 2d). A recent study showed that globalized supply chain reduces global solar-module prices[32]. Our results indicate that the impact of technological transformation between countries might be moderate for China with a fast decline in module prices (Supplementary Fig. 2).

## Trade-offs among land, costs and power

We analysed the trade-offs among land requirements, costs and power capacity (Table 1). The capacity of PV and wind power could provide up to 59% of the projected total power demand in China for 2060, compared with a contribution of 20% by hydrogen, nuclear and biomass in a scenario keeping global warming below 1.5 °C by ref. 2. Expansion of PV and wind power from 1 to 15 PWh year[−1] requires 585,000 km² of land for placing PV panels and 672,000 km² of area for installing onshore and offshore wind turbines, in which 33%, 35%, 16% and 6% of facilities are distributed in deserts, grassland, oceans and cropland, respectively (Extended Data Table 1). Building these PV and wind power plants requires initial investment of $201 billion year[−1] with O&M costs of $47 billion year[−1], the sum of which is 7% of the total spending of China's public finances[37] in 2020. These costs can partly offset the income by saving costs of purchasing fossil fuels ($223 billion year[−1]) and reducing carbon costs ($399 billion year[−1]) by assuming a representative carbon price of $100 per $tCO_2$ from the 1.5 °C scenario by ref. 2.

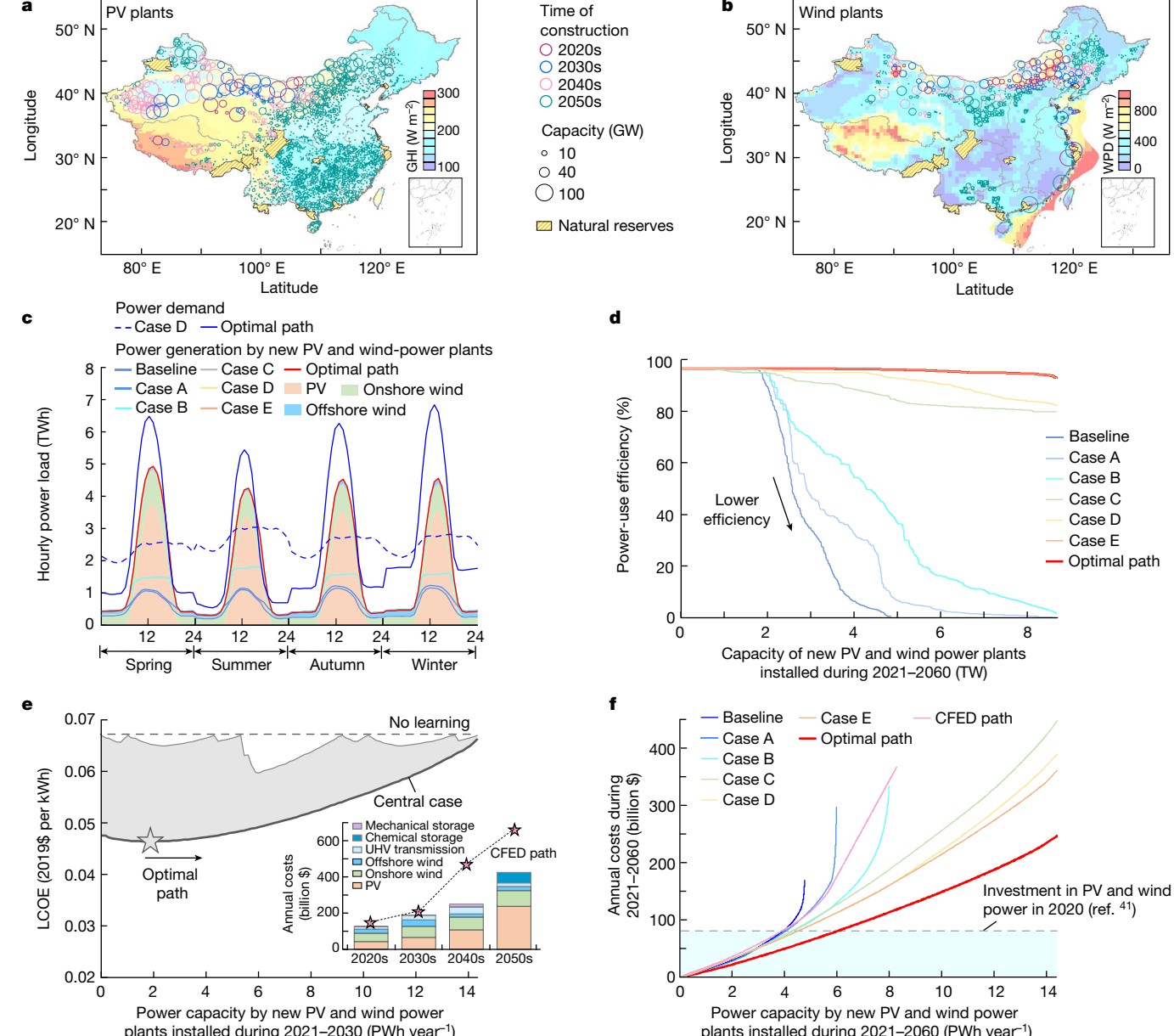

**Fig. 1 | Optimization of the location, capacity and construction time of utility-scale PV and wind power plants during 2021–2060 in China.**
**a**,**b**, Maps of PV (**a**) and wind (**b**) power plants built by decade in the optimal path. The background shows global horizontal irradiance (GHI) and wind power density (WPD). **c**, Seasonal and diurnal variations in the generation and demand of power under an electrification rate of 58% for non-power sectors in 2060. The shading represents the PV and wind power generation without considering curtailments. In a baseline scenario, the capacity of individual PV and wind power plants is limited to 10 GW without electricity transmission and energy storage, whereas the growth rate of PV and wind power is constant during 2021–2060 without considering the dynamics of learning. We design five experiments by sequentially increasing the limit of power capacity from 10 to 100 GW (case A), building new UHV lines (case B), storing energy (case C), improving the electrification of non-power sectors (case D) and considering

flexibility of power loads (case E). Case E becomes equivalent to our optimal path if the construction time of power plants is optimized by accounting for the dynamics of learning. **d**, Power-use efficiency defined as the fraction of the generated power consumed by end users. **e**, Influences of increasing the capacity of new PV and wind power plants built in the 2020s on the LCOE of all new PV and wind power plants built during 2021–2060. The optimal path minimizes the LCOE by optimizing the construction time of individual power plants (shaded area) under a discounting rate of 5%. The inset shows the annual costs by decade. We consider a 'CFED path' by following the rate of installing renewables in China's 14th Five-year Energy Development (CFED)[7] with the projected costs of PV and wind power[1]. **f**, Dependency of the annual average costs of deploying PV and wind power during 2021–2060 on the power capacity of new PV and wind power plants built during 2021–2060 under different scenarios.

We predicted that 183 of 3,844 plants will be built with capacity >10 GW. The average abatement cost will decrease from $62 to $6 per tCO$_2$ as the limit of capacity for individual plants increases from 0.1 to 10 GW (Extended Data Fig. 7). The feasibility of building large power plants in China could be supported by commissions of the Jiuquan

onshore wind power plant at 20 GW and the Yanchi PV power plant at 1 GW, but it entails high requirements on grid integration, electricity transmission and initial investment[38]. Non-economic factors such as ecological preservation, engineering feasibility and political impediment deserve attention.

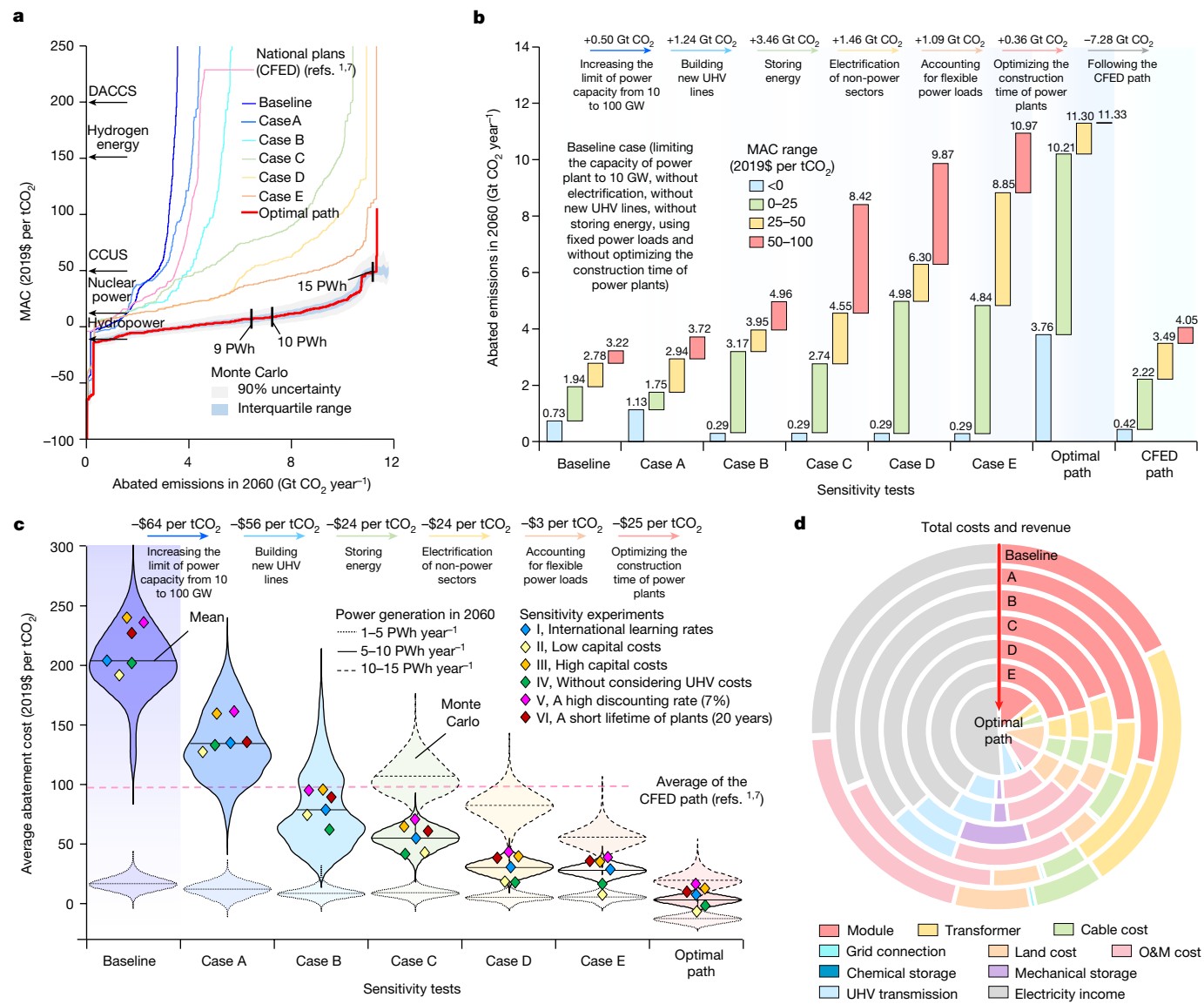

**Fig. 2 | Costs of CO₂ emissions abatement in 2060 by deploying PV and wind power in China. a**, The MAC under a 5% discounting rate in 2060. The configurations of the baseline case, cases A–E, the CFED case and our optimal path are identical to those defined in Fig. 1. Arrows represent the MACs for hydropower, nuclear power, hydrogen energy, carbon capture utilization and storage (CCUS) and direct air carbon capture and storage (DACCS)[1]. **b**, Impacts of our optimizing procedures on the potential of CO₂ emissions reduction based on the ranges of MAC. **c**, Violin plots of the average abatement costs

when building new PV and wind power plants to meet the power demand in 2060. We perform sensitivity experiments by applying international learning rates (Supplementary Table 1) (I), adopting low[36] (II) or high[35] (III) capital costs, neglecting the costs of building new UHV lines (IV), adopting a high discounting rate (7%)[6] (V) and assuming a short lifetime (20 years) of power plants[30] (VI). **d**, Composition of the costs and income when increasing PV and wind power generation from 1 to 10 PWh year⁻¹ to replace fossil fuel in 2060.

## Implication for carbon neutrality

Many scenarios meeting the target of carbon neutrality[8] rely on retrofitting existing plants with CCS, which may be limited by economic costs[1], geological constraints[39] and biomass availability[17]. We analysed the impact of deploying PV and wind power on the demand for CCS with fossil fuel[2] or bioenergy[17] to achieve carbon neutrality in 2060 by considering terrestrial carbon sinks, electrification of non-power sectors (58%)[33] and power supply by other renewables[2] (Fig. 3). A transition from CFED[7] to our optimal path reduces the demand for CCS from 8.9 to 2.8 PWh year⁻¹ in 2060 (Fig. 3a). The share of PV and wind in power supply increases from 12% to 59% during 2021–2060 at an annual rate of 1.8%, 1.4%, 1.0% and 0.7% in the 2020s, 2030s, 2040s and 2050s, respectively, which requires acceleration relative to an annual rate of 1% for China in the 2010s[40]. Although the projected annual growth rates

for wind (1%) and PV (0.8%) in China during the 2020s are comparable with the maximal annual rates of 1% in Spain, 0.9% in Turkey and 0.6% in the USA and New Zealand for wind or 1.1% in Japan and 1% in Germany for PV[13], the expansion of these technologies may present greater challenges in China because of her larger absolute power demand[1].

Upgrading power systems is crucial to accelerating the penetration of renewables in China[7,27–29]. For example, the growth of PV and wind power does not depend on investment in electricity transmission in CFED plans[7] (Fig. 3c). By contrast, our model optimizes the dynamics of learning (Extended Data Fig. 8) and the strategy of energy storage (Extended Data Fig. 9) by accounting for investment in UHV lines (Fig. 3b). An example of the findings shows that, with the increase in PV and wind investment from $0 to $60 billion year⁻¹ over the period 2021–2060, the ratio of cost reduction for PV, wind and CCS to the increase in costs for PV and wind power is 2.7:1 in the optimal path towards carbon

**Table 1 | Capacity, land acquisition and annual costs of new PV and wind power plants built during 2021–2060 under a discounting rate of 5%**

| Annual power generation by PV and wind power plants (PWh year⁻¹) | Total area (Mha) | Length of new UHV lines (million km) | Capacity of energy storage (TW) | Number of new PV and wind power plants | Number of new PV and wind power plants with capacity >10GW | Average power capacity (GW) | Average capacity factor (%) | Annual cost of electricity transmission (billion $ year⁻¹) | Annual cost of energy storage (billion $ year⁻¹) | Annual initial investment cost (billion $ year⁻¹) | Annual O&M cost (billion $ year⁻¹) | Average abatement costs ($ per tCO₂) |
|---|---|---|---|---|---|---|---|---|---|---|---|---|
| 1–5 | 29.05 | 0.46 | 0 | 153 | 48 | 14.33 | 20.59 | 10.49 | 0 | 47.68 | 10.50 | −3.70 |
| 5–10 | 47.93 | 0.46 | 0.42 | 619 | 77 | 4.60 | 19.97 | 8.58 | 6.43 | 62.86 | 16.95 | 0.61 |
| 10–15 | 43.02 | 0.21 | 0.88 | 3,023 | 55 | 1.10 | 17.00 | 3.83 | 12.74 | 82.97 | 16.07 | 15.69 |
| Total | 132.05 | 1.14 | 1.34 | 3,844 | 183 | 2.26 | 18.84 | 23.35 | 19.87 | 201.21 | 46.58 | 5.73 |

neutrality in 2060, which is lower than the ratio of 6.4:1 in CFED plans[7]. This ratio increases to 5.4:1 in our optimal path when investment in PV and wind power systems increases from $60 to $250 billion year⁻¹, but decreases to 0.4:1 in CFED plans[7]. Our results highlight the importance of ramping up PV and wind investment relative to current levels ($77 billion year⁻¹ in 2020)[41] to reduce the economic costs of achieving carbon neutrality.

## Implication for alleviating poverty

Deploying renewables has been suggested as an effective way to reduce poverty[42] by generating revenue from wealthier regions. This impact, however, has not been assessed by a national cost–benefit analysis in China. A higher carbon price generates more revenue for PV and wind power by saving more carbon costs (Fig. 4a). Accounting for the finances embodied in the transmission of electricity (see Methods), we found that the revenue from PV and wind power could be redistributed from the more developed east to the less developed west. Distributing the revenue to less-developed regions as the carbon price increases from $0 to $100 per tCO₂ removes 21 million people from the low-income group (<$5,000 year⁻¹) and adds 6 million people to the high-income group (>$20,000 year⁻¹) (Fig. 4b,c).

Increasing the carbon price from $0 to $100 per tCO₂ reduces income equality, with a decrease in the Gini coefficient from 0.453 to 0.441 (Fig. 4d). Adopting a carbon price of $100 per tCO₂ generates a financial flow of $1,055 billion in the transmission of PV and wind power in 2060, which is 15-fold higher than China's annual investment in poverty alleviation during 2014–2020 (ref. 37). The generation of PV and wind power is dominated by Northwest China (5.9 PWh year⁻¹) and North China (5.2 PWh year⁻¹), whereas the consumption is dominated by East China (5.7 PWh year⁻¹) and Central China (4.3 PWh year⁻¹). The transmission of electricity leads to the largest finance flow ($223 billion year⁻¹) from East China to Northwest China (Fig. 4e). By increasing the carbon price from $0 to $100 per tCO₂, deployment of PV and wind power benefits the poorest residents, with an increase in per-capita income from $29,000 to $34,400 in North China and from $29,100 to $30,600 in Northwest China.

## Implication for climate policies

The gap between current decarbonization rates and the levels required to achieve carbon neutrality remains substantial[8–10]. China has been at the forefront of PV deployment since 2009 (ref. 27), accompanied by an accelerated growth of wind power[1]. Despite these accomplishments,

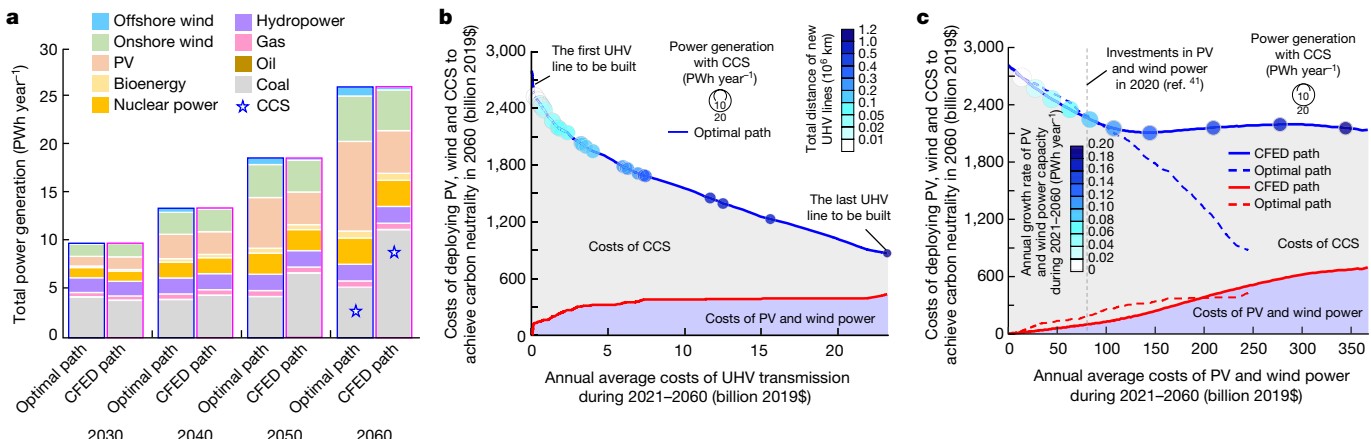

**Fig. 3 | Different paths to achieve carbon neutrality in China by 2060.**
**a**, Composition of the generated power by decade. The projected generation of power by oil, gas, bioenergy, nuclear power and hydropower are derived from the '1.5-°C-limiting' scenario in a multi-model study[2]. The PV and wind power are projected by our optimal path and CFED[7]. Assuming that coal meets the remaining power demand, we estimate the demand for CCS installed with fossil fuel or biomass when achieving carbon neutrality in 2060. **b**, Dependence of the annual costs of PV, wind and CCS when achieving carbon neutrality by 2060 on the costs of transmitting electricity by UHV lines during 2021–2060 in the optimal path. **c**, Dependence of the annual costs of PV, wind and CCS when achieving carbon neutrality by 2060 on the costs of PV and wind power during 2021–2060 in our optimal (dashed lines) path and CFED[7] (solid lines). In **b**, the total capacity of PV and wind power plants built during 2021–2060 in the optimal path depends on the capacity of electricity transmission, whereas the total distance of new UHV lines is indicated by the colour of the circle. The total capacity of PV and wind power built by 2060 in CFED[7] depends on the annual growth rate of PV and wind power during 2021–2060, which is indicated by the colour of the circles in **c**. We predict the costs of CCS based on the MAC of CCS[1] and the demand for CCS when achieving carbon neutrality by 2060 under different scenarios.

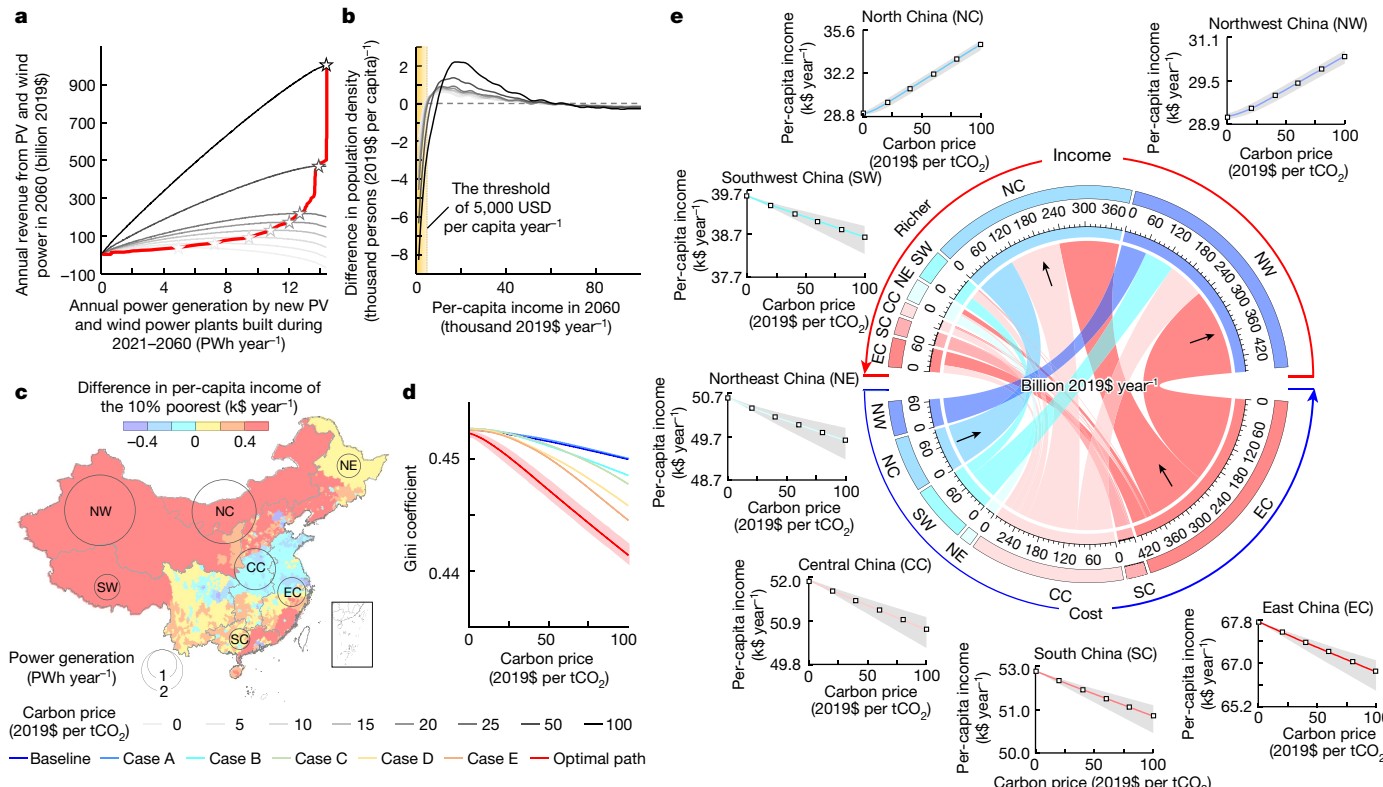

**Fig. 4 | Co-benefits of deploying PV and wind power on poverty alleviation in China. a**, Revenue from PV and wind power generation in 2060 under different carbon prices. **b**, Change in the distribution of per-capita income when the carbon price increases from $0 to $100 per $tCO_2$. **c**, Change in the income of the 10% poorest people in 2060 owing to the deployment of PV and wind power during 2021–2060 under the carbon price of $100 per $tCO_2$. **d**, Variation in the Gini coefficient when the carbon price increases from $0 to $100 per $tCO_2$. The shaded area represents the 90% uncertainty in Monte Carlo simulations. **e**, Flow of finances embodied in the transmission of electricity generated by new PV and wind power plants built during 2021–2060 under the carbon price of $100 per $tCO_2$. The insets show the variations in per-capita income by region when the carbon price increases from $0 to $100 per $tCO_2$ in 2060.

it remains challenging to achieve carbon neutrality by 2060. As fossil fuels continue to dominate energy-related investments[12], renewable growth could slow down as subsidies for companies generating PV and wind power decline[20,21]. Unlike previous studies[1,2,6,27–29], our research reveals greater potential for PV and wind power generation in China, alongside the need for larger investment in power-system upgrades.

Our approach enhances the optimization of PV and wind power systems[2,6,27–29] by applying a spatially explicit method that provides insights into climate mitigation in countries beyond China[14]. First, deeper decarbonization requires greater investments in renewables because of physical constraints in abating more $CO_2$ emissions (for example, larger demand for land and infrastructure), even when accounting for technological improvements[26,30]. Despite the projection of decreasing renewable energy costs[43], we emphasize the importance of policy interventions (for example, building large PV and wind power plants, grid integration, energy storage and demand-side power-load management) to reduce renewable costs. Second, deploying PV and wind power can offer new sources of income in less-developed regions with vast areas of desert and marginal lands. It has implication for accelerating economic development by deploying renewables in semiarid regions such as Africa and the Middle East[44]. Third, optimizing power systems for large developing countries can lower the costs of deploying renewables in the upcoming decades, making it feasible to achieve more ambitious climate targets beyond the 2060 carbon neutrality[45]. Our research highlights the technical and physical constraints on deploying renewables to mitigate $CO_2$ emissions, the importance of scaling up investments to accelerate energy transition to PV and wind power and the optimal route to achieve carbon neutrality in the long run.

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

according to new analysis from OECD and IEA https://www.oecd.org/newsroom/support-for-fossil-fuels-almost-doubled-in-2021-slowing-progress-toward-international-climate-goals-according-to-new-analysis-from-oecd-and-iea.htm (2022).

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

## Methods

### Geospatial data in this study

An overview of our optimization model is shown in Extended Data Fig. 1. We optimized the placement and capacity of PV and wind power plants in our model driven by geospatial data (Supplementary Method 1), including land cover, solar radiation, wind speed, surface air temperature, ground slope, latitude and longitude of the installed PV panels, terrestrial and marine ecological reserves, water depths of offshore stations and marine shipping routes (Supplementary Table 2). All land pixels were categorized into forest, shrubland, savannah, grassland, wetland, cropland, urban and built-up land, mosaics of natural vegetation, snow and ice, deserts and water bodies[46]. The suitability of the installation of PV panels or wind turbines was defined by land cover (Supplementary Table 3). Onshore wind turbines with the capacity of 2–2.5 MW and offshore wind turbines with the capacity of 5–10 MW are considered as the main models used in China at present[47], so we considered models for onshore (General Electric 2.5 MW) and offshore (Vestas 8.0 MW) wind power plants (Supplementary Table 4) at a hub height of 100 m above the ground to convert air kinetic energy to electricity based on the recommended power-generation curve[48] (Supplementary Fig. 3). Resources of solar and wind energy were associated with seasonal and diurnal variabilities and interannual differences. We estimated hourly solar radiation and wind speed at a hub height of 100 m above the ground as averages for 2012–2018 to provide a representative estimate of solar and wind energy in China (Supplementary Method 2). All geospatial data were projected to a resolution of 0.0083° in latitude and 0.033° in longitude for estimating the potential of power generation by installing PV panels or wind turbines in each pixel (Supplementary Methods 3–5).

### The optimization model

We estimated the LCOE of the PV and wind power systems to indicate the grid parity of power generation, which is defined as the normalized net present value of all costs of investments, O&M, land acquisition, transmission and energy storage divided by the power generated during the lifetime (25 years (ref. 30)) of the PV and wind power plants[35]. Before solving the optimization problem, we sought the best strategy for installing PV panels or wind turbines with different shapes to achieve the maximal capacity of power generation in each county (Supplementary Fig. 4). We took the number of pixels installing PV panels or wind turbines and the time to build each PV or wind power plant by decade as decision variables. By accounting for the intertemporal dynamics of learning[26,30], we developed a unique method to optimize the capacity of each power plant, the order of building power plants, the time to build each power plant and the option of energy storage when building a new power plant by solving a cost-minimization problem based on the LCOE for generating power by the projected PV and wind power plants:

$$\min_{\epsilon, n_x, t_x, s_x} \text{LCOE}_\epsilon = \frac{(V_\epsilon + A_\epsilon) \cdot \left[1 + R_y \cdot \sum_{t_p=1}^{T} \frac{1}{(1+r_d)^{t_p}}\right] + G_\epsilon \cdot \sum_{t_g=1}^{L_g} \frac{1}{(1+r_d)^{t_g}}}{E_\epsilon \cdot \sum_{t=1}^{T} \frac{1}{(1+r_d)^{t}}} \quad (1)$$

$$V_\epsilon = \sum_{q=1}^{7} \sum_{x=1}^{n_q} [1 - \xi_x(t_x)] \cdot V_x(n_x), \ x \in q \quad (2)$$

$$E_\epsilon = \sum_{h=1}^{8,760} \left[\sum_{q=1}^{7} \sum_{x=1}^{n_q} E_{x,h}(n_x) - \eta_{\text{tra}} \Theta_h - \eta_{\text{store}} \Lambda_h\right], \ x \in q \quad (3)$$

$$\Theta_h = \sum_{q=1}^{7} \max\left[0, \sum_{x=1}^{n_q} E_{x,h}(n_x) - M_{q,h}\right] = \sum_{q=1}^{7} U_{q,h}, \ x \in q \quad (4)$$

$$\Lambda_h = \sum_{q=1}^{7} \max\left[0, \sum_{x=1}^{n_q} E_{x,h}(n_x) + U_{q,h} - M_{q,h}\right], \ x \in q \quad (5)$$

in which $\epsilon$ is a new power plant ($\epsilon$ = 1 to 3,844), $x$ is a power plant built before $\epsilon$, $n_x$ is the number of pixels installing PV panels or wind turbines in plant $x$, $t_x$ is the time to build plant $x$, $s_x$ is the option of energy storage (1 for pumped hydro and 2 for chemical batteries) when building plant $x$, $T$ is the average lifetime of a power plant, $h$ is hour, $q$ is a region (1–7 for Central China, East China, North China, Northeast China, Northwest China, South China and Southwest China, respectively), $n_q$ is the number of power plants in region $q$, $r_d$ is the discounting rate (5%)[2], $\tau_p$ is a year in the operation of plant $x$, $\tau_g$ is a year during operation of energy storage in plant $x$, $L_g$ is the lifetime of storage (50 years for pumped hydro[6] and 15 years for chemical batteries[49]), $E_\epsilon$ is the total power generation, $V_\epsilon$ is the total investment in power plants, $A_\epsilon$ is the total cost of electricity transmission, $G_\epsilon$ is the total cost of storage, $R_y$ is the ratio of O&M costs to investment costs (1% for PV[50] and 3% for onshore and offshore wind power plants[51]), $V_x$ is the investment in plant $x$, $\xi_x$ is the ratio of cost reduction by learning when building plant $x$, $E_{x,h}$ is the hourly generation of power in a county, $\Theta_h$ is the hourly transmission of electricity, $\Lambda_h$ is the hourly storage of electricity, $\eta_{\text{tra}}$ is the fraction of electricity lost during transmission, $\eta_{\text{store}}$ is the fraction of electricity lost during storage, $M_{q,h}$ is the hourly consumption of electricity in region $q$ and $U_{q,h}$ is the hourly transmission of electricity from other regions to region $q$.

We optimized the increase in power capacity at an interval of 10 years during 2021–2060 because it generally takes 10 to 20 years for new technologies to be widely applied[52]. Given the variation of renewable energy within a decade, we performed a sensitivity experiment by optimizing the model at an interval of 5 years, in which the installed PV and wind power capacity and total costs both change moderately (Extended Data Fig. 8). Nevertheless, simulating the penetration of renewable energy within a decade will be useful to improve the optimization model.

We considered the connection of power plants in a county to one of the substations from the UHV transmission lines projected in the CFED plan[7] (Supplementary Data Set 1) with the costs of building new UHV lines (Supplementary Method 6) and developing systems for storing energy (Supplementary Method 7). We estimated $U_{q,h}$ using three assumptions. First, the electricity generated by all PV and wind power plants in a region is used to meet the power demand in this region as a priority. Second, the extra electricity is connected to a substation in the UHV line and transmitted to a region in which the next substation in the line is located. Third, the electricity transmitted to a region is distributed to each county based on the distribution of the consumption of electricity in this region. We have considered the impact of transmission access on where and when to build new PV and wind power plants. When optimizing the construction time of 3,844 PV and wind power plants, we have accounted for the costs of building new UHV transmission lines when a new power plant is built, which influences the LCOE of this plant and the construction time of all new plants. By optimizing the construction time of each new power plant, we considered that a new UHV line required for this new plant will be constructed at the same time.

As a caveat of this study, we do not have explicit information for all UHV transmission lines, so we assumed that the UHV lines projected by the central government are used as a proxy for UHV lines between the main regions in the country. This assumption is useful to determine the demand for electricity transmission between regions, but it could lead to bias in our cost estimation because of the lack of detailed information for all UHV lines. For example, the projection of 128 UHV lines with a capacity of 12 GW each from Huaidong to Wan'nan in our model indicates that at least a total transmission capacity of 1,536 GW is required for transmitting electricity from the region centred in Huaidong to the region centred in Wan'nan, but the ultimate UHV lines built between

these two regions might be different from our prediction. This limitation can be addressed when the detailed information for all UHV transmission lines are available. To consider the outflow of electricity generated in a county, we sought the substation of UHV lines that is closest to this county and then we estimated the cost of electricity transmission from this county to the transmission substation and the cost of electricity transmission using one of the UHV lines. Although this study projected the construction of a large transmission capacity to optimize power systems, it is important to account for the physical, technical and economic constraints. These include the demand for advanced polymer matrix composites that can operate under a voltage of >1,000 kV, the construction of UHV lines over challenging terrains, the maintenance of these lines and ensuring the security of electricity transmission under extreme weather conditions.

We sought the optimal system for storing energy when building a new power plant using either mechanical storage (pumped hydro) with a lifetime of 50 years and a round-trip efficiency of 70% or chemical storage (batteries) with a lifetime of 15 years and a round-trip efficiency of 85% (see the parameterization of two systems in Supplementary Table 5) to minimize the LCOE (Extended Data Fig. 9).

Last, $\xi_x$ is calculated as a function of the total capacity of installed PV or wind power (Supplementary Method 8) based on the measured rates of learning in China (Supplementary Table 1). We examined the sensitivity to adopting the international rates of learning in our model (Fig. 2c).

## Capacity and costs of power generation

For a new PV or wind power plant $x$, the annual generation of power was calculated:

$$E_{x,h} = \sum_{j=1}^{8,760} \sum_{i=1}^{n_x} W_{i,j} \tag{6}$$

in which $i$ is a pixel, $j$ is the number of hours in a year and $W_{i,j}$ is the hourly generation of power in a pixel installing PV panels (calculated in Supplementary Method 3), onshore wind turbines (calculated in Supplementary Method 4) or offshore wind turbines (calculated in Supplementary Method 5). The parameters used to estimate the projected PV and wind power generation are listed in Supplementary Table 6.

The investment cost of a new PV or onshore wind power plant $x$ was calculated[6]:

$$V_x = \sum_{i=1}^{n_x} (\mu_{\text{fix}} P_i + \mu_{\text{land}} S_i) + 2\mu_{\text{line}} \sqrt{\pi \sum_{i=1}^{n_x} S_i} + \mu_{\text{tran}} \text{int}\left(\frac{\sum_{i=1}^{n_x} P_i}{P_{\text{fix}}}\right) \tag{7}$$

in which $i$ is a pixel, $P_i$ is the installed capacity of PV panels (calculated in Supplementary Method 3) or onshore wind turbines (calculated in Supplementary Method 4), $S_i$ is the area of pixels installing PV panels or wind turbines, $P_{\text{fix}}$ is the capacity of a voltage transformer (300 MW), $\mu_{\text{fix}}$ is unit capital costs, $\mu_{\text{land}}$ is unit cost of land acquisition, $\mu_{\text{line}}$ is unit cost of line connection and $\mu_{\text{tran}}$ is unit cost of voltage transformation.

We assumed that the installed voltage transformer has a capacity of 300 MW (ref. 53). When estimating the unit cost of land acquisition, we considered that onshore wind turbines take up only 2% of area in a pixel and PV panels take up 100% of area in a pixel[54]. We derived $\mu_{\text{fix}}$ as the sum of costs for modules ($\mu_{\text{module}}$), inverters ($\mu_{\text{inverter}}$), mounting materials ($\mu_{\text{mounting}}$), secondary equipment ($\mu_{\text{sec}}$), installation ($\mu_{\text{ins}}$), administration ($\mu_{\text{adm}}$) and grid connection ($\mu_{\text{grid}}$) using the data for PV panels ($0.64 per watt) published by the China Photovoltaic Industry Alliance[55] and using the data for onshore wind turbines ($0.68 per watt) from a previous study[56]. We demonstrated the impact of using different capital costs by examining the sensitivity to adopting high capital costs ($0.73 and $0.88 per watt for PV panels and onshore wind turbines, respectively)[35] or low capital costs ($0.23 and $0.76 per watt for PV panels and onshore wind turbines, respectively)[36] in the sensitivity tests (Fig. 2c).

The investment cost of an offshore wind power plant $x$ was calculated on the basis of the distance of offshore wind turbines in this power plant to the onshore power station[57]:

$$V_x = \sum_{i=1}^{n_x} [\mu_{\text{baseline}} (z_0 + z_1 D_{L,i}) (z_2 + z_3 D_{P,i}) P_i] \tag{8}$$

in which $i$ is a pixel, $\mu_{\text{baseline}}$ is the unit cost of offshore wind turbines, $P_i$ is the installed capacity of offshore wind turbines (calculated in Supplementary Method 5), $D_{L,i}$ is the distance of the offshore wind turbines to the onshore power station and $D_{P,i}$ is the water depth of the installed offshore wind turbines. The coefficients $z_0$ (0.0057), $z_1$ (0.7714), $z_2$ (0.0084) and $z_3$ (0.8368) were calibrated using engineering data[57]. The parameters used to determine the costs of PV and wind power generation are listed in Supplementary Table 7. The parameters used to determine the costs of UHV transmission and energy storage are listed in Supplementary Table 8.

We adopted a fixed ratio of O&M costs to investment costs for the projected PV and wind power plants[50,51]. We adopted 25 years (ref. 30) as the average lifetime of PV or wind power plants. We considered the costs of electricity transmission by UHV when increasing the installed capacity of a power plant. We sought the geographic centre among all pixels suitable for power generation and then increased the number of surrounding pixels ($n_x$) installing PV panels or wind turbines. The capacity of power generation by each power plant increases as the number of pixels installing PV panels or wind turbines increases in the order of the distance to the geographic centre. The inclusion of more pixels in a power plant, however, increases not only the capacity of this PV or wind power plant but also the total costs in the power systems. The LCOE for a new power plant first decreased when we increased the power capacity by increasing the number of pixels for installation of PV panels or wind turbines, because the capital costs were divided by the generated power, but then increased owing to the increasing costs of purchasing land and the decreasing power-use efficiency (Extended Data Fig. 2). We sought the optimal capacity for each power plant for achieving the minimum of the LCOE.

## MAC

We assumed that the electricity generated by new PV and wind power plants was used to replace oil, gas and coal in the order of fuel price to produce the highest profits[6]. Solving the cost-minimization problem in equation (1) was constrained by the target of the annual abatement of $CO_2$ emissions by substituting fossil fuels when a new PV or wind power plant $\epsilon$ was built ($F_\epsilon$):

$$F_\epsilon = \theta_{\text{fossil}} \cdot E_\epsilon - \sum_{q=1}^{7} \sum_{x=1}^{n_q} (\gamma_x \cdot S_x) - \frac{\sum_{q=1}^{7} \sum_{x=1}^{n_q} (\upsilon_x \cdot S_x)}{25}, \ x \in q \tag{9}$$

in which $E_\epsilon$ is the total power generation, $S_x$ is the area of pixels installing PV panels or wind turbines, $\theta_{\text{fossil}}$ is the $CO_2$ emission factor of coal (0.84 kg $CO_2$ kWh$^{-1}$), oil (0.72 kg $CO_2$ kWh$^{-1}$) or gas (0.46 kg $CO_2$ kWh$^{-1}$)[58] that is substituted by PV and wind power, $\gamma_x$ is the flux of the terrestrial carbon sink disaggregated from a bottom-up estimate[59] and $\upsilon_x$ is the concentration of soil carbon in lands covered by vegetation[60] transferred to PV panels or wind turbines (Supplementary Table 9).

We derived the MAC for a new PV or wind power plant $\epsilon$, $\text{MAC}_\epsilon$, based on the abated $CO_2$ emissions:

$$\text{MAC}_\epsilon = \frac{\text{LCOE}_\epsilon \cdot E_\epsilon - \text{LCOE}_{\epsilon-1} \cdot E_{\epsilon-1} - \varrho \cdot (E_\epsilon - E_{\epsilon-1})}{F_\epsilon - F_{\epsilon-1}} \tag{10}$$

in which $\varrho$ is the price of coal, oil or gas. We obtained the prices of coal ($0.043 ± 0.015 per kWh as the 95% confidence interval)[61], oil ($0.141 ± 0.057 per kWh)[62,63] and gas ($0.058 ± 0.016 per kWh)[64] in China as the averages during 2010–2020, when they are considered

to generate electricity with an efficiency of 35%, 38% and 45%, respectively[65]. The generation of power by fossil fuel in the 2010s in China was dominated by coal, with a contribution[66] of 96% in 2020, but the composition of fossil fuel in the future is as yet unknown for China. We assumed that the future fossil-fuel composition in China will be constant in our central case, but we performed two sensitivity experiments to consider the impact of changes in fuel composition. First, the share of oil for generating power increases from the current level in 2020 (0.3%) to 50% in 2060 by substituting coal when the share of gas is identical to that in the central case. Second, the share of gas for generating power increases from the current level in 2020 (4.1%) to 50% in 2060 by substituting coal when the share of oil is identical to that in the central case (Extended Data Fig. 6).

### Hourly power demand

We predicted the hourly power demand by 2060 based on the flexibility of power loads by sector (Supplementary Table 10). First, we scaled up the historical hourly power loads from electrical grids[67] in 2018 by the increase in total power demand under the projected rate of electrification[33] in 2060 (58%) for six non-power sectors, including agriculture, industry, transport, building, service and household electric appliances, in 31 provinces. We assumed that the power loads are flexible for agriculture, industry, building, service and household electric appliances, except for heating and cooling in houses and electric cars, so we could simulate the profiles of the hourly power demand to match the hourly power generation by PV and wind endogenously in our optimization model. Second, we predicted the hourly power demand by electric cars. We obtained the profile of traffic flow in each street every five minutes in 2018 in Shenzhen[68], which was assumed to represent the variation of traffic flow in the future owing to a lack of data for other cities in China. When electricity is used by electric cars, one-third of vehicles are charged immediately, one-third are charged in one hour and one-third are charged in two hours[25].

Third, we predicted the hourly power loads for heating and cooling in houses. We obtained the hourly energy used for space heating and cooling in houses by region in China[69]. Finally, we considered the impact of temperature on the power demand for heating and cooling in houses and electric cars based on the projected temperature under climate warming. The electricity used for heating increases by 0.98% as the annual average temperature decreases by 1 °C when the hourly temperature is below 16 °C (ref. 70), whereas the electricity used for cooling increases by 0.63% as the annual average temperature increases by 1 °C when the hourly temperature is above 28 °C (ref. 25). We predicted the hourly power demand for heating and cooling for 31 provinces for 2021–2060 based on the gridded hourly temperature[71] averaged for 2016–2020 and the projected change in annual average temperature during 2021–2060 under the SSP1-2.6 scenario from an Earth System Model[17]. The power demand shifts in the daytime to match the peak of hourly PV and wind power generation (Supplementary Fig. 5).

### Actual PV and wind power plants

We adopted a pixel resolution of $1 \times 3$ km$^2$ for installing PV panels or wind turbines, which allows us to predict the location and capacity of individual PV and wind power plants in our optimization model. We cannot validate the locations and capacities of the projected PV or wind power plants that had not yet been built, so we used the locations and capacities of the commissioned PV and wind power plants in OpenStreetMap[31] that were closest to the projected PV or wind power plants to evaluate our prediction (Extended Data Fig. 3). We estimated the geographical distance of locations between the projected and actual PV and wind power plants. A full validation of our optimization model required detailed information on the PV and wind power plants when they are to be built in the coming decades, so we only compare the projected capacities of power plants normalized by current area with the actual capacities of the existing power plants in OpenStreetMap[31].

### Distribution of income

We estimated the impact of finances embodied in the flow of electricity generated by new PV and wind power plants on the redistribution of income in 2060. First, we estimated the distribution of income among the population at a county level based on the frequency distribution of income among the residents of urban and rural populations derived from a national survey[72] in 2015. We considered that the annual growth rate of population is 2% for Gansu, Inner Mongolia, Ningxia, Qinghai, Xinjiang and Xizang, which is higher than other provinces (1%) resulting from more new jobs and higher income created in these less-developed provinces[66]. Second, we compiled the per-capita disposable income for urban and rural populations at the county level[66] for 2015–2019. We made a linear projection of per-capita disposable income to 2060 based on the rate of growth of income by province during 2021–2060. We calibrated the growth rate of per-capita income for each income group at the county level during 2015–2060 to guarantee that the projected per-capita income as an average for each county in 2060 was equal to the projection for 2060. Given a carbon price ($\varsigma$), we assumed that only power plants with MACs below this carbon price were constructed. We estimated the revenue ($R_\epsilon$) from power generation when building a new PV or wind power plant ($\epsilon$):

$$R_\epsilon = \varrho \cdot E_\epsilon + \zeta \cdot F_\epsilon - LCOE_\epsilon \cdot E_\epsilon \tag{11}$$

in which $\varrho$ is the price of coal, oil or gas in China that is substituted by PV or wind power, $F_\epsilon$ is total abatement of $CO_2$ emissions, $E_\epsilon$ is total PV and wind power generation and $LCOE_\epsilon$ is the LCOE for the projected PV and wind power plants after building plant $\epsilon$.

Revenue from PV and wind power generation could be derived from the electricity price minus the LCOE, but the electricity price can be influenced by many socio-political factors[25,50]. We focused on analysing the impact of carbon price as a proxy for climate policies on the revenue of PV and wind power, so we considered that the electricity price depends on the fossil fuel prices and carbon price. The prices of fossil fuel may increase in the future owing to the scarcity of fossil fuel[73], which can increase the revenue of replacing fossil fuel with renewables, including PV and wind power. To estimate the prices of fossil fuel in 2060, we randomly draw the prices from the normal distributions, the average and standard deviations of which are estimated from the prices that are available from 2010 to 2020 (Extended Data Fig. 6). Predicting the impact of energy scarcity on the prices of fossil fuel and thus the revenue of PV and wind power is not considered in this study.

We represented the income inequality by dividing the population into 2,002 groups based on the order of income in each county. We excluded pixels in urban areas for the construction of utility-scale PV or wind power plants, so we allocated the revenue among the rural population in each county. When a carbon tax was levied on fossil fuels, we predicted the change in the per-capita income for each group in the population by considering the increased costs of power generation, the revenue from PV and wind power generation and the costs of carbon tax saved by reducing the use of fossil fuel (Supplementary Method 9). Finally, we estimated the income Gini coefficient in China using a formula[74] based on the changes in the fractions of income and population in each population group for 2,373 counties in 2060 when the carbon price increases from $0 to $100 per $tCO_2$.

### Uncertainty analyses

We estimated the uncertainties in the MAC and the Gini coefficient by running an ensemble of Monte Carlo simulations 40,000 times[75]. We randomly varied the parameters in these simulations, including: (1) the variability of PV power generation (±5%) over a suitable pixel ($W_{ijy}$) owing to the impact of aerosol deposition on PV panels[76] and the variability of wind power generation (±2%) over a suitable pixel ($W_{ijy}$) owing to the

impact of climate change on wind resources[77], (2) the growth rate of power demand by province during 2020–2060 (±1%)[7], (3) the parameters used in the calculation of initial investment costs based on the variability of capital costs (±10%) from previous estimates[55,56], (4) the historical rates of learning for different cost components measured in China (Supplementary Table 1) and (5) the parameters used for calculating the costs of UHV transmission and energy storage from different studies (Supplementary Table 8). Last, we adopted the medians of the MAC and the Gini coefficient to represent our best estimates, whereas we used the 90% uncertainties and interquartile ranges to represent their uncertainties.

## Data availability

Data used in this study are publicly available at the Zenodo repository: https://zenodo.org/record/7963012#.ZGzUd31Bw2w. General Electric model and Vestas model in the Windturbines database: https://en.wind-turbine-models.com/turbines; MCD12Q1 dataset: https://lpdaac.usgs.gov/products/mcd12q1v006/; GEOS-5 dataset: https://opendap.nccs.nasa.gov/dods/GEOS-5/fp/0.25_deg/ assim; Maritime Boundaries Geodatabase: http://www.vliz.be/en/ imis?dasid=5465&doiid=312; Mask of terrestrial ecological reserve: http://www.resdc.cn/data.aspx?DATAID=137; marine ecological reserve: https://www.protectedplanet.net/country/CHN; Radar Topography Mission (SRTM) Global Enhanced Slope (GES) dataset: https://lpdaac.usgs.gov/products/srtmgl1v003/; MERRA-2 dataset: https://disc.gsfc.nasa.gov/datasets/M2T1NXADG_5.12.4/ summary?keywords=SO2.

## Code availability

Further material is available in the Supplementary Information. The model used in this study can be accessed at the Zenodo repository: https://zenodo.org/record/7963012#.ZGzUd31Bw2w.

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

**Acknowledgements** We thank M. Harder for useful discussion. This study was funded by the National Key R&D Program of China (no. 2022YFF0802504), the National Natural Science Foundation of China (nos. 42341205 and 41877506) and the Chinese Young Thousand Talents programme. R.W., J. Chen, X.T. and R.Z. acknowledge support from the Shanghai International Science and Technology Partnership Project (no. 21230780200). K.T. benefited from state assistance managed by the National Research Agency in France under the Programme d'Investissements d'Avenir under the reference ANR-19-MPGA-0008. P.C. acknowledges support from the ANR CLAND Convergence Institute 16-CONV-0003. J.P. and J.S. acknowledge the financial support from the Catalan Government grant SGR 2021-1333, the Spanish Government grants PID2019-110521GB-I00, PID2020-115770RB-I00 and TED2021-132627B-I00 funded by MCIN, AEI/10.13039/501100011033 funded by the European Union Next Generation EU/PRTR and the Fundación Ramón Areces grant CIVP20A6621.

**Author contributions** R.W. initiated the study, led the project and designed the research. Y.W. compiled data, performed the research and prepared graphs. W.L., X.X., J.L., S.X., Y.X. and R.Y. provided analysing tools. K.T. contributed to economic analysis. P.C., K.T., J.P., Y.B., J.S. and D.H. contributed to energy analysis. K.T., J. Cao, J. Chen, L.W., X.T. and R.Z. contributed to policy analysis. R.W. wrote the first draft of the paper. All authors critically revised successive drafts of the paper and approved the final version.

**Additional information**

**Correspondence and requests for materials** should be addressed to Rong Wang.

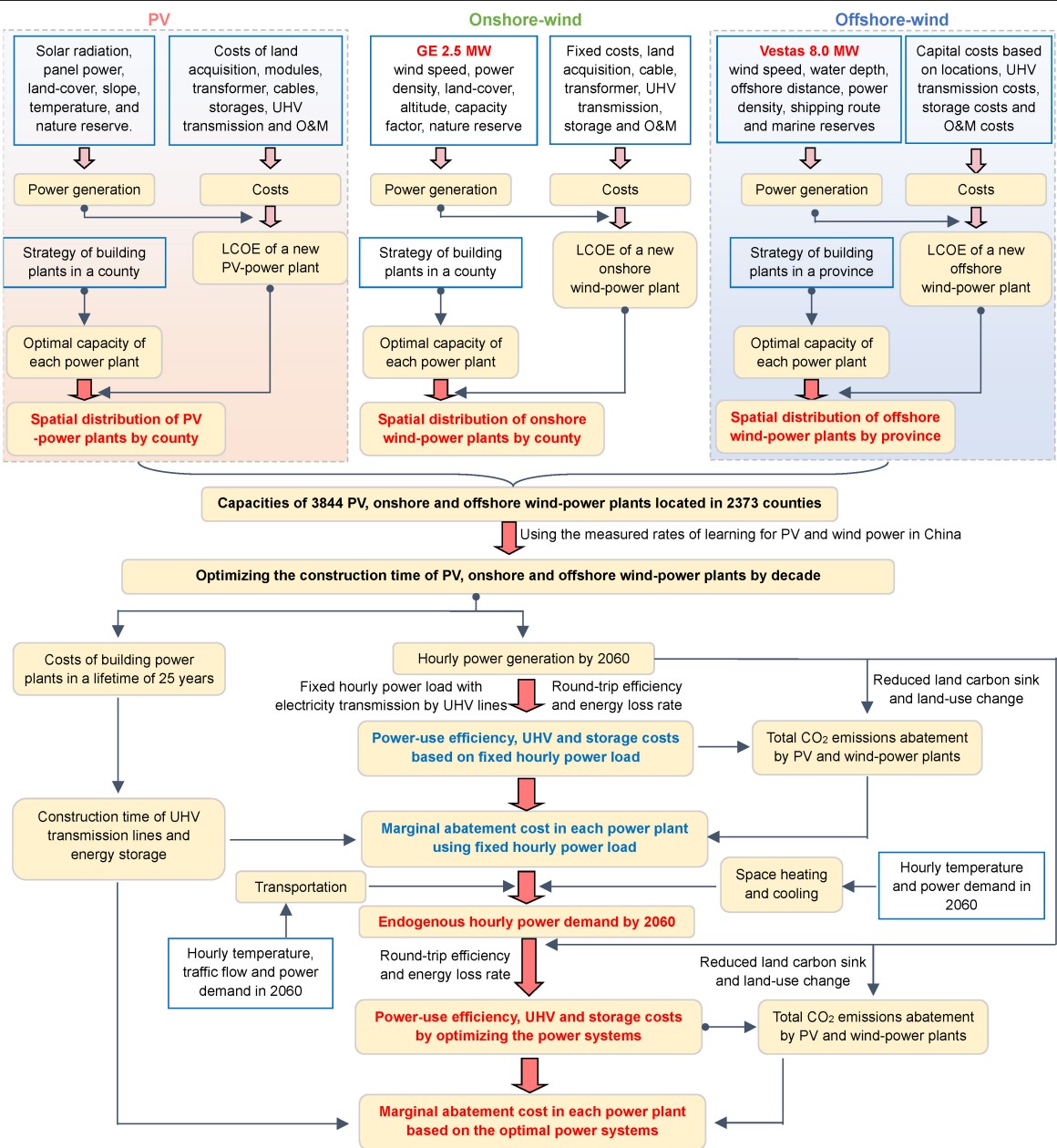

**Extended Data Fig. 1 | Procedures for optimizing the location, capacity and construction time of new PV and wind power plants in China.** The LCOE indicates the grid parity of PV and wind power generation coordinated with electricity transmission and energy storage in the power systems. We take the number of pixels installing PV panels or wind turbines and the construction time of each PV or wind power plant by decade as the decision variables to minimize the LCOE of all PV and wind power plants.

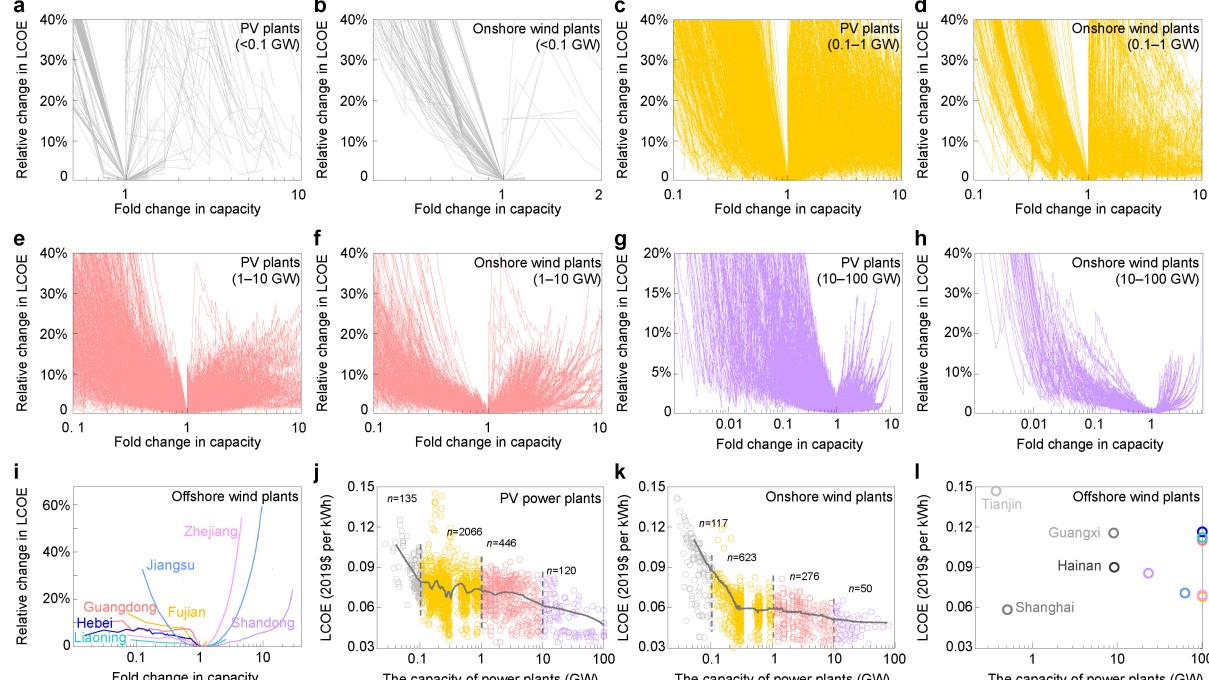

**Extended Data Fig. 2 | Impact of increasing the power capacity of PV or wind power plants on the LCOE. a–h**, Relative change in the LCOE as a percentage relative to the minimum when increasing the capacity of each PV (**a,c,e,g**) or onshore wind (**b,d,f,h**) power plant. The power plants are categorized into four groups, the optimized capacity of which falls in the range <0.1 GW (**a,b**), 0.1–1 GW (**c,d**), 1–10 GW (**e,f**) and 10–100 GW (**g,h**). **i**, Relative change in the LCOE as a percentage relative to the minimum when increasing the capacity of each offshore wind power plant. We estimate the fold change in the capacity of each power plant when increasing the number of pixels installing PV panels or wind turbines relative to the optimal capacity reaching the minimum of the LCOE for each power plant. **j–l**, Relationship between the minimum of the LCOE and the optimal capacity (<100 GW) of the projected PV (**j**), onshore wind (**k**) and offshore wind (**l**) power plants. The power plants are categorized into four groups based on the optimal capacity, for which the moving averages for 50 plants are shown by the grey lines in **j** and **k**.

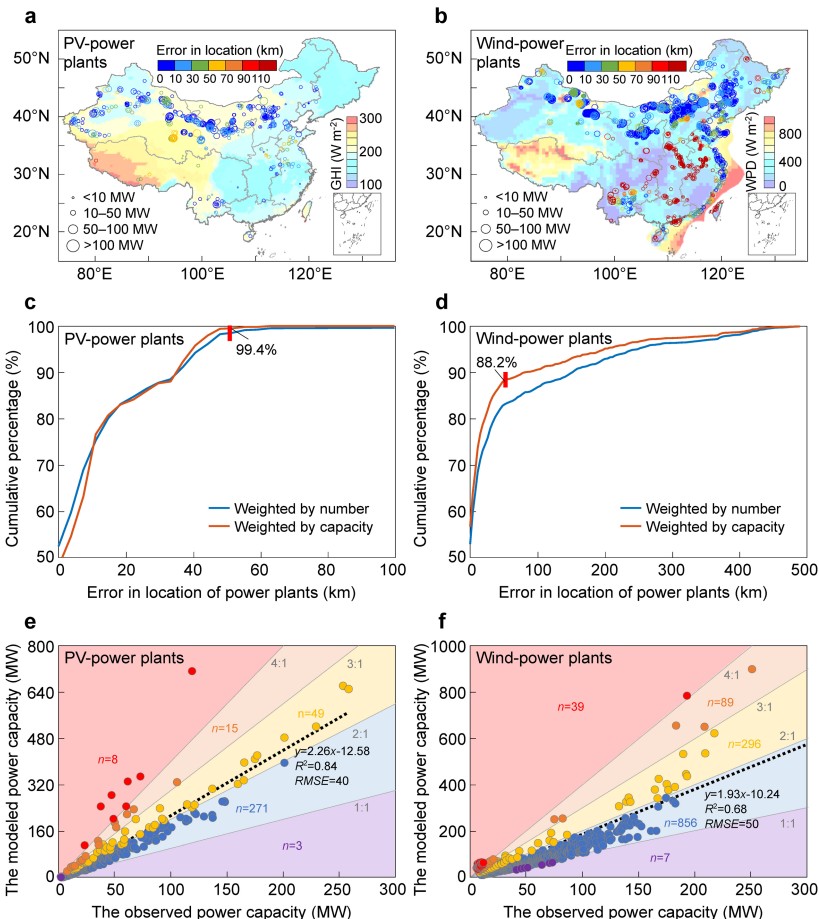

**Extended Data Fig. 3 | Evaluation of the location and capacity of PV and wind power plants in our optimization model. a,b,** Difference in the predicted location of PV (**a**) and wind (**b**) power plants in our optimization model relative to the location of PV panels and wind turbines installation observed by OpenStreetMap[31]. **c,d,** Frequency distribution of the difference in the location of PV (**c**) and wind (**d**) power plants weighted by the number of power plants (blue line) or the capacity of power plants (red line). **e,f,** Comparison of the projected capacity of PV (**e**) and wind (**f**) power plants in our optimization model normalized by the area of power plants with the capacity of PV and wind power plants observed by OpenStreetMap[31].

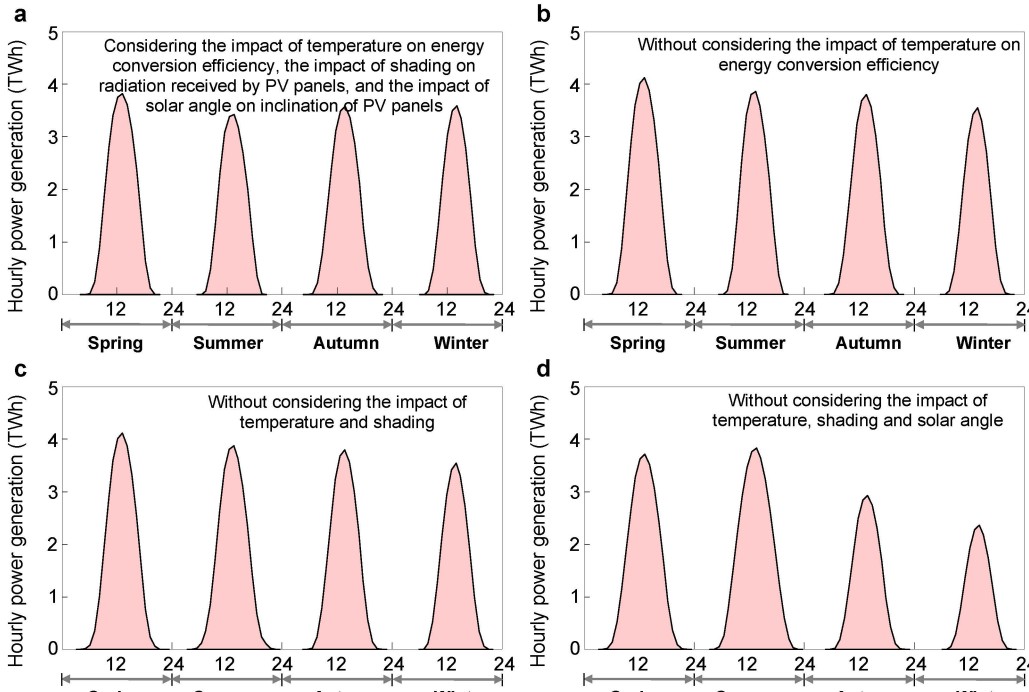

**Extended Data Fig. 4 | Seasonal variations in the hourly power generation by PV plants. a**, Hourly power generation by PV plants by considering the impact of temperature on energy-conversion efficiency, the impact of shading on radiation received by PV panels and the impact of solar angle on the inclination of PV panels. **b–d**, Hourly power generation by PV plants in sensitivity tests without considering the impact of temperature on energy-conversion efficiency (**b**), without considering the impact of temperature and shading (**c**) or without considering the impact of temperature, shading and solar angle (**d**).

**a**

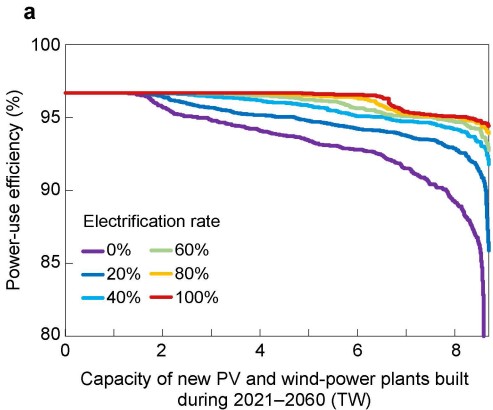

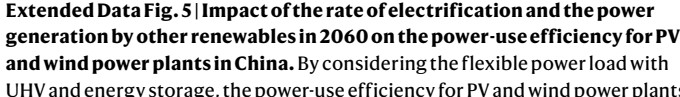

**b**

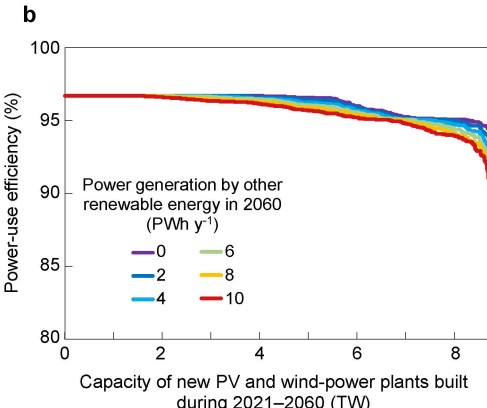

**Extended Data Fig. 5 | Impact of the rate of electrification and the power generation by other renewables in 2060 on the power-use efficiency for PV and wind power plants in China.** By considering the flexible power load with UHV and energy storage, the power-use efficiency for PV and wind power plants is estimated when the electrification rate in 2060 increases from 0 to 20%, 40%, 60%, 80% and 100% (**a**) and the power generation by other renewables in 2060 increases from 0 to 2, 4, 6, 8 and 10 PWh year⁻¹ (**b**).

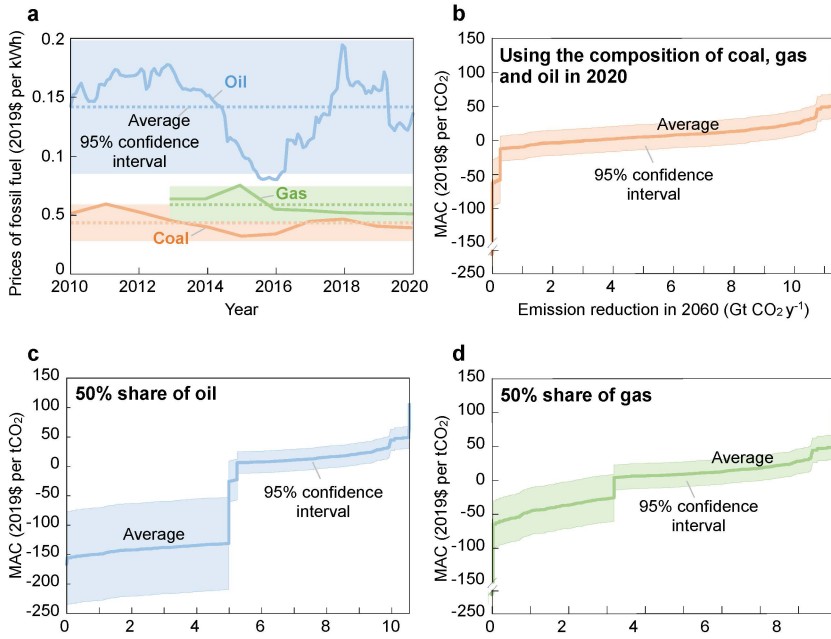

**Extended Data Fig. 6 | Impact of the fossil fuel composition on the MAC for PV and wind power plants in 2060. a**, Prices of coal, oil and gas for power generation in China during 2010–2020. The solid line denotes the average price, whereas the shading denotes the 95% confidence interval. **b**, MAC for PV and wind power plants in the central case, in which the consumption of coal, oil and gas for 2021–2060 are projected by scaling up the consumption in 2020 with the projected growth of total power demand. **c**, As in **b** but for a sensitivity case in which the share of oil in electricity generation increases linearly from the current level in 2020 (0.3%) to 50% in 2060 by replacing coal, but the share of gas is identical to the central case. **d**, As in **b** but for a sensitivity case in which the share of gas in electricity generation increases linearly from the current level in 2020 (4.1%) to 50% in 2060 by replacing coal, but the share of oil is identical to the central case.

**a**

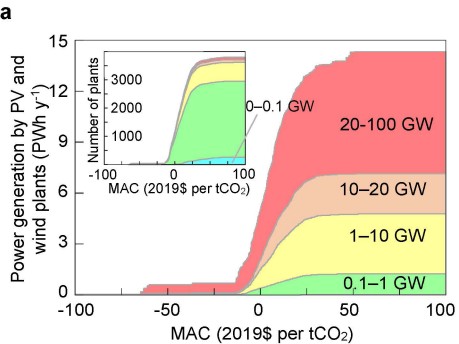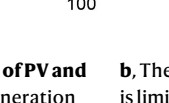

**b**

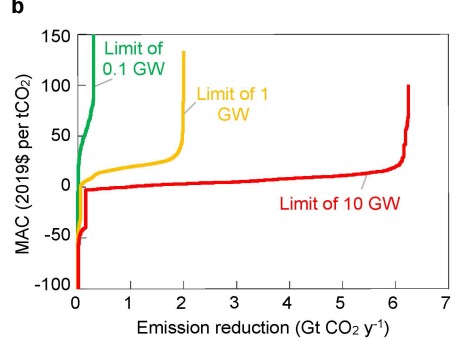

**Extended Data Fig. 7 | Impact of changing the limit of the capacity of PV and wind power plants on the MAC in 2060. a**, Composition of power generation by PV and wind plants. The inset shows the number of PV and wind power plants.

**b**, The MAC for PV and wind power when the capacity of individual power plants is limited to 0.1 GW (green), 1 GW (orange) and 10 GW (red), respectively.

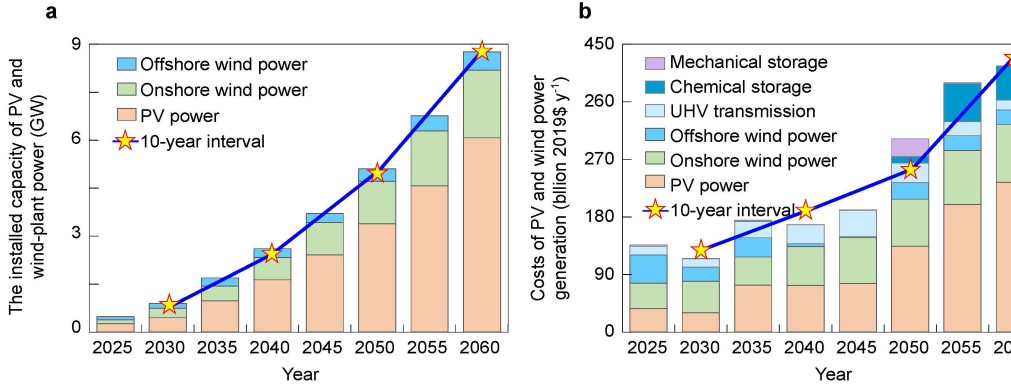

**Extended Data Fig. 8 | Impact of changing the interval of optimization on the installed capacity and costs of PV and wind power plants built during 2020–2060.** The installed capacity (**a**) and costs (**b**) of PV and wind power plants built during 2020–2060 are estimated in our model by optimizing the construction time of individual power plants at a temporal interval of 5 years (bars) or 10 years (stars).

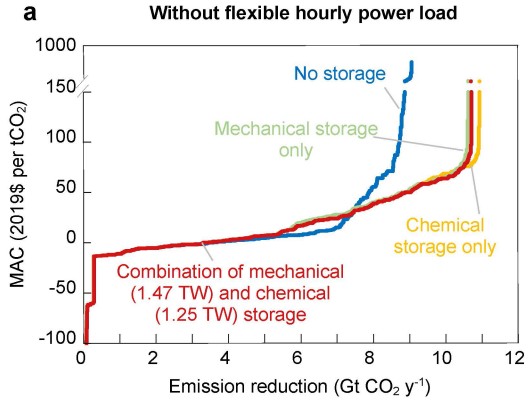
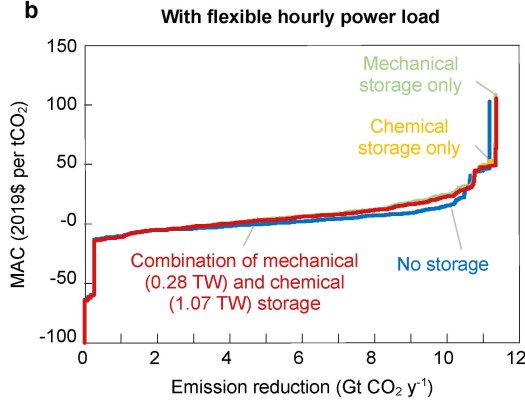

**Extended Data Fig. 9 | Impact of adopting different strategies of energy storage on the MAC of PV and wind power in China.** MACs of PV and wind power in 2060 are estimated in the scenarios without (**a**) or with (**b**) considering the flexible hourly power load. We compare MACs in the sensitivity tests without storing energy (blue line), using mechanical storage (pumped hydro) for all power plants (green line), using chemical storage (batteries) for all power plants (orange line) or using the optimized strategy of energy storage for each power plant in the central case (red line). We show the total capacity of mechanical (pumped hydro) and chemical (batteries) storage in the central case.

**Extended Data Table 1 | Distribution of pixels installing PV panels or wind turbines**

| Land-cover | Area (km²) | | | |
|---|---|---|---|---|
| | PV | Onshore-wind | Offshore-wind | Total |
| Closed Shrubland | 22 (0%) | 76 (0.01%) | 0 (0%) | 97 (0.01%) |
| Open Shrubland | 247 (0.04%) | 101 (0.02%) | 0 (0%) | 348 (0.03%) |
| Woody savanna | 26487 (4.5%) | 7323 (1.34%) | 0 (0%) | 33213 (3%) |
| Savanna | 44857 (7.7%) | 15759 (2.88%) | 0 (0%) | 59280 (5.3%) |
| Grassland | 226923 (39%) | 265175 (48%) | 0 (0%) | 392156 (35%) |
| Croplands | 0 (0%) | 70562 (13%) | 0 (0%) | 70562 (6%) |
| Vegetation mosaics | 0 (0%) | 4009 (0.73%) | 0 (0%) | 4009 (0.36%) |
| Snow, ice | 1137 (0.19%) | 0 (0%) | 0 (0%) | 1137 (0.1%) |
| Desert | 284989 (48%) | 184324 (34%) | 0 (0%) | 375351 (33%) |
| Water bodies | 0 (0%) | 0 (0%) | 188485 (100%) | 188485 (16%) |
| **Total** | **584661 (100%)** | **547329 (100%)** | **188485 (100%)** | **1124637 (100%)** |