## [Peer Review File · Nature]

Manuscript Title: Accelerating the energy transition towards photovoltaic and wind in China

Reviewer Comments & Author Rebuttals

Reviewer Reports on the Initial Version:

Referees' comments:

Referee #1 (Remarks to the Author):

This paper uses an optimisation model with spatial detail to investigate the timing and placement of variable renewable energy, storage, and transmission in China through 2060. The authors examine the sensitivity of their results to alternate assumptions about transmission and battery storage deployment, electrification, learning-by-doing, and others. Although there is a large literature examining deep decarbonisation pathways in China, the paper extends the literature by focusing on optimal placement and timing of wind and solar to meet the country's carbon neutrality goal.

The model and analysis are limited in their scope. The authors do not examine energy system or economy decarbonisation like the models in Duan et al (reference 1) or other carbon neutrality studies for China. And the electric sector representation appears limited by only modeling the investment and operations of a small subset of technologies, excluding non-renewable capacity, energy storage options beyond batteries, load flexibility, etc. As described below, many of the exogenous assumptions are also limiting (e.g., assuming static load shapes based on historical data, despite the expectation that electrification will play a key role in emissions reductions).

There are several shortcomings that should be addressed before the paper is considered for publication:

- Improve the representation of load shapes: The paper's current treatment of inflexible load and source for hourly shapes are inadequate for examining the role of renewable energies and storage. First, the authors should at the very least use the existing literature on deep decarbonisation impacts on load shapes to develop more defensible hourly shapes that reflect the electrification of transport, industry, etc. that they are assuming in their scenarios. The current approach of assuming static shapes from historical data does not reflect these changes. In addition, it would be good to see the authors extend their model to look at load shape flexibility and how that could influence optimal select and placement of renewables, storage, and transmission. Previous research (Brown et al 2018 and others) has indicated that emissions reductions strategies are sensitive to these dynamics.
- Clarify the paper's methodological contributions and reference broader literature: Similar approaches employed for other geographies that are not mentioned in the literature, including several papers that employ methodologies with greater endogeneity of other investments, end use decisions, and greater variety of technologies. A couple notable papers for the EU include Sasse and Trutnevyte (2020), Schlachtberger et al (2017), Eriksen et al (2017).
- Extend the model to include additional technologies: The absence of endogenous capacity investment in utility scale renewable technologies beyond wind and solar, rooftop PV, nuclear, CCS, energy storage options beyond batteries, etc. made the analysis feel limited relative to the Duan et

al multi-model study and to decarbonisation studies of other countries. This is especially important if you are looking at 'trade-offs of land, costs and electricity' (Line 112).

- Provide additional detail about the model and scenarios and re-organise the paper to make it easier to follow: I have several suggestions below for improving the flow of the paper and provided a clearer sense of inputs and outputs of the modeling framework in the main text. It took a couple reads and the Supplementary Materials to answer some of my questions while others didn't seem to be addressed at all. The exogenous method of handling load and bioenergy, nuclear, hydro, and hydrogen feel buried in the caption of Figure 2, which is over half way through the paper.

I had a number of additional suggestions as I reviewed the paper:

- Abstract: The first sentence should indicate somehow that this is a projection and consider adding a range to highlight the uncertainty about the future extent of wind and solar deployment. Perhaps mention 'capacity' (so not confused with peak demand)?
- Line 51: 'security' depends on a broad range of factors.
- Line 58: Lower LCOE costs but not necessarily lower system costs. After all, adding batteries to solar increases asset level LCOE but can decrease system costs, which is ultimately what matters. Grant et al is good reference, but it also illustrates value in CCS even where wind and solar are high.
- Lines 64-76: Needs more explanation about MAC curves, including their limitations and proper interpretation. Need to explain that negative MAC means that would happen without policy on the margin.
- Lines 73-74: Can also mention shortcomings of IAMs and their lack of temporal and technological detail (Keppo et al 2021, Victoria et al 2021).
- Lines 83-84: Why are climate impacts on hydropower and thermal plants omitted?
- Line 91: 'increased by 40%' relative to what?
- Line 94: Would better set up the paper to have a clearer explanation of the optimisation problem here in the main text. Also, calling this 'LCOE' is a little misleading, since it more accurately seems like the 'normalised net present value'. LCOE has a lot of baggage in the literature and many notable shortcomings in relation to not capturing changes in system value.
- Line 107: 'prediction' Projection probably better. Many difficult-to-model factors involved in siting, which means models are typically used to generate insights rather than predictive values.
- Lines 121-122: It would be good to reference your figures on capacity and generation mixes around the point in the paper so that a curious reader could see them.
- Line 124: Why not technological removals such as bioenergy with carbon capture and direct air capture? These technologies play large roles in the Duan et al paper.
- Lines 130-143: Is dynamic that learning-by-doing leads to frontloading of abatement effort? Similar finding in Victoria et al (2020). But raises question about appropriate discounting and spillovers both within China and internationally.
- Table 1: How is 'electricity income' calculated? Is this based on electricity prices coming from the dual variable on the market clearing constraint in your model or on exogenous electricity prices?
- Line 156: Is this because resource constrained? Why doesn't price adjust so that the net present value of cost equals revenue for built capacity in equilibrium (Brown and Reichenberg 2021)?
- Lines 161-163: Not conceptually appropriate. Mechanics of calculating MACs for wind and solar are very different from calculating shadow prices on carbon constraints in an economy wide model. These are not comparable metrics despite the same units. Thus, 'IAMs could overestimate the costs of mitigation' is too strong of a conclusion!

- Figure 1: This is an excellent visualisation that clarified many of the questions I had about the Table 1! I recommend that this come first in the text. A small suggestion is to use a bar chart (as the authors do elsewhere) to show costs instead of a pie chart.
- Lines 211-213: Why not marginal curtailment?
- Line 234: 'Electricity generation' instead of 'energy'? Important to clarify between electricity and energy throughout the paper.
- Line 236: 'projected' instead of 'predicted'.
- Line 247-248: How are regional and global spillovers addressed?
- Line 253: 'total energy supply' Is this electricity or energy? If the latter, is this primary energy or final energy?
- Figure 3: Another good multipanel visualization. For panel d, I recommend separating coal with and without CCS for clarity. In this panel, do you mean LCOE instead of 'electricity price'? 'Electricity price' seems to imply that this was an exogenous assumption.
- Lines 271-272: Capacity or generation assumed to be constant?
- Lines 473-475: The assumed lifetime of storage of five years is a severe underestimate! In the literature, lifetimes of 15-25 years are common for battery storage and longer for other options.
- Line 482: Where are the costs associated with building and operating other technologies in this objective function formulation?

Papers:

- Brown and Reichenberg (2021) "Decreasing market value of variable renewables can be avoided by policy action" Energy Economics
- Brown et al (2018) "Synergies of sector coupling and transmission reinforcement in a cost-optimised, highly renewable European energy system" Energy
- Eriksen et al (2017) "Optimal heterogeneity in a simplified highly renewable European electricity system" Energy
- Keppo et al (2021) "Exploring the possibility space: taking stock of the diverse capabilities and gaps in integrated assessment models" ERL
- Sasse and Trutnevyte (2020) "Regional impacts of electricity system transition in Central Europe until 2035" Nature Communications
- Schlachtberger et al (2017) "The benefits of cooperation in a highly renewable European electricity network" Energy
- Victoria et al (2021) "Solar photovoltaics is ready to power a sustainable future" Joule
- Victoria et al (2020) "Early decarbonisation of the European energy system pays off" Nature Communications

Referee #2 (Remarks to the Author):

The manuscript presents a comprehensive model of massive deployment of wind and solar PV power in China to facilitate its decarbonisation by 2060. The model is remarkable in its detail and complexity including high-granularity spatial disaggregation, taking into account the terrain, administrative boundaries of counties, climatic conditions, demand projections, learning rates, storage and transmission needs and opportunities. Its findings include a range of marginal

abatement cost of carbon (MACC) depending on the deployed solar and wind capacity, investment needs, and the electricity trade flows across different regions. Despite its sophistication, the model is well explained and, as far as I can judge, rigorous and reasonable.

An outstanding finding of the paper which would resonate beyond its immediate objective is demonstrating and quantifying the cost (LCOE) dependence on the quantity of wind and solar power deployed. It is particularly remarkable that LCOE and MACC significantly increase at higher deployment, even when learning is taken into account. This challenges a widespread view (e.g. [2]) that the costs of solar and wind will keep declining while their deployment will grow exponentially and that energy transition as a result will have net negative cost. Increasing marginal costs of deployment may also explain the S-shaped curves of wind and solar growth in many European countries [5]. It would be good if this insight could be made more prominent thus signalling the importance of the study beyond China and its contribution to the more general debate on energy transitions.

Another interesting finding is that about 2.6 TW of wind and solar can be deployed profitably (and require some \$1trln stimulus investment), while about 6 TW would need some subsidy associated with the price of carbon in the range of 50-100 USD/ton (if I read Fig. 1 and Table 1 right).

I believe that the main area of improvement for this manuscript would be to better highlight its novelty and implications as well as its limitations and how these should be addressed in future research. Since the manuscript attempts several different tasks I will explain this comment in relation to each of them.

Novelty and contribution

1. ****Wind and solar power deployment necessary and feasible for China's net zero by 2060 (CNZ).**** The manuscript starts with citing 20 TW of renewable energy necessary for CNZ (25-26).

Unfortunately I could not find this number in the cited ref 1. Instead it envisions (Fig 4-E) non-fossil electricity generation of 20 TWh/year consistent across models and broken down into solar, wind, hydro, nuclear etc. Both the units and the concept are different between ref. 1 and what is cited in the manuscript. In addition, the IAM models used for ref. 1 seem to envision very different levels of deployment of renewables. The manuscript could sharpen its contribution by indicating which of these different models produce more likely predictions.

* There are other studies quoted in the manuscript, but since they all use different concepts (capacity vs. generation, time horizons) and probably different targets (For example in 121-123 the units of generation (for PV and wind) and capacity (for BNHH) are mixed which is confusing), it should be better how the manuscript reduces the existing uncertainty in this matter. In other words, how does 8.6 TW of capacity of solar and wind by 2060 that the study sees as necessary and feasible relate to other estimates?

* It may also be that the message of the manuscript is more sophisticated than just giving a single number, but it only makes it more pressing to explain how the range of estimates (under different MACC/carbon prices) relate to previous estimates. Presenting this in a figure or table may be useful.

2. ****MACC of wind and solar power deployment in China.**** The manuscript does a fantastic job by demonstrating how MACC depends on many assumptions and how it varies with increasing capacity

of deployment and with various deployment strategies. It also provides a table of previous MACC estimates as a supplementary spreadsheet. However, these estimates are from very diverse studies, some of which are 10 years old or older. It would be useful to highlight which of the MACC estimates are state-of-the-art and how does the paper compares with those most current and rigorous estimates.

* Incidentally, the comparison (161-163) of the cost of mitigation in IAMs to MACC (from ref.1) should be better justified, because the former includes hard-to-decarbonise sectors and not only electricity and therefore it is logical that carbon prices in IAMs are higher than MACC for renewables. The Supplementary materials from Ref. 1 contain LCOEs which can be more directly compared with the findings of the manuscript

3. **Optimal location and timing of deployment of wind and solar power to reach CNZ.** The manuscript mentions other studies investigating this matter although apparently less sophisticated. How are the results of these other studies compare to those of the manuscript?

4. **Investment requirements for wind and solar power in China.** I really liked the differentiation between investments that can be provided as stimulus (i.e. for profitable plants) and the need for subsidies. But the same question: has this been estimated before? How do the results compare?

5. **Impacts of wind and solar power on economic inequality between regions in China.** Once again, it's an elegant method and an interesting finding, but is there other literature on this and how do the results compare? Or is it done for the first time?

6. **Beyond wind/solar and beyond China.** What are the implications of the novel method and new results for other technologies and deployment of wind and solar in other countries? China is a very large and diverse country and I believe many lessons would be relevant to other countries and regions. I think it would be important to know for wider Nature readership. I see several results potentially important for wider readership

1. The already mentioned documentation quantification of rising MACC with large levels of deployment might be very useful for modellers and for policy makers .

2. The fact that optimised deployment of wind and solar power requires mega plants located in remote regions and backed by interregional transmission and storage

3. The fact that large-scale deployment of wind and solar power can be done profitably to some extent (but requires massive investment and hence stimulus) and requires carbon tax or other subsidies to penetrate to substantial levels.

Limitations and connections with other literature

Though the manuscript uses a sophisticated and rigorous model it should still highlight its limitations based on how it frames and scopes the problem.

1. **MACC calculations.** Unless I am mistaken, these depend upon assumptions about the costs and composition (gas vs. coal) of fossil-fuel based power production. How was this modelled and what are the sensitivities?

2. As far as I understand, the paper envisions a large number of very large solar and wind power plants. Many of them are envisioned to be dozens or even hundreds of GW capacity. The plants of these size are unprecedented or rarely observed empirically (for a variety of non-economic reasons). At the same time, limiting plant size significantly reduces the abatement potential of wind and solar power in China (173-175). Furthermore, the model seems to predict much larger plant size than

empirically observed (Fig. S8). I would think therefore that the effect of limiting the plant size should be highlighted more prominently, including for MACC and other relevant calculations. The paper might want to mention the non-economic limitations of building such giant plants.

3. The manuscript focuses on the end-point of CNZ by 2060 but it does not analyse the rate of deployment required for reaching this point (beyond optimisation with respect to the learning rates). At the same time, the feasible rates of deployment of renewables and other low-carbon technologies have been of concern in prior studies e.g. [3-5]. Are the required rates of deployment comparable to those so far observed in different countries, including in those that devote a lot of resources to energy transitions (e.g. UK, Germany). This may also help to signal the overarching limitation of the manuscript which is focusing on economic aspects of renewables deployment whereas in real life they're also limited by socio-political factors.

4. It is not clear how the validation of the model with space observations was done and therefore what are the limitations of such validation.

Clarity

I find the figures presented in the manuscript very clear. At the same time I think the flow of text should be improved. In particular, the abstract contains many references but I think it should be more self-contained and explain the background and what are the main results. The rest of the text could be more directed at a wider audience not familiar with special concepts and the wider significance of the problems that the manuscript addresses should be explained. Conclusions should not repeat what the paper has done but rather focus on the implication of the results.

References

1. Duan, H. *et al.* Assessing China's efforts to pursue the 1.5°C warming limit. *Science* **372**, 378–385 (2021).
2. Ives, M. *et al.* A new perspective on decarbonising the global energy system*. (Smith School of Enterprise and the Environment, 2021). [https://www.ucl.ac.uk/climate-action-unit/sites/climate_action_unit/files/energy_transition_spm_ch.pdf](https://www.ucl.ac.uk/climate-action-unit/sites/climate_action_unit/files/energy_transition_spm_ch.pdf)
3. Kramer, G. J. & Haigh, M. No quick switch to low-carbon energy. *Nature* **462**, 568–569 (2009).
4. Wilson, C., Grubler, A., Bauer, N., Krey, V. & Riahi, K. Future capacity growth of energy technologies: are scenarios consistent with historical evidence? *Climatic Change* **118**, 381–395 (2013).
5. Cherp, A., Vinichenko, V., Tosun, J., Gordon, J. & Jewell, J. National growth dynamics of wind and solar power compared to the growth required for global climate targets. *Nature Energy* **6**, 742–754 (2021). <https://doi.org/10.1038/s41560-021-00863-0>

Referee #3 (Remarks to the Author):

General comment

The authors develop a spatially explicit approach to discuss the future construction and layout of wind and PV in China. The methodology of the study is detailed and transparent, and the robustness of the results is confirmed by extensive sensitivity analysis. However, the authors need to improve the methodology and refine the model to better fit the reality. In addition, the study need to take more policy considerations into account, especially the recent energy development plans of each province and region, to make it more policy influential and of public interest.

Line 67-68

The authors mention that MAC for PV and wind power varies from –US\$50 to US\$500 (tCO₂)⁻¹ in previous estimates (Supplementary Spreadsheet S1). However, many of these literatures provide incremental abatement cost (not MAC) or national/provincial level MAC (eg. “The national MAC is \$69/ton CO₂ in 2007 in China, and Beijing is the highest (\$417/ton CO₂) and Jiangsu is the lowest (\$0.1/ton CO₂)”)

Line 121-122

The authors mention that 14.4 PWh is half of the electricity supply in 2060. This value substantially exceeds the currently available studies related to IAM and energy system models.

Line 130-135

The authors assume the lifetime for PV and wind to be 40 years, much higher than other studies (20-30 years). Sensitivity analysis for the lifetime of PV and wind might need to be presented. In addition, although the study includes a sensitivity analysis for the discount rate, a discount rate of 7% may be slightly high for China. Using this discount rate as the central case may underestimate near-term effort requirements. In addition, after comparison, there is a discrepancy between the cost data of the 36th reference cited by authors and the data published by the China Electricity Council, to which the authors may make additional explanations.

Line 135-138

The study notes that the average capacity of these 80 plants is 32.5 GW. with current installed capacity of nuclear and thermal plants ranging from 1-4 GW, and with such a large individual plant set-up, the study needs to consider engineering feasibility.

Line 158-159

The main objective of the study is the layout of renewable energy sources, and the authors cite the IEA report on carbon neutrality in China as a value for the electrification rate, and perform sensitivity tests for the absence of electrification in the industrial and transportation sectors. In addition, the authors should refer to other studies that use energy system models to calculate cases that include higher electrification rates or perform sensitivity analysis directly on total electricity demand.

Line 196-197

This figure is the highlight of this paper. However, the information given in this figure is the result considering only optimality, so the layouts vary very much under different capacity levels of wind

and PV deployment, which will confuse decisionmakers. Currently, most provinces and regions already have their own renewable energy development plans, but they have not been considered in this study.

Line 233-238

The study uses the current electric load profile to characterize the future load profile, ignoring very critical factors such as temporal heterogeneity of the added load, demand-side response, and V2G. And these factors can have a very significant impact on the integration of energy storage and renewable energy.

Line 250-253

In the study, the average size of power plants reached 14 GW and 12 GW at both 2030 and 2040 time points, and both were distributed only in northwest and northern China (Fig. 1). However, this is quite different from the existing wind farm and PV plant sizes and future development plans. Does the study not consider the actual size and location distribution of existing PV plants or wind farms?

Line 269-272

In Figure 3d, except for the scenario where coal generation does not rise for the electricity price of \$0.09, the rebound in coal generation in 2060 after a steady decline until 2050 is very counterintuitive and inconsistent with realistic policies that the authors need to explain. At the same time, the authors assume a constant amount of nuclear, hydro, hydrogen and bioenergy generation in the future, which is also a counterfactual assumption.

Line 346-349

It is possible that the value of the \$1 trillion investment by 2040 may be relatively low due to the discount rate setting and investment cost assumptions.

Line 473-475

It is slightly conservative that electric energy storage facilities only have a five-year lifetime. From the supplementary information, it is clear that the authors assumed the lifetime of 8,000 cycles, i.e., more than four complete charge and discharge cycles per day, which may not correspond to the actual battery operation.

Line 525-539

It is unreasonable to calculate the power demand from thermal power by 2060 with the assumption of power sector sharing 80% of national total carbon emissions and as high as 5500 MtCO₂y⁻¹ carbon sinks. The emission factor of 0.827 kg CO₂ kWh⁻¹ for thermal power (including gas-fired power plants) is too high. In addition, why no CCS including BECCS considered for the year 2060 to achieve carbon neutrality?

Author Rebuttals to Initial Comments:

Referee #1: PV, wind power, MACC

Comment A1

This paper uses an optimisation model with spatial detail to investigate the timing and placement of variable renewable energy, storage, and transmission in China through 2060. The authors examine the sensitivity of their results to alternate assumptions about transmission and battery storage deployment, electrification, learning-by-doing, and others. Although there is a large literature examining deep decarbonisation pathways in China, the paper extends the literature by focusing on optimal placement and timing of wind and solar to meet the country's carbon neutrality goal.

The model and analysis are limited in their scope. The authors do not examine energy system or economy decarbonisation like the models in Duan et al (reference 1) or other carbon neutrality studies for China. And the electric sector representation appears limited by only modeling the investment and operations of a small subset of technologies, excluding non-renewable capacity, energy storage options beyond batteries, load flexibility, etc. As described below, many of the exogenous assumptions are also limiting (e.g., assuming static load shapes based on historical data, despite the expectation that electrification will play a key role in emissions reductions).

Response

Thank you very much for the positive comments. We are pleased to see the Reviewer appreciating that our paper extends the literature by clarifying the optimal placement and timing of solar and wind to meet the national target of carbon neutrality. Following the Reviewer's comments, we have improved our model by implementing the flexible power load profile and the optimal option between mechanical (pumped hydro) or chemical (batteries) storages under different rates of electrification. These revisions allow our model to better represent the demand-side responses to a large-scale deployment of PV and wind power in China (see our response to **Comment A2**).

However, following the Editor's comment, we continue to focus our paper on estimating the costs and potential of PV and wind power in China, thus without extending our model to cover all renewable energy. This strategy allows us to better examine the sensitivity of PV and wind power to global and regional spillovers, size of power plants, discounting, lifetime of power plants, rate of electrification, power generation by other renewables, flexible power load, marginal curtailment, changes in fossil fuel composition and using different data of capital costs. Nevertheless, after optimizing the flexible power load to match the hourly power generation with transmission and energy storage in our model, the impact of power generation by other renewable energy on the low MACs of PV and wind power is small in the new **Fig. S19** (see our response to **Comment A4**). These new results suggested that the low MACs to abate CO₂ emissions and thus the large benefits of deploying utility-scale PV and wind power are relatively robust.

To show the advantage of our spatially explicit method, we have also compared our estimate of PV and wind power to a recent multi-model study prescribing the MACC from the other regions of the world for China (Duan et al., 2021). It shows that the power generation of PV and wind power has been underestimated by up to 60% in these IAMs in the new **Fig. 3d** (see our response to **Comment A17**).

In addition, following the comments of **Reviewers #2**, we have further clarified the novelty of our paper by systematically comparing the projected power capacity, electricity generation, levelized cost of electricity (LCOE) and marginal abatement cost (MAC) (new **Fig. S1**) as well as the timing and placements of new plants, investment costs, inequality impacts

between our study and previous studies. By performing these comparisons, we believe that our study is important by revealing a larger contribution of PV and wind power to achieving carbon neutrality in China than previously thought.

Reference:

Duan, H. *et al.* Assessing China's efforts to pursue the 1.5°C warming limit. *Science* **372**, 378–385 (2021).

Comment A2

There are several shortcomings that should be addressed before the paper is considered for publication:

- **Improve the representation of load shapes:** The paper's current treatment of inflexible load and source for hourly shapes are inadequate for examining the role of renewable energies and storage. First, the authors should at the very least use the existing literature on deep decarbonisation impacts on load shapes to develop more defensible hourly shapes that reflect the electrification of transport, industry, etc. that they are assuming in their scenarios. The current approach of assuming static shapes from historical data does not reflect these changes. In addition, it would be good to see the authors extend their model to look at load shape flexibility and how that could influence optimal select and placement of renewables, storage, and transmission. Previous research (Brown et al 2018 and others) has indicated that emissions reductions strategies are sensitive to these dynamics.

Response

We agree very much with the Reviewer for accounting between feedbacks of decarbonization on load shares. We have optimized the flexible hourly power load on the demand side to match the hourly power generation by PV and wind and implemented the optimal option of energy storage between mechanical (pumped hydro) or chemical (batteries) storage for each power plant in our revised paper.

We have implemented the flexible hourly power load in our optimisation model. First, we determined the hourly power load by sector. For space heating and cooling in houses and electric cars, we considered the growth of power demand during 2023–2060 under different rates of electrification (0, 20%, 40%, 60%, 80% or 100%) and the impact of climate warming based on the dependence of heating and cooling power on temperature (Brown et al., 2018; Bistline, 2021), the projected hourly gridded temperature during 2023–2060 under the SSP1-2.6 and SSP2-4.5 scenarios (Gasser et al., 2017), the hourly traffic flow in 2018 in a typical city in China (Shenzhen Municipal Government data open platform, 2018) and the pattern of charging electric cars (Brown et al., 2018). For other sectors, we scaled the historical hourly power load from electrical grids in 2018 (National Development and Reform Commission, 2019) by the projected growth of power demand under different rates of electrification (0, 20%, 40%, 60%, 80% or 100%) during 2023–2060 (International Energy Agency, 2021). Second, for sectors except space heating and cooling in houses and electric cars, we optimized the flexible hourly power load to match the hourly power generation by PV and wind to minimize the LCOE when meeting a given target of emission abatements, which accounted for the responses of power load on the demand side to deploying large-scale renewable energy in China (Liu et al., 2021).

In addition, we optimized the option of energy storage for each power plant by comparing LCOE when deploying the mechanical storage (pumped hydro storage) with a lifetime of 50 years and a round-trip efficiency of 70% or the chemical storage (batteries) with a shorter lifetime of 15 years but a higher round-trip efficiency of 85% with different properties of

storages (see details in the new **Table S9**).

We examined the effects of optimizing the flexible power load and energy storage by performing three sensitivity experiments in the new **Fig. 1**. First, we fixed the hourly power load when the extra electricity that cannot be used on the demand side is not generated. Second, we fixed the hourly power load when the surplus of electricity generation by PV and wind that is not used on the demand side could be stored in pumped hydro or chemical batteries for use at a later time. Third, the hourly power load is optimized endogenously during 2023–2060 to match the hourly power generation by PV and wind as considered in the central case, but without deploying any energy storage.

We found that optimizing the flexible power load is effective in reducing the system costs of PV and wind power in the new **Fig. 1a**. We obtained the flexible hourly power load by decade during 2023–2060 (**Fig. S16**) and for 11 sectors (**Fig. S17**), which shifts in the daytime to match power generation by PV and wind due to adaptation of energy end-users (Liu et al., 2021). We also examined the impacts of optimizing the flexible power load on power-use efficiency and MAC under an electrification rate of 0, 20%, 40%, 60%, 80% and 100% in the new **Fig. 2e,f**. We found that optimizing the option of energy storage has the effect to reduce MAC when using the fixed power load, but is less effective when using the flexible power load because the hourly power generation can match well the power demand (**Fig. S18**).

To clarify the impacts of flexible power load, transmission and energy storage on the system costs of deploying PV and wind, we revised the following section in lines 294–336:

Impacts of flexible power load, transmission and energy storage

*The main solar and wind resources are identified in the West and North of China^{14,15}, but a large fraction of electricity needs to be transmitted to the East and South of China where the demand lies. The PV and wind power generation has diurnal variabilities (**Fig. S13**) and seasonal patterns, which does not match perfectly with diurnal and seasonal cycles of power demand¹⁴. The power generation peaks in spring due to variations in temperature, shade, solar angle and adjustment of inclination of PV panels (see factorial experiments in **Fig. S14**), but the demand of electricity peaks in summer due to electricity used for air conditioning. Global warming may increase the demand of cooling in summer, but reduce the demand of heating in winter⁴³. Different from previous studies in China^{14,15,28-32}, we considered load flexibility by modeling the response of power load profile to a large-scale deployment of PV and wind power (**Methods**). Beyond hourly power load for space heating, cooling and electric cars under the projection of temperatures⁴⁴, the hourly power load in the remaining sectors is optimized to match hourly power generation by PV and wind (**Table S1**). Optimisation of flexible power load reduces MAC by reducing the costs of energy storage. For example, under an electrification rate of 58% by 2060⁴¹, MAC of PV and wind to abate 8 Gt C y⁻¹ is reduced from 24.9 USD (t CO₂)⁻¹ using fixed power load to -1.7 USD (t CO₂)⁻¹ by optimizing the flexible power load even without deploying energy storage (Fig. 1).*

*The strategy of combining optimal energy storage with flexible power load improves the match between power demand and generation, thereby reducing marginal curtailment of renewable energy⁵¹. We assessed the impact of marginal curtailment by avoiding overinvestments in installed capacities with constraints of power demand⁵². The impact of marginal curtailment on MACC is large when using fixed power load without storage, but becomes small by optimizing the flexible power load with storage (**Fig. S15**). Based on the flexible power load by decade during 2030–2060 (**Fig. S16**) and for 11 sectors (**Fig. S17**), the power demand shifts in the daytime to match power generation by PV and wind under adaptation of energy end-users⁵³ (Fig. 2). When the capacities of PV and wind power reach 6 TW without UHV or storage, the power-use efficiency, defined as the fraction of hourly*

generated electricity used in final consumption, increases from 10% using fixed power load to 69% by optimizing the flexible power load. In addition, optimizing the option of energy storage for each power plant has the effect to reduce MAC using fixed power load, but becomes less effective after optimizing the flexible power load (Fig. S18). Interestingly, by optimizing power load to match power generation, the impacts of electrification rate (Fig. 2e,f) and power generation by other renewable energy (Fig. S19) on MAC both become small at the left side of MACC, indicating that low MACs to abate emissions by PV and wind are relatively robust with respect to uncertainties in the rate of electrification¹⁴ and deployment of other renewable energy in electricity systems².

To clarify our methods used to optimize the flexible power load and the option of energy storage, we added a new section in **Methods** in lines 771-821:

Flexible power load and energy storage

We simulated flexible hourly power load by decade during 2023–2060 (Fig. S16) and for 11 end-use sectors (agriculture, industry, building, service, electric cars, space heating, cooling, cooking, water heating, house electric appliance and others) (Fig. S17), which is coordinated with the optimal option between mechanical (pumped hydro) or chemical (batteries) energy storage for each power plant under climate warming. This consideration represents, to some extent, the demand-side responses⁴³ to the future large-scale deployment of PV and wind power in China.

The hourly power load profiles for electric cars, space heating and cooling were simulated by province based on the projected temperature under climate warming. We obtained hourly temperature at a resolution of 10 km in China during 2023–2060 based on the gridded hourly temperature averaged for 2016–2020⁸⁷ and change in annual mean temperature during 2020–2060 for China under SSP1-2.6 for the “1.5°C-limiting” scenario or SSP2-4.5 for the “no-policy” scenario in an Earth system model⁴⁴, due to the lack of hourly temperature in this Earth system model⁴⁴. First, for electric cars, we obtained the profile of traffic flow for each street per 5 minutes in 2018 in Shenzhen⁸⁸, which was assumed to represent the profile of traffic flow in future cities due to the lack of data for other cities in China. We scaled this hourly power load by the projected increase in electricity demand when 0, 20%, 40%, 60%, 80% and 100% of fossil fuel in the transportation sector was replaced by electricity during 2023–2060⁴¹. We considered the impact of temperature on the use of electricity for heating and cooling in electric cars: the electricity used for heating increases by 0.98% as temperature decreases by 1 °C when temperature is below 16 °C⁸⁹, while the electricity used for cooling increases by 0.63% as temperature increases by 1 °C when temperature is above 28 °C⁴³. When the electricity is used by electric cars, one third of vehicles are charged immediately, one third are charged in one hour and one third are charged in two hours⁴³. We simulated the hourly power load for electric cars in 31 provinces during 2023–2060. Second, in the residential sector, we obtained the hourly residential energy consumption used for space heating and cooling in the northern urban area, northern rural area, southern urban area and southern rural area, respectively⁹⁰. We considered the same impact of temperature change on residential heating and cooling as explained above. We scaled the hourly load profile by the projected increase in power demand when 0, 20%, 40%, 60%, 80% and 100% of fossil fuel in the residential sector was replaced by electricity during 2023–2060⁴¹.

For the remaining sectors, we optimized the flexible hourly power load. First, we scaled historical hourly power load from electrical grids in 2018⁴⁵ by the projected increase in electricity⁴¹ during 2023–2060. We assumed that the rate of electrification in these sectors was identical to that in the transportation and residential sectors. We then simulated the hourly power load by decade when the hourly power load profile is optimized to match the

hourly power generation by PV and wind by minimizing MAC when meeting a given target of CO₂ emission abatements (**Table S1**). In addition, we compared the LCOE for each power plant when using mechanical storage (pumped hydro storage) with a lifetime of 50 years and a round-trip efficiency of 70% or using chemical storage (battery storage) with a shorter lifetime of 15 years but a higher round-trip efficiency of 85%. The properties of these two energy storage systems are listed in **Table S9**. We sought the optimal system of energy storage for each power plant to minimize LCOE in the central case. We examined the impact of flexible power load by performing three sensitivity tests (see Fig. 1). First, by fixing the hourly power load without optimisation, the extra electricity that cannot be used on the demand side is not generated. Second, by fixing the hourly power load without optimisation, the surplus of electricity generation by PV and wind power that is not used on the demand side is stored by pumped hydro or chemical batteries for use at a later time. Third, the hourly power load is optimized to match the hourly power generation by PV and wind as considered in the central case, but without deploying energy storage.

To show the impacts of flexible power load, transmission and storage on MAC curve, we revised the **Fig. 1**:

Fig. 1 | Marginal abatement cost (MAC) curves for PV and wind-power plants in China by 2060. (a) MAC in the central case at plant level by coordinating PV and wind power with ultra-high-voltage (UHV) transmission, energy storage, learning and flexible hourly power load under a discounting rate (r_d)² of 5% y^{-1} . Relative to the central case, the sensitivity of MAC is tested to use of fixed hourly power load, exclusion of UHV, storages and learning, deployment of PV, onshore or offshore wind-power plants alone and consideration of a higher discounting rate¹⁵ ($r_d=7\% \text{ y}^{-1}$). The 90% uncertainty (gray shade) and inter-quartile range (blue shade) in MACs are derived from Monte Carlo simulations by randomly varying model parameters. **(b)** Composition of the levelized cost of electricity (LCOE) for PV and

wind-power plants contributing to installed capacities from 0 to 8 TW using fixed hourly power load (red frame) or optimizing the flexible hourly power load (blue frame), when other configurations are identical to the central case. (c) Location of the projected PV and wind-power plants and the optimal time to build each power plant based on the range of installed capacities in the central case. The color of symbol indicates the year of building each power plant, while the color of background indicates the abated CO₂ emissions in each region.

To show the impacts of flexible power load, transmission and storage on LCOE, MAC, power-use efficiency, we revised the Fig. 2:

Fig. 2 | Impacts of hourly power load, ultra-high-voltage (UHV) transmission and energy storage on the efficiency and costs of deploying PV and wind power in China. (a, b) Levelized cost of electricity (LCOE) (black lines) at plant level and power-use efficiency (PUE) for all plants (blue lines) when (a) using fixed power load or (b) optimizing the flexible power load to match the hourly power generation by PV and wind. (c, d) Seasonal variations in power generation and demand when (c) using fixed power load or (d) optimizing the flexible power load with or without UHV and storage when the installed capacities of PV and wind power reach 8 TW under an electrification rate of 58% by 2060⁴¹. (e, f) Changes in marginal abatement cost (MAC) curve when 0%, 20%, 60%, 80% or 100% of fossil fuel in non-power sectors is electrified when (e) using fixed power load without UHV or energy storage or (f) optimizing the flexible power load with UHV and storage.

To provide information on the power load by sector, a new **Table S1** was added to the **Supporting Information**:

Table S1. Simulation of the hourly power load by sector during 2023–2060.

Type	Sectors	Methods
We account for the power load profile change under climate warming	Cooling and Space heating	The hourly power load during 2023–2060 is simulated based on the projected power demand in the residential sector when achieving different electrification rates of 0, 20, 40, 60, 80 and 100% by 2060, the hourly gridded temperature in 2020 compiled from the Goddard Earth Observing System Model (https://gmao.gsfc.nasa.gov/GMAO_products/NRT_products.php) and the projected change in annual mean temperature during 2020–2060 over the China region under the SSP1-2.6 or SSP2-4.5 scenario in an Earth system model (Gasser et al., 2017).
	Transportation	The hourly power load during 2023–2060 is simulated based on the projected power demand in the transportation sector when achieving different electrification rates of 0, 20, 40, 60, 80 and 100% by 2060, the hourly traffic flow data in Shenzhen city in 2018, the hourly gridded temperature in 2020 compiled from the Goddard Earth Observing System Model (https://gmao.gsfc.nasa.gov/GMAO_products/NRT_products.php) and the projected change in annual mean temperature during 2020–2060 over the China region under the SSP1-2.6 or SSP2-4.5 scenario in an Earth system model (Gasser et al., 2017).
We optimize the hourly power load profile to match the hourly power generation by PV and wind	Agriculture	
	Industry	The hourly power load in each sector during 2023–2060 is simulated by decade when the hourly power load profile is optimized to match the hourly power generation by PV and wind to minimize MAC when meeting different targets of CO ₂ emission abatements. To obtain the hourly power load before this optimisation, the historical hourly power load profile from the provincial electrical grids in 2018 (National Development and Reform Commission, 2019) is scaled by the projected increase in electricity (International Energy Agency, 2021) during 2023–2060 when achieving different electrification rates of 0, 20, 40, 60, 80 and 100% by 2060.
	Building	
	Services	
	Cooking	
	Water heating	
	Household electric appliance	
Others		

To show the effects of optimizing the power load by decade, a new **Fig. S16** was added to the **Supporting Information**:

Fig. S16. Impact of a large-scale deployment of PV and wind power on the hourly power load from 2030 to 2060. Comparison between the fixed hourly power load before optimisation (a, c, e, g) and the optimized flexible hourly power load to match the power generation by PV and wind (b, d, f, h) in 2030 (a, b), 2040 (c, d), 2050 (e, f) and 2060 (g, h). Each line indicates the hourly power load as an average by month in a province.

To show the profile of optimized hourly power load by sector, a new **Fig. S17** was added to the **Supporting Information**:

Fig. S17. Hourly power loads under a large-scale deployment of PV and wind power in 2020. The optimized flexible hourly power load for all sectors (a), agriculture (b), industry (c), building (d), service (e), transportation (f), cooling (g) and space heating (h) in the residential sector, cooking (i), water heating (j), household electric appliance (k) and others (l). The flexible hourly power load in agriculture (b), industry (c), building (d), services (e), cooking (i), water heating (j), household electric appliance (k) and others (l) is optimized to match the hourly power generation by PV and wind. The hourly power load for space heating (h) and cooling (i) in the residential sector is estimated based on the projected hourly gridded temperature and the dependence of power demand on temperature. The hourly power load for transportation (f) is estimated based on the hourly traffic flow in 2019, the projected hourly gridded temperature and the dependence of power demand on temperature. Each line indicates the hourly power load as an average by month in a province.

To show the impact of optimizing energy storage on MAC curve, a new **Fig. S18** was added to the **Supporting Information**:

Fig. S18. Marginal abatement cost (MAC) curves for PV and wind-power plants in China by 2060 using mechanical (pumped-hydro) or chemical (batteries) storage. MAC curves using (a) the fixed hourly power load or (b) the optimized hourly power load in the scenarios without energy storages (blue line), deploying pumped-hydro storage only (green line), deploying chemical (batteries) storages only (orange line) and deploying the cheapest storage option (red line) for each power plant. The capacity of mechanical (pumped-hydro) or chemical (batteries) storages deployed in the projected power plants in the central case (red line) are given in the parentheses.

To show the properties of mechanical (pumped-hydro) and chemical (batteries) storages, a new **Table S9** was added to the **Supporting Information**:

Table S9. Parameters of the pumped-hydro storage and chemical storages.

Type	Pumped-hydro storage	Chemical batteries	Reference(s)
Lifetime (years)	50	15	Cole and Frazier, 2019; Chen et al., 2021
Round-trip efficiency (%)	70%	85%	Cole and Frazier, 2019; Chen et al., 2021
Energy loss rate (%)	0%	1%	Xiong and Singh, 2015
Per unit throughput (kWh kWh ⁻¹)	N/A	6000	Chen et al., 2021
Minimum/Maximum residual energy rate (%)	100%/0%	100%/0%	Chen et al., 2021
Energy-specific cost (2019 USD kWh ⁻¹)	2020	100	345
	2030	100	198
	2040	100	174
	2050	100	149
	2060	100	124

	2020	1200	595.73	
	2030	1200	374.45	
Power-specific cost (2019 USD kW ⁻¹)	2040	1200	327.22	Cole and Frazier, 2019; Hiesl et al., 2020
	2050	1200	280.78	
	2060	1200	234.34	
Optional cost (2019 USD kWh ⁻¹)	0.0015	0.0015		Zhang et al., 2016

References:

- Bistline, J. E. T. *et al.* Deep decarbonization impacts on electric load shapes and peak demand. *Environ. Res. Lett.* **16**, 094054 (2021).
- Bistline, J. E. T. & Blanford, G. J. Impact of carbon dioxide removal technologies on deep decarbonization of the electric power sector. *Nat. Commun.* **12**, 3732 (2021).
- Brown, T., Schlachtberger, D., Kies, A., Schramm, S. & Greiner, M. Synergies of sector coupling and transmission reinforcement in a cost-optimised, highly renewable European energy system. *Energy* **160**, 720–739 (2018).
- Cole, W. J. & Frazier, A. *Cost projections for utility-scale battery storage*. Report number: NREL/TP-6A20-73222. (National Renewable Energy Lab, Golden, CO, United States, 2019).
- Gasser, T. *et al.* The compact Earth system model OSCAR v2.2: description and first results. *Geosci. Model Dev.* **10**, 271–319 (2017).
- General Administration of Quality Supervision. *Inspection and Quarantine of the People's Republic of China. Indoor air quality standard. GB/T 18883–2002*. URL: <https://www.mee.gov.cn/image20010518/5295.pdf> (2002).
- He, J. *et al.* Towards carbon neutrality: A study on China's long-term low-carbon transition pathways and strategies. *Environ. Sci. Ecotechnol.* **9**, 100134 (2022).
- Hiesl, A., Ajanovic, A. & Haas, R. On current and future economics of electricity storage. *Greenh. Gases.* **10**, 176–1192 (2020).
- International Energy Agency. *An Energy Sector Roadmap to Carbon Neutrality in China*. (International Energy Agency, Paris, France, 2021).
- Liu, X., Zhao, T., Chang, C. T. & Fu, C. J. China's renewable energy strategy and industrial adjustment policy. *Renew. Energ.* **170**, 1382–1395 (2021).
- National Development and Reform Commission. *Typical power load curves of provincial power networks in China*. (2019).
- Shenzhen Municipal Government data open platform. Street real-time data. URL: https://opendata.sz.gov.cn/data/dataSet/toDataDetails/29200_00403589 (2018).
- Xiong, P. & Singh, C. Optimal planning of storage in power systems integrated with wind power generation. *IEEE T. Sustain. Energ.* **7**, 232–240 (2015).
- Zhang, T., Emanuel, A. E. & Orr, J. A. Distribution feeder upgrade deferral through use of energy storage systems. 2016 IEEE Power and Energy Society General Meeting. 1–5 (2016).

Zheng, X. & Wei, C. *Household energy consumption in China: 2016 report*. (Springer, Singapore, 2019).

Comment A3

• Clarify the paper's methodological contributions and reference broader literature: Similar approaches employed for other geographies that are not mentioned in the literature, including several papers that employ methodologies with greater endogeneity of other investments, end use decisions, and greater variety of technologies. A couple notable papers for the EU include Sasse and Trutnevyte (2020), Schlachtberger et al (2017), Eriksen et al (2017).

Response

To clarify the paper's methodological contribution by referring to studies of energy system modeling for Europe (e.g., Sasse et al., 2020; Schlachtberger et al., 2017; Eriksen et al., 2017) and the USA (e.g., Jacobson et al., 2016; Clark et al., 2017), we added the following sentence in lines 104-108: “*In contrast to previous studies^{2,14,15,28-32}, we developed an optimisation model with geospatial details of power plants, flexible power load, transmission, optimal energy storage and intertemporal dynamics of learning to estimate the MACC in the optimal path of deploying PV and wind power at plant level during 2023–2060 and assess the impact on poverty alleviation in China.*”, and added the following sentence in lines 78-81: “*Although a spatially explicit method has been applied to represent energy systems in Europe¹⁹⁻²¹ and USA^{22,23}, a similar approach is urgently needed to estimate marginal abatement cost (MAC) curves (MACCs) for utility-scale PV and wind-power plants in China as the largest CO₂ emitter with an ambitious target of achieving C neutrality by 2060⁶.*”.

References:

Clack, C. T. *et al.* Evaluation of a proposal for reliable low-cost grid power with 100% wind, water, and solar. *Proc. Natl. Acad. Sci.* **114**, 6722–6727 (2017).

Eriksen, E. H., Schwenk-Nebbe, L. J., Tranberg, B., Brown, T., & Greiner, M. Optimal heterogeneity in a simplified highly renewable European electricity system. *Energ.* **133**, 913–928 (2017).

Jacobson, M. Z., Delucchi, M. A., Cameron, M. A. & Frew, B. A. Low-cost solution to the grid reliability problem with 100% penetration of intermittent wind, water, and solar for all purposes. *Proc. Natl. Acad. Sci.* **112**, 15060–15065 (2015).

Sasse, J. P. & Trutnevyte, E. Regional impacts of electricity system transition in Central Europe until 2035. *Nat. Commun.* **11**, 1–14 (2020).

Schlachtberger, D. P., Brown, T., Schramm, S. & Greiner, M. The benefits of cooperation in a highly renewable European electricity network. *Energ.* **134**, 469–481 (2017).

Comment A4

• Extend the model to include additional technologies: The absence of endogenous capacity investment in utility scale renewable technologies beyond wind and solar, rooftop PV, nuclear, CCS, energy storage options beyond batteries, etc. made the analysis feel limited relative to the Duan et al multi-model study and to decarbonisation studies of other countries. This is especially important if you are looking at ‘trade-offs of land, costs and electricity’ (Line 112).

Response

Our study focused on estimating the MACC of PV and wind power using a spatially explicit method. These aspects have not been considered in previous studies such as refs 14, 15, 28–

32 cited in our main text. We believe that this novelty of our paper is in line with the Editor's comment: *"I should also point out that referee #1 requests an extension of your model by including other technologies. We will not ask for such extension, given your focus on wind and solar power. However, some caveats in this regard maybe necessary"*. Therefore, we continue to focus on PV and wind power without extending our model to cover all renewable energy. This strategy allows us to better examine the sensitivity of PV and wind power to global and regional spillovers, size of power plants, discounting, lifetime of power plants, rate of electrification, power generation by other renewables, flexible power load, marginal curtailment, changes in fossil fuel composition and using different data of capital costs.

Nevertheless, we showed that, after optimizing the flexible power load to match the hourly power generation with transmission and energy storage, the impact of power generation by other renewable energy on the low MACs of PV and wind power is small in the new **Fig. S19**. These results suggested that low MACs to abate CO₂ emissions and thus the large benefits of deploying utility-scale solar and wind power are relatively robust.

We agree very much with the Reviewer on the importance to deploy all types of renewable energy toward carbon neutrality in China. Given the importance of this topic, it is our intent to keep working on other renewable energy beyond PV and wind.

To clarify the importance of PV and wind power relative to other renewable energy in China, we revised the following sentence in lines 68-72: *"In general, photovoltaic (PV) and wind power has a wider range of applications than hydropower⁹, consumes less water than thermal solar power¹⁰, poses fewer safety issues than nuclear power¹¹, has fewer detrimental effects on food and ecosystems than bioenergy¹² and entails probably lower system costs than C capture and storage (CCS) at some locations¹³."*

To clarify the importance of modeling other renewable energy, we added the following sentences in lines 119-125: *"The majority of PV and wind power was projected to be affordable³⁶, so we focused on their MAC and estimated the first MACC for PV and wind power at plant level in China. While our method can be applied to other renewable energy and for other regions, MACC for China in most IAMs² was often prescribed from the other regions of the world. We assessed the sensitivity of MACC to socio-economic and technological factors, but the value of PV and wind power may decline if the penetration of other renewable energy is accelerated³⁷."*

To clarify the impact of power generation by other renewable energy on the MACC of PV and wind power, we added the following sentence in lines 329-334: *"Interestingly, by optimizing power load to match power generation, the impacts of electrification rate (Fig. 2e,f) and power generation by other renewable energy (Fig. S19) on MAC both become small at the left side of MACC, indicating that low MACs to abate emissions by PV and wind are relatively robust with respect to uncertainties in the rate of electrification¹⁴ and deployment of other renewable energy in electricity systems²."*

To show the advantage of our spatially explicit method, we also compared our estimate of PV and wind power with a recent multi-model study prescribing the MACC from the other regions of the world (Duan et al., 2021). It shows that the power generation of PV and wind power has been underestimated by up to 60% in the IAMs in the new **Fig. 3d**, confirming that it is important to examine the contribution of PV and wind power to carbon neutrality in China (see our response to **Comment A17**).

To show the impact of power generation by other renewable energy on the MACC for PV and wind, a new **Fig. S19** was added to the **Supporting Information**:

Fig. S19. The impact of power generation by other renewable energy on marginal abatement cost (MAC) of PV and wind power in China. MAC for PV and wind using (a) the fixed power load without UHV or energy storage or (b) the optimized flexible power load with UHV and storage when power generation by other renewable energy reaches 1, 5, 8 and 10 PWh y⁻¹ in 2060.

Reference:

Duan, H. *et al.* Assessing China's efforts to pursue the 1.5°C warming limit. *Science* **372**, 378–385 (2021).

Comment A5

• Provide additional details about the model and scenarios and re-organise the paper to make it easier to follow: I have several suggestions below for improving the flow of the paper and provided a clearer sense of inputs and outputs of the modeling framework in the main text. It took a couple reads and the Supplementary Materials to answer some of my questions while others didn't seem to be addressed at all. The exogenous method of handling load and bioenergy, nuclear, hydro, and hydrogen feel buried in the caption of Figure 2, which is over half way through the paper.

Response

Following this comment, we provided a summary of our modeling framework in a new **Method Summary** section at the beginning of our paper. We also provided details about optimisation of the flexible hourly power load in both **Method Summary** and **Methods** sections (see our response to **Comment A2**), brought forward **Fig. 1** to the beginning of our **Results** section (see our response to **Comment A21**) and moved the details about the projection of bioenergy, nuclear, hydro and hydrogen from the caption of **Fig. 2** to the **Method Summary** (see our response to **Comment A28**).

Comment A6

I had a number of additional suggestions as I reviewed the paper:

- Abstract: The first sentence should indicate somehow that this is a projection and consider adding a range to highlight the uncertainty about the future extent of wind and solar deployment. Perhaps mention ‘capacity’ (so not confused with peak demand)?

Response

We provided the range of electricity generation and power capacity from two references (Duan et al., 2021; Goldman Sachs Research, 2021) in the first sentence of **Abstract** in lines 24-26: “*The pledge toward carbon neutrality in China implies a power demand^{1,2} of 7,000–*

20,000 TWh y⁻¹ with capacities exceeding 8,000 GW¹ from renewable energy.”.

References:

Duan, H. *et al.* Assessing China’s efforts to pursue the 1.5°C warming limit. *Science* **372**, 378–385 (2021).

Goldman Sachs Research. Carbonomics: China net zero—The clean tech revolution. URL: <https://wwwqa.goldmansachs.com/insights/pages/gs-research/carbonomics-china-netzero/report.pdf> (2021).

Comment A7

- Line 51: ‘security’ depends on a broad range of factors.

Response

We agree with the Reviewer that energy security depended on many factors. To avoid this confusion, we revised “*energy security*” to “*energy transitions*” in line 63.

Comment A8

- Line 58: Lower LCOE costs but not necessarily lower system costs. After all, adding batteries to solar increases asset level LCOE but can decrease system costs, which is ultimately what matters. Grant et al is good reference, but it also illustrates value in CCS even where wind and solar are high.

Response

We agree with the Reviewer that adding batteries can increase asset level LCOE but decrease system costs. We also agree with the Reviewer that there is a value of CCS at some places (e.g. oil refineries) where the resource of wind and solar is promising. To clarify the value of PV and wind relative to CCS, we revised the following sentence in lines 68-72: “*In general, photovoltaic (PV) and wind power has a wider range of applications than hydropower⁹, consumes less water than thermal solar power¹⁰, poses fewer safety issues than nuclear power¹¹, has fewer detrimental effects on food and ecosystems than bioenergy¹² and entails probably lower system costs than C capture and storage (CCS) at some locations¹³.”.*

Comment A9

- Lines 64-76: Needs more explanation about MAC curves, including their limitations and proper interpretation. Need to explain that negative MAC means that would happen without policy on the margin.

Response

MACC is useful for policy makers (Jackson, 1991), but we agree with the Reviewer that MACC is limited for some reasons, e.g. “*MAC curves, for example, omit ancillary benefits of greenhouse gas emission abatement, treat uncertainty in a limited manner, exclude intertemporal dynamics and lack the necessary transparency concerning their assumptions.*” (Kesicki and Ekins, 2012). Our study addressed these limitations by estimating socio-economic benefits of deploying PV and wind power (e.g. Gini coefficient), treating uncertainty with a number of sensitivity experiments and performing Monte Carlo simulations, optimizing the timing and placements of PV and wind-power plants and evaluating the benefits of flexible power load, transmission and storage. We believe that addressing these points can strengthen the novelty of our paper.

To clarify a proper interpretation of MACC, we added the following sentence in lines 82-83:

“MAC, defined as the additional cost of avoiding one more unit of CO₂ emissions, is useful when policy makers seek cost-effective means to reduce CO₂ emissions²⁴.”

To clarify the limitations in the conventional method of MACC, we added the following sentence in lines 89-91: “Conventional MAC method is limited by overlooking ancillary benefits of energy transition, lacking consideration of system costs, representing uncertainty factors incompletely and ignoring intertemporal dynamics²⁵.”

We also agree very much with the Reviewer that negative values of MAC should be better explained in our paper. To clarify the implication of negative values of MAC, we revised the following sentence in lines 85-88: “MAC would become negative for utility-scale PV and wind power that is properly coordinated with transmission and storage¹⁴, indicating deployment of renewable energy with revenue on the margin.”

References:

Jackson, T. Least-cost greenhouse planning supply curves for global warming abatement. *Energ. Polic.* **19**, 35–46 (1991).

Kesicki, F. & Ekins, P. Marginal abatement cost curves: a call for caution. *Clim. Polic.* **12**, 219–236 (2012).

Comment A10

- Lines 73-74: Can also mention shortcomings of IAMs and their lack of temporal and technological details (Keppo et al 2021, Victoria et al 2021).

Response

To clarify the shortcoming of previous IAMs (Duan et al., 2021) in lacking the temporal and technological details (Keppo et al., 2021; Victoria et al., 2021), we added the following sentence in lines 101-104: “Spatial heterogeneity of PV and wind resources³³ and temporal dynamics of learning¹⁷ reduce the confidence in the projection of China’s energy transitions by integrated assessment models (IAMs) prescribing MACC from other regions².”

References:

Duan, H. *et al.* Assessing China’s efforts to pursue the 1.5°C warming limit. *Science* **372**, 378–385 (2021).

Keppo, I. *et al.* Exploring the possibility space: taking stock of the diverse capabilities and gaps in integrated assessment models. *Environ. Res. Lett.* **16**, 053006 (2021).

Victoria, M. *et al.* Solar photovoltaics is ready to power a sustainable future. *Joule* **5**, 1041–1056 (2021).

Comment A11

- Lines 83-84: Why are climate impacts on hydropower and thermal plants omitted?

Response

We agree with the Reviewer that considering climate impacts on other renewable energy is important. For example, the projected climate change will increase the potential of hydropower by 3 to 6 % in the 21st century (Liu et al., 2016). However, this is beyond the main scope of our study as we focused on estimating the costs and potential of PV and wind power (see our response to **Comment A4**). In our revised paper, we adopted the power generation by other renewable energy under the “no-policy” and “1.5°C-limiting” scenarios from a multi-model study (Duan et al., 2021) (see our response to **Comment A28**).

To clarify the possibility of extending our study to model other renewable energy, we added the following sentence in lines 121-123: “*While our method can be applied to other renewable energy and for other regions, MACC for China in most IAMs² was often prescribed from the other regions of the world.*”.

References:

Duan, H. *et al.* Assessing China’s efforts to pursue the 1.5°C warming limit. *Science* **372**, 378–385 (2021).

Liu, X., Tang, Q., Voisin, N. & Cui, H. Projected impacts of climate change on hydropower potential in China. *Hydrol. Earth Syst. Sci.* **20**, 3343–3359 (2016).

Comment A12

- Line 91: ‘increased by 40%’ relative to what?

Response

To clarify that this “40%” is relative to a case where we only considered building one power plant in each county, we revised the following sentence in lines 135-138: “*Optimizing the strategy of building power plants in each county increases total capacities of PV and wind power in China by 40% relative to a case where we only considered one power plant in a county (Fig. S6).*”.

Comment A13

- Line 94: Would better set up the paper to have a clearer explanation of the optimisation problem here in the main text. Also, calling this ‘LCOE’ is a little misleading, since it more accurately seems like the ‘normalised net present value’. LCOE has a lot of baggage in the literature and many notable shortcomings in relation to not capturing changes in system value.

Response

LCOE is widely used in the literature to estimate the grid parity (e.g. Tu *et al.*, 2019; Lorenczik, 2020), so it is useful to provide LCOE from our study to be comparable with the literature. We agree with the Reviewer that the conventional method to calculate LCOE is subject to the problem of lacking system costs (<https://www.wri.org/insights/insider-not-all-electricity-equal-uses-and-misuses-levelized-cost-electricity-lcoe>), but our study represented the costs of electricity systems by including the costs of transmission and energy storage in our calculation of LCOE.

To clarify the target of our optimisation and the calculation of LCOE by including system costs in our model, we added the following sentences in lines 141-146: “*To represent grid parity of renewable energy, we followed the method to calculate LCOE³⁹, defined as the normalized net-present-value of costs of initial investments, operation and maintenance (O&M), land acquisition, transmission and energy storages divided by electricity generation over the lifetime of a power plant. By minimizing LCOE, we optimized the location and size of power plants with transmission of the electricity to end users based on flexible power load (Methods).*”.

References:

Lorenczik, S. *et al.* Projected costs of generating electricity-2020 edition. (Organisation for Economic Co-Operation and Development, Paris, France, 2020).

Tu, Q., Betz, R., Mo, J., Fan, Y. & Liu, Y. Achieving grid parity of wind power in China—Present levelized cost of electricity and future evolution. *Appl. Energ.* **250**, 1053–1064

(2019).

Comment A14

- Line 107: ‘prediction’ Projection probably better. Many difficult-to-model factors involved in siting, which means models are typically used to generate insights rather than predictive values.

Response

We agree very much the Reviewer that there are many difficult-to-model factors such as ecological impacts (Hernandez et al., 2014), engineering feasibility (Schmidt et al., 2009), political impediments (Sovacool et al., 2009) and public perceptions of renewable energy (Irfan et al., 2021).

To clarify the impact of non-economic factors, we added the following sentences in lines 590-599: “*Many non-economic factors limit the size of power plants, such as ecological impact⁷³, engineering feasibility⁷⁴ and political impediment⁷⁵, but a key issue might be the legacy of monopoly control¹⁶ in China. The debate of the size of power plant largely depends on the ownership of electricity systems, who deploys capitals to bolster transmission and storage systems and earns a return on those investments¹⁶. In China, the central government has the control of electricity systems⁷⁶ and has ability to cover the required investments in system upgrade, but the problem is how to distribute the revenue of investments. These socio-political factors⁷⁶ may slow the pace of building large PV and wind-power plants, affect the prices of power generation by PV and wind and increase overall costs of emission mitigation.*”.

References:

Irfan, M., Elavarasan, R. M., Hao, Y., Feng, M. & Sailan, D. An assessment of consumers’ willingness to utilize solar energy in China: End-users’ perspective. *J. Clean. Prod.* **292**, 126008 (2021).

Hernandez, R. R. et al. Environmental impacts of utility-scale solar energy. *Renew. Sustain. Energ. Rev.* 29, 766–779 (2014).

Schmidt, R. C. & Marschinski, R. A model of technological breakthrough in the renewable energy sector. *Ecol. Econ.* 69, 435–444 (2009).

Sovacool, B. K. Rejecting renewables: The socio-technical impediments to renewable electricity in the United States. *Energ. Polic.* 37, 4500–4513 (2009).

Comment A15

- Lines 121-122: It would be good to reference your figures on capacity and generation mixes around the point in the paper so that a curious reader could see them.

Response

To clarify the generation mixes of electricity, we revised the following sentence in lines 511-518: “*The power generation by PV (9.2 PWh y^{-1}) and wind (5.9 PWh y^{-1}) represents half of power demand projected for 2060, thereby reducing the demand of CCS from 10.9 to 2.7 PWh y^{-1} under the projected power² of oil (0.01 PWh y^{-1}), gas (0.65 PWh y^{-1}), bioenergy (0.74 PWh y^{-1}), hydro (1.74 PWh y^{-1}) and nuclear (2.72 PWh y^{-1}) to achieve C neutrality in the “1.5°C-limiting” scenario (Fig. 3d), while the remaining CO₂ emissions will be offset by terrestrial C sink (1.87 Gt CO₂ y^{-1})⁴⁸.*”

Comment A16

- Line 124: Why not technological removals such as bioenergy with carbon capture and direct air capture? These technologies play large roles in the Duan et al paper.

Response

We estimated the demand of CCS rather than BECCS, because CCS might be deployed when retrofitting coal, gas, oil or biomass-fired power plants, while biomass might be used to produce cellulosic ethanol and Fischer-Tropsch biofuels without CCS (Fajardy et al., 2019). We focused on the costs and potential of PV and wind power based on the projection of oil, gas, nuclear, hydro and bioenergy under the “no-policy” and “1.5°C-limiting” scenarios from a multi-model study (Duan et al., 2021), so it is necessary to avoid limiting the deployment of CCS to biomass in our paper.

To clarify the reason of estimating the demand of CCS in our paper, we added the following sentences in lines 174-184: “*Given uncertainties in the geological constraints on CCS⁴⁷, it is interesting to know the demand of CCS needed to achieve China’s C neutrality, which depends on terrestrial C sink (1.87 Gt CO₂ y⁻¹)⁴⁸, rate of electrification (58%)⁴¹, total electricity demand⁴¹ and future power generation by PV, wind and other low-carbon energy². We adopted total power generation by oil, gas, bioenergy, nuclear and hydropower as the average of three IAMs (SWITCH, IPAC and GCAM_TU) under the “no-policy” and “1.5°C-limiting” scenarios from a multi-model study². The power generation by PV and wind was estimated according to MACC from this study under the “no-policy” and “1.5°C-limiting” scenarios based on the average prices of carbon in the IAMs². Assuming that coal provides the remaining electricity, we estimated total demand of CCS in electricity systems to meet the national target of C neutrality⁶.*”.

References:

Duan, H. et al. Assessing China’s efforts to pursue the 1.5°C warming limit. *Science* **372**, 378–385 (2021).

Fajardy, M., Koeberle, A., MacDowell, N. & Fantuzzi, A. BECCS deployment: a reality check. URL:

<https://www.imperial.ac.uk/media/imperial-college/grantham-institute/public/publications/briefing-papers/BECCS-deployment---a-reality-check.pdf> (2019).

Comment A17

- Lines 130-143: Is dynamic that learning-by-doing leads to frontloading of abatement effort? Similar finding in Victoria et al (2020). But raises question about appropriate discounting and spillovers both within China and internationally.

Response

Learning-by-doing leads to frontloading of abatement efforts, which is shown by the peak of rate of deploying PV and wind plants in 2040s in the new Fig. 3a. We agree very much with the Reviewer for examining the impacts of discounting and spillovers on the intertemporal dynamic of learning (Victoria et al., 2020). To consider the impact of discounting, we performed a sensitivity test under a high discounting rate of 7% y⁻¹ (Lu et al., 2021), which was compared with our central case adopting a discounting rate of 5% y⁻¹ (Duan et al., 2021) in the new Fig. 3a. It showed that adopting a higher discounting rate delayed the deployment of PV and wind, while the fraction of PV and wind power installed by 2050 in total power installed by 2060 decreased from 77 to 75%.

The impact of spillovers within China has been partly represented when we modeled the decline of capital costs as the installed capacities of PV and wind increase. Although we are

unable to estimate the impact of global spillover before we can extend our model from the region to the globe, we compared the relative increase in the projected capacities of PV and wind power between China and the other regions of the world by gathering data in the literature (Carrara et al., 2020; Edenhofer et al., 2011; IEA, 2017). We further performed additional experiments by applying the projected decline in capital costs of PV and wind based on the relative increase in the installed capacities in the other regions of the world. When we applied the projected rates of growth of PV and wind power in the other regions of the world, it led to a slower decline of capital costs during 2020–2060 than our central case based on the projected increase in the capacities of PV and wind power in China (**Fig. S27**). The projected increase in offshore wind power is larger in Europe than for China during 2030–2050, indicating that capital costs may decline at a higher rate by considering regional spillovers, but the installed capacities of offshore wind are far lower than PV and onshore wind in our projection. These new results suggested that the impact of spillovers might be dominated by the transfer of technologies from China to the other regions of the world due to a faster growth of the installed capacities of PV and onshore wind power in China when meeting the ambitious target of carbon neutrality.

To clarify the impact of global and regional spillovers, we added the following sentences in lines 555-567: “*Learning-by-doing¹⁷ brings abatement efforts forward, where the projected PV and wind capacities will be completed by 80% by 2050. The time of deploying renewable energy depends on appropriate discounting and technological spillovers⁶⁶. Using a discounting rate of 7% y^{-1} by Chen et al.¹⁵ rather than 5% y^{-1} by Duan et al.², the fraction of PV and wind capacities installed by 2050 will decrease from 77 to 75% (Fig. 3a). The effect of global spillovers on the rate of deploying PV and wind in China might be moderate, because the increase in capacities of PV and onshore wind power is faster in China than that projected for the other regions of the world⁶⁷⁻⁶⁹. If we apply the relative increase in global or European PV and wind capacities, the decline of capital costs would be slower than we estimated (**Fig. S27**). The projected increase in offshore wind power is faster in Europe⁶⁷ than for China during 2030–2050, implying a faster decline of capital costs by considering the technological transfer from Europe to China, but the installed capacities of offshore wind are far lower than for PV and onshore wind in our projection.*”.

To show the impact of discounting on the deployment of renewable energy in China, we revised the **Fig. 3** as:

Fig. 3 | Optimal path to achieve C neutrality by building PV and wind-power plants in China by 2060. (a) Dependence of LCOE of the projected PV and wind-power plants built by 2060 on the share of electricity generation by PV and wind-power plants built in 2020s in total electricity demand in 2060. The optimal path minimizes LCOE of the projected power plants in 2060 by randomly varying the time each plant is built by decade (the shaded area) under a discounting rate (r_d) of 5% y^{-1} in the central case² or 7% y^{-1} in the sensitivity case¹⁵ (red). The insert shows the share of electricity generation by PV and onshore wind and offshore wind-power plants built by decade in total electricity generation in 2060 under the optimal path. (b, c) The (b) time and (c) LCOE of building each PV or wind-power plant in the optimal path. (d) Composition of power generation by decade. Power generation by oil, gas, bioenergy, nuclear and hydropower is derived as the average of three models (SWITCH, IPAC and GCAM_TU) under the “no-policy” and “1.5°C-limiting” scenarios from a multi-model study², while power generation by PV and wind is derived from MACC in this

study under the two scenarios based on the prices of carbon in the models², respectively. Assuming that coal meets the remaining power demand, the demand of CCS (red dot) is estimated to meet the national carbon neutrality. The estimate of PV and wind power in a previous study² and the resultant demand of CCS are shown by the hatched bar and the blue dot, respectively. (e, f) Spatial distributions of (e) PV and (f) wind-power plants constructed by decade under the optimal path. The size and color of the symbol denote the installed capacity and the time to build each power plant. The background shows the global horizontal irradiance (GHI) and wind-power density (WPD).

To show the impact of potential global spillovers on the deployment of renewable energy in China, a new **Fig. S27** was added to the **Supporting Information**:

Fig. S27. Comparison of the projected capacity and decline of capital costs for PV and wind-power plants between China and other regions. (a) Comparison of the projected relative increase in the installed capacity of PV and onshore and offshore wind power plants in China from this study during 2020–2060 and in Europe by Carrara et al. (2020) and the projected global PV and wind power estimated by the International Energy Agency (IEA) (IEA, 2017) or Edenhofer et al. (2011). The installed capacity in 2020 is normalized to 1 for each estimate. (b) The projected decline of capital costs of PV and onshore and offshore wind-power plants due to learning-by-doing during 2020–2060 based on the projected relative increase in the installed capacity of PV and onshore and offshore wind-power plants. (c) Sensitivity of the marginal abatement cost (MAC) curve for PV and wind power in China to using the projected decline of capital costs based on the relative increase in the installed capacity of PV and wind power in China from this study, based on the relative increase in the installed capacity of PV and wind power in Europe (Carrara et al., 2020) or based on the relative increase in the installed global capacity of PV and wind power from two different estimates (Edenhofer et al., 2011; IEA, 2017).

References:

- Carrara, S. *et al.* Raw materials demand for wind and solar PV technologies in the transition towards a decarbonised energy system. Luxembourg: Publications Office of the European Union. URL: https://eitrawmaterials.eu/wp-content/uploads/2020/04/rms_for_wind_and_solar_published_v2.pdf (2020).
- Duan, H. *et al.* Assessing China's efforts to pursue the 1.5°C warming limit. *Science* **372**, 378–385 (2021).
- Edenhofer, O. *et al.* Renewable energy sources and climate change mitigation: Special report of the intergovernmental panel on climate change. Cambridge University Press. URL: https://www.ipcc.ch/site/assets/uploads/2018/03/SRREN_Full_Report-1.pdf (2011).
- International Energy Agency (IEA). *Energy technology perspectives 2017: Catalysing Energy Technology Transformations* (2017).
- Lu, X. *et al.* Combined solar power and storage as cost-competitive and grid-compatible supply for China's future carbon-neutral electricity system. *Proc. Natl. Acad. Sci. U.S.A.* **118**, e2103471118 (2021).
- Victoria, M., Zhu, K., Brown, T., Andresen, G. B. & Greiner, M. Early decarbonisation of the European energy system pays off. *Nat. Commun.* **11**, 1–9 (2020).

Comment A18

- Table 1: How is 'electricity income' calculated? Is this based on electricity prices coming from the dual variable on the market clearing constraint in your model or on exogenous electricity prices?

Response

The prices of electricity in China are under control of the central government, while they are subsidized by the central government and could be lower than actual costs of electricity systems (Lam, 2004). We realize that it is unreasonable to assume the exogenous prices of electricity, so we replaced the calculation of "electricity income" based on prices of electricity with the new calculation of revenue of transiting fossil fuel to PV or wind energy based on the prices of coal, oil and gas and the prices of carbon in the new Table 1. We estimated this revenue by subtracting the system costs of electricity generation from the total income by replacing fossil fuel based on the prices of coal, oil and gas and the carbon tax based on the prices of carbon and the abated emissions from coal, oil and gas. By analyzing the fluctuation of prices for coal, oil and gas during 2010–2020, we applied the average prices as well as their uncertainties in our model to estimate MAC and the revenue for transiting fossil fuel to PV or wind energy in China.

In addition, following the comment of **Reviewer #2**, we have also considered the impact of changes in the future composition of fossil fuel on MAC and the revenue (see our response to **Comment B9**).

To clarify our calculation of the revenue of transiting fossil fuel to PV or wind energy in China, we added a footnote in the new Table 1 in lines 573-577: "*b, The revenue of transiting fossil fuel to PV or wind-power plants is calculated by subtracting the overall costs of electricity generation from the income by replacing fossil fuel based on the costs of fossil fuel in the electricity generation and the tax of carbon emissions from coal, oil and gas under a carbon price⁵⁷ of 100 USD (t CO₂)⁻¹. The revenue under a carbon price of 0 USD (t CO₂)⁻¹ is*

given in parentheses for a comparison. ”.

The new **Table 1** was revised:

Table 1. Electricity generation, costs and emission reduction by deployment of photovoltaic (PV) and wind-power plants during 2023–2060.

MAC (USD (t CO ₂) ⁻¹)	No. of plants	Area (10 ³ km ²)	Capacity potential (TW)	Capacity factor (%)	Electricity generation (TWh y ⁻¹)	Initial investment (billion USD) ^a	O&M (billion USD y ⁻¹)	UHV & storage cost (billion USD) ^a	Revenue (billion USD y ⁻¹) ^b	LCOE (USD kWh ⁻¹)	Emission reduction (Gt CO ₂ y ⁻¹)
<-20	152	157	1.8	19	2,775	665 (17)	3	223 (6)	101 (20)	0.025	2.3
-20 – -10	256	193	1.7	20	2,845	796 (20)	6	203 (5)	97 (14)	0.029	2.4
-10 – 0	494	219	1.2	23	2,399	870 (22)	9	128 (3)	73 (4)	0.039	2.0
0 – 10	1,027	169	1.7	18	2,502	1,170 (29)	7	330 (8)	68 (-4)	0.048	2.1
10 – 20	1,172	267	2.1	18	3,104	1,617 (40)	12	481 (12)	76 (-14)	0.057	2.5
>20	743	147	1.1	17	1,519	857 (21)	6	345 (9)	34 (-10)	0.064	1.1
Total	3,844	1,151	9.4	19	15,145	5,976 (149)	44	1,709 (43)	448 (10)	0.042	12.5

Notes: a, Annualized costs (billion USD y⁻¹) under a discounting rate² of 5% y⁻¹ are in parentheses.

b, The revenue of transiting fossil fuel to PV or wind-power plants is calculated by subtracting the overall costs of electricity generation from the income by replacing fossil fuel based on the costs of fossil fuel in the electricity generation and the tax of carbon emissions from coal, oil and gas under a carbon price⁵⁷ of 100 USD (t CO₂)⁻¹. The revenue under a carbon price of 0 USD (t CO₂)⁻¹ is given in parentheses for a comparison.

Reference:

Lam, P. L. Pricing of electricity in China. *Energ.* **29**, 287–300 (2004).

Comment A19

- Line 156: Is this because resource constrained? Why doesn't price adjust so that the net present value of cost equals revenue for built capacity in equilibrium (Brown and Reichenberg 2021)?

Response

We agree very much with the Reviewer that cannibalization can depress the market value of renewable energy, because the price of electricity might be reduced by market integration so that the net present value of costs equals the revenue in equilibrium. However, Brown and Reichenberg (2021) also suggested that the declining market value is not necessarily a sign of integration, but a result of policy choices. The prices of electricity in China are controlled by the central government (Lam, 2004), and it is possible to assume that policy choices allow the revenue of deploying renewable energy to maintain over time, thereby providing an economic incentive to achieve the national target of carbon neutrality in China (Xi, 2020).

To clarify our assumption on the cannibalization of revenue, we added the following sentence in lines 200-202: “*The cannibalization due to a decline of prices for renewable energy may*

*depress its future market value*⁴⁹, but it is reasonable to assume that policy in China allows the revenue of PV and wind power to accumulate over time as an incentive to meet the national target of C neutrality⁶.”.

References:

Brown, T. & Reichenberg, L. Decreasing market value of variable renewables can be avoided by policy action. *Energ. Econ.* **100**, 105354 (2021).

Lam, P. L. Pricing of electricity in China. *Energ.* **29**, 287–300 (2004).

Xi, J.P. Xi delivered an important speech during the general debate of the 75th session of the United Nations General Assembly. http://www.gov.cn/xinwen/2020-09/22/content_5546168.htm (2020).

Comment A20

- Lines 161-163: Not conceptually appropriate. Mechanics of calculating MACs for wind and solar are very different from calculating shadow prices on carbon constraints in an economy wide model. These are not comparable metrics despite the same units. Thus, ‘IAMs could overestimate the costs of mitigation’ is too strong of a conclusion!

Response

We realize that MAC for PV and wind power on the supply side should not be compared with the shadow prices on carbon constraints in an economy wide model. We corrected this mistake by stating that the first estimate of MAC curves of deploying PV and wind power in our model is useful to examine the costs of China’s energy transitions in IAMs that prescribed the MACC from the other regions of the world (Duan et al., 2021).

To clarify the appropriate application of MACC, we deleted the comparison between our MAC and the carbon prices in IAMs, and added the following sentence in lines 212-215: “*Negative MACs of PV and wind power at plant level in our study challenge high carbon prices (100–700 USD (t CO₂)⁻¹ except for IPAC²) to abate 6 Gt CO₂ y⁻¹ in IAMs² that prescribed MACC from other regions of the world without considering the spatial heterogeneity in China.*”.

Reference:

Duan, H. et al. Assessing China’s efforts to pursue the 1.5°C warming limit. *Science* **372**, 378–385 (2021).

Comment A21

- Figure 1: This is an excellent visualisation that clarified many of the questions I had about the Table 1! I recommend that this come first in the text. A small suggestion is to use a bar chart (as the authors do elsewhere) to show costs instead of a pie chart.

Response

Thank you very much for the positive comment. We moved Fig. 1 to the beginning of our Results section, and replaced the pie chart with a bar chart.

The new Fig. 1 reads as:

Fig. 1 | Marginal abatement cost (MAC) curves for PV and wind-power plants in China by 2060. (a) MAC in the central case at plant level by coordinating PV and wind power with ultra-high-voltage (UHV) transmission, energy storage, learning and flexible hourly power load under a discounting rate (r_d)² of 5% y^{-1} . Relative to the central case, the sensitivity of MAC is tested to use of fixed hourly power load, exclusion of UHV, storages and learning, deployment of PV, onshore or offshore wind-power plants alone and consideration of a higher discounting rate¹⁵ ($r_d=7\% \text{ y}^{-1}$). The 90% uncertainty (gray shade) and inter-quartile range (blue shade) in MACs are derived from Monte Carlo simulations by randomly varying model parameters. **(b)** Composition of the levelized cost of electricity (LCOE) for PV and

wind-power plants contributing to installed capacities from 0 to 8 TW using fixed hourly power load (red frame) or optimizing the flexible hourly power load (blue frame), when other configurations are identical to the central case. (c) Location of the projected PV and wind-power plants and the optimal time to build each power plant based on the range of installed capacities in the central case. The color of symbol indicates the year of building each power plant, while the color of background indicates the abated CO₂ emissions in each region.

Comment A22

- Lines 211-213: Why not marginal curtailment?

Response

Marginal curtailment improves the efficiency of using renewable energy, but it leads to loss of renewable energy (Jacobsen et al., 2012). There are many strategies to avoid curtailment (Li et al., 2015). In our revised model, we have considered the flexible power load with UHV transmission and storage to improve the match between power demand and generation (please see our response to **Comment A2**), which reduces the need of marginal curtailment.

To quantify the impact of flexible power load with transmission and storage on marginal curtailment, we performed new experiments where we avoided overinvestments in the installed capacities by applying the constraints of power demand to power generation (Jacobsen et al., 2012). We found that the impact of marginal curtailment is small in our central case by optimizing flexible power load with transmission and storage, but the impact is large in a sensitivity test using fixed power load without transmission or energy storage (**Fig. S15**). These new results supported that optimizing flexible power load with transmission and storage has the effect to reduce the need of marginal curtailment and increase the maximal potential of emission abatements.

To clarify the effect of flexible power load with transmission and storage in reducing the need of marginal curtailment, we added the following sentences in lines 316-321: “*The strategy of combining optimal energy storage with flexible power load improves the match between power demand and generation, thereby reducing marginal curtailment of renewable energy*⁵¹. We assessed the impact of marginal curtailment by avoiding overinvestments in installed capacities with constraints of power demand⁵². The impact of marginal curtailment on MACC is large when using fixed power load without storage, but becomes small by optimizing the flexible power load with storage (**Fig. S15**).”.

To show the impact of marginal curtailment on MACC, a new **Fig. S15** was added to the **Supporting Information**:

Fig. S15. The impact of marginal curtailment on the marginal abatement cost (MAC) curve for PV and wind power in 2060. To assess the impact of marginal curtailment, we compare the MAC curves between two experiments without (blue line) or with (red line) constraints by power demand to avoid overinvestment in the installed capacities (Jacobsen et al., 2012). (a) The impact of marginal curtailment on MAC curves in the central case with energy storage, ultra-high-voltage (UHV) transmission, the projected rate of electrification and the optimisation of flexible power load. A sensitivity case without electrification of non-power sectors (dashed line) shows the impact of total electricity demand on the MAC curves. (b–d) The impact of marginal curtailment on MAC curves in the sensitivity cases (b) without storage, (c) using the fixed hourly power load without storage and (d) using the fixed hourly power load without storage or UHV.

References:

Li, C. et al. Comprehensive review of renewable energy curtailment and avoidance: a specific example in China. *Renew. Sustain. Energ. Rev.* **41**, 1067–1079 (2015).

Jacobsen, H. K. & Schröder, S. T. Curtailment of renewable generation: Economic optimality and incentives. *Energ. Polic.* **49**, 663–675 (2012).

Comment A23

- Line 234: ‘Electricity generation’ instead of ‘energy’? Important to clarify between electricity and energy throughout the paper.

Response

We corrected “energy” as “electricity generation” and checked it throughout our paper.

Comment A24

- Line 236: ‘projected’ instead of ‘predicted’.

Response

We corrected “*predicted*” as “*projected*” and checked it throughout our paper.

Comment A25

- Line 247-248: How are regional and global spillovers addressed?

Response

We addressed the impacts of regional and global spillovers by performing additional experiments and gathering new data in our response to **Comment A17**.

Comment A26

- Line 253: ‘total energy supply’ Is this electricity or energy? If the latter, is this primary energy or final energy?

Response

Sorry for this confusion. We corrected “*total energy supply*” as “*total electricity demand*” in lines 377.

Comment A27

- Figure 3: Another good multipanel visualization. For panel d, I recommend separating coal with and without CCS for clarity. In this panel, do you mean LCOE instead of ‘electricity price’? ‘Electricity price’ seems to imply that this was an exogenous assumption.

Response

Thank you very much for the positive comment.

Specifying the application of CCS for coal or other energy (e.g. biomass) is beyond the scope of our model, so we estimated the total demand of CCS based on the projection of PV and wind power (see our response to **Comment A16**). To avoid an unjustified assumption on the exogenous prices of electricity, we replaced the scenarios using different electricity prices with the two scenarios of “*no policy*” and “*limiting 1.5 C*” using power generation by other renewable energy from a multi-model study (Duan et al., 2021).

The new **Fig. 3** is revised:

Fig. 3 | Optimal path to achieve C neutrality by building PV and wind-power plants in China by 2060. (a) Dependence of LCOE of the projected PV and wind-power plants built by 2060 on the share of electricity generation by PV and wind-power plants built in 2020s in total electricity demand in 2060. The optimal path minimizes LCOE of the projected power plants in 2060 by randomly varying the time each plant is built by decade (the shaded area) under a discounting rate (r_d) of 5% y^{-1} in the central case² or 7% y^{-1} in the sensitivity case¹⁵ (red). The insert shows the share of electricity generation by PV and onshore wind and offshore wind-power plants built by decade in total electricity generation in 2060 under the optimal path. (b, c) The (b) time and (c) LCOE of building each PV or wind-power plant in the optimal path. (d) Composition of power generation by decade. Power generation by oil, gas, bioenergy, nuclear and hydropower is derived as the average of three models (SWITCH, IPAC and GCAM_TU) under the “no-policy” and “1.5°C-limiting” scenarios from a multi-model study², while power generation by PV and wind is derived from MACC in this

study under the two scenarios based on the prices of carbon in the models², respectively. Assuming that coal meets the remaining power demand, the demand of CCS (red dot) is estimated to meet the national carbon neutrality. The estimate of PV and wind power in a previous study² and the resultant demand of CCS are shown by the hatched bar and the blue dot, respectively. (e, f) Spatial distributions of (e) PV and (f) wind-power plants constructed by decade under the optimal path. The size and color of the symbol denote the installed capacity and the time to build each power plant. The background shows the global horizontal irradiance (GHI) and wind-power density (WPD).

Reference:

Duan, H. *et al.* Assessing China's efforts to pursue the 1.5° C warming limit. **Science** **372**, 378–385 (2021).

Comment A28

- Lines 271-272: Capacity or generation assumed to be constant?

Response

We realize that it is unreasonable to assume that the power capacity and electricity generation of other renewable energy beyond PV and wind are constant under different scenarios. In the revised paper, we used the projection of power generation by oil, gas, bioenergy, nuclear and hydropower during 2030–2060 under the “1.5°C-limiting” scenario to match the national target of carbon neutrality, which is compared to the projection under the “no-policy” scenario (Duan *et al.*, 2021). It allows the power generation by other renewable energy to be variable over the period of 2030–2060 under different scenarios.

To clarify the projection of other renewable energy under different scenarios, we moved the methodological details from the caption of Fig. 2 in our previous paper to the **Method summary** section in our revised paper in lines 166-184: “*We simulated hourly power load for PV and wind by subtracting the power generation by oil, gas, bioenergy, nuclear and hydropower under the 1.5°C-limiting scenario² from the total power demand under different electrification rates from 0 to 100% by 2060⁴¹. For space heating and cooling in houses and electric cars, we estimated hourly power load based on the projected increase in power demand⁴¹ and power-temperature relationships under projected temperature in the SSP1-2.6 or SSP2-4.5 scenario⁴⁴. For other sectors (Table S1), we scaled historical hourly power load from electrical grids in 2018⁴⁵ by the projected increase in electricity⁴¹ during 2023–2060 and optimized the hourly power load⁴⁶ to match hourly power generation by PV and wind. In a sensitivity test, we fixed the hourly power load without this optimisation⁴⁶. Given uncertainties in the geological constraints on CCS⁴⁷, it is interesting to know the demand of CCS needed to achieve China's C neutrality, which depends on terrestrial C sink (1.87 Gt CO₂ y⁻¹)⁴⁸, rate of electrification (58%)⁴¹, total electricity demand⁴¹ and future power generation by PV, wind and other low-carbon energy². We adopted total power generation by oil, gas, bioenergy, nuclear and hydropower as the average of three IAMs (SWITCH, IPAC and GCAM_TU) under the “no-policy” and “1.5°C-limiting” scenarios from a multi-model study². The power generation by PV and wind was estimated according to MACC from this study under the “no-policy” and “1.5°C-limiting” scenarios based on the average prices of carbon in the IAMs². Assuming that coal provides the remaining electricity, we estimated total demand of CCS in electricity systems to meet the national target of C neutrality⁶.”.*

Reference:

He, J. *et al.* Towards carbon neutrality: A study on China's long-term low-carbon transition pathways and strategies. *Environ. Sci. Ecotechnol.* **9**, 100134 (2022).

Comment A29

- Lines 473-475: The assumed lifetime of storage of five years is a severe underestimate! In the literature, lifetimes of 15-25 years are common for battery storage and longer for other options.

Response

We realize that the assumed lifetime of five years in our previous paper is too short for battery storage. In our revised paper, we considered two energy storage systems (please refer to our response to **Comment A2**), while we adopted a lifetime of 50 years for mechanical (pumped hydro) storage (Chen et al., 2021) and 15 years for chemical (batteries) storage (Danish Energy Agency, 2017).

To clarify the lifetime of energy storage systems, we added the following sentence in lines 809-814: “*In addition, we compared the LCOE for each power plant when using mechanical storage (pumped hydro storage) with a lifetime of 50 years and a round-trip efficiency of 70% or using chemical storage (battery storage) with a shorter lifetime of 15 years but a higher round-trip efficiency of 85%. The properties of these two energy storage systems are listed in Table S9. We sought the optimal system of energy storage for each power plant to minimize LCOE in the central case.*”.

References:

Chen, X. et al. Pathway toward carbon-neutral electrical systems in China by mid-century with negative CO₂ abatement costs informed by high-resolution modeling. *Joule* **5**, 2715–2741 (2021).

Danish Energy Agency. Technology Data for Renewable Fuels. URL: https://ens.dk/sites/ens.dk/files/Analyser/technology_data_for_renewable_fuels.pdf (2017).

Comment A30

- Line 482: Where are the costs associated with building and operating other technologies in this objective function formulation?

Response

We estimated the costs of PV and wind power including initial investment of building the projected PV and wind-power plants, operation and maintenance, power generation, electricity transmission and energy storage. As our paper focused on estimate the costs and potential of PV and wind power, we did not account for the costs of deploying other renewable energy in our paper (please refer to our response to **Comment A4**).

To clarify the caveat of lacking other renewable energy beyond PV and wind power, we added the following sentence in lines 871-874: “*We focused on estimating the costs of deploying PV and wind power in China’s energy system, but the costs for deploying other renewable technologies beyond PV and wind are not considered in our model.*”.

Comment B1

The manuscript presents a comprehensive model of massive deployment of wind and solar PV power in China to facilitate its decarbonisation by 2060. The model is remarkable in its detail and complexity including high-granularity spatial disaggregation, taking into account the terrain, administrative boundaries of counties, climatic conditions, demand projections, learning rates, storage and transmission needs and opportunities. Its findings include a range of marginal abatement cost of carbon (MACC) depending on the deployed solar and wind capacity, investment needs, and the electricity trade flows across different regions. Despite its sophistication, the model is well explained and, as far as I can judge, rigorous and reasonable.

An outstanding finding of the paper which would resonate beyond its immediate objective is demonstrating and quantifying the cost (LCOE) dependence on the quantity of wind and solar power deployed. It is particularly remarkable that LCOE and MACC significantly increase at higher deployment, even when learning is taken into account. This challenges a widespread view (e.g. [2]) that the costs of solar and wind will keep declining while their deployment will grow exponentially and that energy transition as a result will have net negative cost. Increasing marginal costs of deployment may also explain the S-shaped curves of wind and solar growth in many European countries [5]. It would be good if this insight could be made more prominent thus signalling the importance of the study beyond China and its contribution to the more general debate on energy transitions.

Another interesting finding is that about 2.6 TW of wind and solar can be deployed profitably (and require some \$1trln stimulus investment), while about 6 TW would need some subsidy associated with the price of carbon in the range of 50-100 USD/ton (if I read Fig. 1 and Table 1 right).

I believe that the main area of improvement for this manuscript would be to better highlight its novelty and implications as well as its limitations and how these should be addressed in future research. Since the manuscript attempts several different tasks I will explain this comment in relation to each of them.

Response

Thank you very much for the positive comments. We are pleased to see that our model is rigorous and our finding is interesting. We have revised our paper carefully to consider the Reviewer's comments by systematically comparing our findings with previous studies (see our response to **Comments B2–B7**), clarifying the novelty of our study by revising the **Abstract** and the final paragraph (see our response to **Comments B8, B13**) and clarifying the limitations of our modeling approach by performing additional experiments and gathering new data (see our response to **Comments B9–B11**). We hope that these revisions allow our paper to match the wider readership of the Journal.

Comment B2

Novelty and contribution

1. ****Wind and solar power deployment necessary and feasible for China's net zero by 2060 (CNZ).**** The manuscript starts with citing 20 TW of renewable energy necessary for CNZ (25-26). Unfortunately I could not find this number in the cited ref 1. Instead it envisions (Fig 4-E) non-fossil electricity generation of 20 TWh/year consistent across models and broken down into solar, wind, hydro, nuclear etc. Both the units and the concept are different between

ref. 1 and what is cited in the manuscript. In addition, the IAM models used for ref. 1 seem to envision very different levels of deployment of renewables. The manuscript could sharpen its contribution by indicating which of these different models produce more likely predictions.

* There are other studies quoted in the manuscript, but since they all use different concepts (capacity vs. generation, time horizons) and probably different targets (For example in 121-123 the units of generation (for PV and wind) and capacity (for BNHH) are mixed which is confusing), it should be better how the manuscript reduces the existing uncertainty in this matter. In other words, how does 8.6 TW of capacity of solar and wind by 2060 that the study sees as necessary and feasible relate to other estimates?

* It may also be that the message of the manuscript is more sophisticated than just giving a single number, but it only makes it more pressing to explain how the range of estimates (under different MACC/carbon prices) relate to previous estimates. Presenting this in a figure or table may be useful.

Response

In our previous paper, we derived the “20 TW of renewable energy” from power generation (20 TWh/year) in Fig. 4E of Ref. 1 (Duan et al., 2021) by applying a capacity factor of 0.2 in our model. We realize that this number is very confusing. In our revised paper, we referred to the power capacity of renewable energy installed by 2060 in **Exhibit 67** of Goldman Sachs Research (2021), we revised the first sentence of **Abstract** in lines 24-26: “*The pledge toward carbon neutrality in China implies a power demand^{1,2} of 7,000–20,000 TWh y⁻¹ with capacities exceeding 8,000 GW¹ from renewable energy.*”.

Exhibit 67: ...with the potential for >4,000 GW of solar and c.3,000 GW of wind power generation capacity additions to 2060
China net zero power generation capacity bridge (2019-60E, GW)

Source: Goldman Sachs Global Investment Research.

We agree very much with the Reviewer that we need to compare our estimates of power capacity, electricity generation, levelized cost of electricity (LCOE) and marginal abatement cost (MAC) in our study with the same concepts in the literature. We provided the new **Fig.**

S1 as a scatter plot of LCOE or MAC against the power capacity or electricity generation. We clarified the latest estimates for China as the “more likely predictions” using the size of symbol. All data in the literature are listed in a new **Supplementary Spreadsheet S1**, but we only used the data with explicit information for LCOE, MAC, power capacity or electricity generation for a comparison in the new **Fig. S1**.

Most of previous studies estimated LCOE for utility-scale PV and wind power plants in China, but the estimate of MAC is lacking in the literature. Our estimate of LCOE (0.04 USD kWh⁻¹ for all plants) is closer to the bottom quartile (0.03 USD kWh⁻¹) of previous estimates than the average (0.07 USD kWh⁻¹), while previous estimates of MAC are higher or close to the values on our MAC curve. The estimated power capacity (8.6 TW) of PV and wind by 2060 in our paper is higher than previous estimates, mainly due to the large PV and wind-power plants in our optimisation model. We have examined the feasibility of building these large power plants (see our response to **Comment B10**).

We have also compared the installed power capacity of PV and wind between our optimisation model and the regional renewable energy development plans (Global Energy Interconnection Development and Cooperation Organization, 2020). We found that power capacity of PV and wind is close to our estimate by 2060, but could be seriously underestimated by the regional renewable energy development plans (Global Energy Interconnection Development and Cooperation Organization, 2020), which does not account for the effect of learning (see our response to **Comment C9**).

These new results suggested that the potential of deploying PV and wind power in China could be underestimated by previous studies without optimizing the timing and placements of PV and wind-power plants or without considering the benefits of learning.

To compare the estimated power capacity, electricity generation, LCOE and MAC between our study and previous studies, we added the following sentences in lines 228-245: “*We compared LCOE, MAC, electricity generation and power capacity of PV and wind between our study and previous studies (Fig. S1). Many studies estimated LCOE for PV and wind, but the MAC is lacking in the literature. Our estimate of LCOE (0.04 USD kWh⁻¹ for all plants) is closer to the bottom quartile (0.03 USD kWh⁻¹) of previous estimates than to the average (0.07 USD kWh⁻¹), while previous estimates of MAC are higher or close to the values on our MACC. A recent study¹⁴ estimated MAC at 27 USD (t CO₂)⁻¹ for PV and wind power under an 80% renewable penetration in China, when they assumed that 4 TW of PV and wind capacities would be installed by 2050 without considering the effect of learning and flexible power load; MAC for 4 TW is -6 (90% uncertainty of -21 to 18) USD (t CO₂)⁻¹ in our estimate. Lu et al.¹⁵, without providing the MAC, suggested an electricity price of PV at 0.03 USD kWh⁻¹ for 2060, compared to a LCOE of 0.024–0.057 USD kWh⁻¹ in our estimate, likely because Lu et al.¹⁵ adopted a high learning rate (42.4% by Lu et al.¹⁵ against 7–20% in our Table S2) and Lu et al.¹⁵ did not include the costs of transmission (0.002–0.006 USD kWh⁻¹ in our estimate). These results suggested that, by optimizing the placement and timing of PV and wind-power plants with flexible power load, transmission and energy storage in China, MAC of PV and wind power could be lower than previously thought, with higher potential of emission abatements.*”.

To compare the estimated power capacity, electricity generation, LCOE and MAC between our study and previous studies, a new **Fig. S1** was added to the **Supporting Information**:

Fig. S1. Comparison of levelized cost of electricity (LCOE) and marginal abatement cost (MAC) for solar photovoltaic (PV) and wind-power plants in China between this study and previous studies. (a, b) A scatter plot of LCOE against electricity generation (a) and installed capacity (b) for PV (circle), onshore wind (square) and offshore wind (pentagram) power in China. Estimates from this study are shown by the crosses. (c, d) A scatter plot of MAC against electricity generation (c) and installed capacity (d) for PV (circle), onshore wind (square) and offshore wind (pentagram) power. The MAC curves are estimated in this study. The size of the symbol indicates the year of the estimate (i.e. the smallest one for the estimate of 2015 and the largest one for the estimate in 2060). The error bars show the range of estimate for different provinces in China. The data in the literature are listed in **Supplementary Spreadsheet S1**, while only the data with explicit information for LCOE, MAC, electricity generation and power capacity are included in this comparison.

References:

Duan, H. *et al.* Assessing China's efforts to pursue the 1.5°C warming limit. *Science* **372**, 378–385 (2021).

Goldman Sachs Research. Carbonomics: China net zero—The clean tech revolution. URL: <https://wwwqa.goldmansachs.com/insights/pages/gs-research/carbonomics-china-netzero/report.pdf> (2021).

Global Energy Interconnection Development and Cooperation Organization. Research on China's energy transition and 14th Five-year Power planning (2020).

Comment B3

2. ****MACC of wind and solar power deployment in China.**** The manuscript does a fantastic job by demonstrating how MACC depends on many assumptions and how it varies with increasing capacity of deployment and with various deployment strategies. It also provides a table of previous MACC estimates as a supplementary spreadsheet. However, these estimates are from very diverse studies, some of which are 10 years old or older. It would be useful to highlight which of the MACC estimates are state-of-the-art and how does the paper compares with those most current and rigorous estimates.

Response

Thank you for the positive comment. Following this comment, we provided the detailed information for the estimated power capacity, electricity generation, LCOE and MAC in the literature in the new **Supplementary Spreadsheet S1**. We clarified the latest estimates for China as the “more likely predictions” using the size of symbol in the new **Fig. S1** (see our response to **Comment B2**). In the literature, there are many studies estimating LCOE for utility-scale PV and wind-power plants in China, but there is a lack of MAC estimates in China. We compared our study with the “most current and rigorous” estimates (Chen et al., 2021; Lu et al., 2021).

To compare our study with the most current and rigorous estimates in the literature, we added the following sentences in lines 233-245: “*A recent study¹⁴ estimated MAC at 27 USD ($t\ CO_2$)⁻¹ for PV and wind power under an 80% renewable penetration in China, when they assumed that 4 TW of PV and wind capacities would be installed by 2050 without considering the effect of learning and flexible power load; MAC for 4 TW is -6 (90% uncertainty of -21 to 18) USD ($t\ CO_2$)⁻¹ in our estimate. Lu et al.¹⁵, without providing the MAC, suggested an electricity price of PV at 0.03 USD kWh⁻¹ for 2060, compared to a LCOE of 0.024–0.057 USD kWh⁻¹ in our estimate, likely because Lu et al.¹⁵ adopted a high learning rate (42.4% by Lu et al.¹⁵ against 7–20% in our **Table S2**) and Lu et al.¹⁵ did not include the costs of transmission (0.002–0.006 USD kWh⁻¹ in our estimate). These results suggested that, by optimizing the placement and timing of PV and wind-power plants with flexible power load, transmission and energy storage in China, MAC of PV and wind power could be lower than previously thought, with higher potential of emission abatements.*”.

In addition, to show the advantage of our spatially explicit method, we have compared our estimate of PV and wind power to a previous estimate prescribing MACC from the other regions of the world for China (Duan et al., 2021). The new **Fig. 3d** shows that the power generation by PV and wind in China could have been underestimated by up to 60% in the IAMs (see our response to **Comment A17**).

Reference:

Chen, X. *et al.* Pathway toward carbon-neutral electrical systems in China by mid-century with negative CO₂ abatement costs informed by high-resolution modeling. *Joule* **5**, 2715–2741 (2021).

Duan, H. *et al.* Assessing China’s efforts to pursue the 1.5° C warming limit. *Science* **372**, 378–385 (2021).

Lu, X. *et al.* Combined solar power and storage as cost-competitive and grid-compatible supply for China’s future carbon-neutral electricity system. *Proc. Natl. Acad. Sci. U.S.A.* **118**, e2103471118 (2021).

Comment B4

* Incidentally, the comparison (161-163) of the cost of mitigation in IAMs to MACC (from ref.1) should be better justified, because the former includes hard-to-decarbonise sectors and not only electricity and therefore it is logical that carbon prices in IAMs are higher than MACC for renewables. The Supplementary materials from Ref. 1 contain LCOEs which can be more directly compared with the findings of the manuscript.

Response

We agree very much with the Reviewer that our estimates of MAC for PV and wind on the supply side cannot be compared with the costs of decarbonization derived from the prices of carbon on the demand side in the IAMs (Duan et al., 2021). We deleted this comparison between MAC and the prices of carbon. To clarify the limitations in the IAMs prescribing MACC from the other regions of the world for China, we added the following sentence in lines 101-104: “*Spatial heterogeneity of PV and wind resources³³ and temporal dynamics of learning¹⁷ reduce the confidence in the projection of China’s energy transitions by integrated assessment models (IAMs) prescribing MACC from other regions².*”.

We included data from Ref. 1 (Duan et al., 2021) in the comparison of LCOE between our study and previous studies in the new **Fig. S1** (see our response to **Comment B2**).

Reference:

Duan, H. *et al.* Assessing China’s efforts to pursue the 1.5° C warming limit. *Science* **372**, 378–385 (2021).

Comment B5

3. **Optimal location and timing of deployment of wind and solar power to reach CNZ.**
The manuscript mentions other studies investigating this matter although apparently less sophisticated. How are the results of these other studies compare to those of the manuscript?

Response

There are two previous studies (Chen et al., 2021; Lu et al., 2021) investigating the spatial and temporal distributions of wind and PV power to reach carbon neutrality in China, but neither of them optimized the timing and placement of each PV and wind-power plant as we did. For example, Lu et al. (2021) stated “*we assumed in our analysis that the annual installed capacity would increase linearly from 250 GW to 1200 GW from 2021 to 2040, decreases linearly from 1200 GW to 250 GW from 2041 to 2060.*”, while Chen et al. (2021) did not consider the effect of learning in the optimisation of the time needed to build each power plant. Therefore, it is hard to perform a direct comparison between our study and previous studies. By referring to Fig. 3A of Chen et al. (2021) and Fig. 1B of Lu et al. (2021), we found that their power generation by PV and wind was relatively homogeneously distributed in the North and Northwest of China, which became more concentrated in space by specifying the location of each power plant in our model. In addition, by referring to Fig. 2B of Chen et al. (2021), we found that 80% of the projected PV and wind power capacities were installed by 2040 in their estimates, higher than 42% in our estimate, likely because they did not consider the effect of learning in their model (Chen et al., 2021).

To clarify the similarity and differences in the timing and placement of PV and wind-power plants between our study and previous studies, we added the following paragraph in lines 388-403: “*The total installed capacities of PV and wind power reach 7.3 TW by 2050 under a discounting rate of 5% y⁻¹, but the timing of abatement depends on the rate of discounting. The fraction of total PV and wind capacities installed by 2050 decreases from 80 to 75% if the discounting rate increases from 3 to 7% y⁻¹ (Fig. S20). We adopt a lifetime of 25 years for PV and wind-power plants, but a longer lifetime of power plants (e.g. using crystalline silicon PV*

modules⁵⁴) has the effect to reduce costs, where an increase in the lifetime of plants from 15 to 35 years reduces the average MAC for PV and wind-power plants from 10 to -7 USD (t CO₂)⁻¹ (Fig. S21). In addition, we examined the sensitivity of MACC to different capital costs. The average MAC for PV and wind-power plants increases from -2 USD (t CO₂)⁻¹ using capital costs by component (Table S8) to 1 USD (t CO₂)⁻¹ if we adopt high capital costs³⁹, or decreases to -6 USD (t CO₂)⁻¹ if we adopt low capital costs⁵⁵ (Fig. S22). The electricity generation by PV and wind is almost homogeneously distributed in the North and Northwest of China in previous studies^{14,15}, which becomes concentrated in space by optimizing the location and size of each power plant in our study. We projected that 42% of PV and wind capacities will be installed by 2040 due to a reduction of capital costs by learning¹⁷, lower than the fraction of 80% without considering the effect of learning¹⁴.”.

References:

Chen, X. *et al.* Pathway toward carbon-neutral electrical systems in China by mid-century with negative CO₂ abatement costs informed by high-resolution modeling. *Joule* **5**, 2715–2741 (2021).

Lu, X. *et al.* Combined solar power and storage as cost-competitive and grid-compatible supply for China’s future carbon-neutral electricity system. *Proc. Natl. Acad. Sci. U.S.A.* **118**, e2103471118 (2021).

Comment B6

4. ****Investment requirements for wind and solar power in China.**** I really liked the differentiation between investments that can be provided as stimulus (i.e. for profitable plants) and the need for subsidies. But the same question: has this been estimated before? How do the results compare?

Response

Yan et al. (2019) estimated that 22% of current PV power plants could achieve the LCOE below the grid-supplied electricity price to generate the profits, but their analysis was limited to 344 prefecture-level Chinese cities in 2018 without covering the country for the future period. Liu et al. (2020) estimated that it was difficult to achieve the wind grid parity using a system dynamics model, but the model was limited to the period of 2017–2025 without estimating the demand of total subsidies for wind power by 2060 in China. Chen et al. (2021) estimated that the total overall costs of deploying PV and wind power were \$200 billion y⁻¹ to achieve power capacities of 2 TW in their Fig. 2A, but it was unclear how many of these power plants were profitable and how many needed the subsidies. Therefore, to date, our study provides the first estimate of the stimulus of profitable power plants and the demand of subsidies for the remaining power plants in China to achieve carbon neutrality by 2060.

To clarify the similarity and differences in the estimate of investment costs between our study and previous studies, we added the following sentences in lines 542-547: “*We identified a stimulus of 58 billion USD y⁻¹ as initial investments for profitable power plants with a negative MAC and subsidies of 91 billion USD y⁻¹ for power plants with a positive MAC. To achieve installed capacities of 2 TW, overall costs (29 billion USD y⁻¹) in our estimate are lower than 100 billion USD y⁻¹ from a previous estimate¹⁴. One fifth of PV plants in 344 prefecture-level Chinese cities are profitable⁶⁴ in 2018, but the commissioned wind plants may require subsidies⁶⁵ by 2025.*”.

References:

Chen, X. *et al.* Pathway toward carbon-neutral electrical systems in China by mid-century with negative CO₂ abatement costs informed by high-resolution modeling. *Joule* **5**, 2715–

2741 (2021).

Yan, J., Yang, Y., Elia Campana, P. & He, J. City-level analysis of subsidy-free solar photovoltaic electricity price, profits and grid parity in China. *Nat. Energ.* **4**, 709–717 (2019).

Liu, Y., Zheng, R., Yi, L. & Yuan, J. A system dynamics modeling on wind grid parity in China. *J. Clean. Prod.* **247**, 119170 (2020).

Comment B7

5. ****Impacts of wind and solar power on economic inequality between regions in China.**** Once again, it's an elegant method and an interesting finding, but is there other literature on this and how do the results compare? Or is it done for the first time?

Response

We noticed two recent studies performing a quantitative analysis of the impact of deploying PV power on poverty alleviation using the entropy weight method at an individual level (Wang et al., 2020) or using the statistical model based on a panel dataset in 211 pilot counties (Zhang et al., 2020). They both suggested that deploying PV increased per-capita disposable income in poor rural area in China, but neither of them extended their studies to the nation due to the lack of a full cost analysis for all projected PV and wind power plants to be built by 2060 in China. We did not see any studies estimating the impact of deploying wind power on poverty alleviation in the literature. Therefore, to date, our study provided the first estimate of the impact of deploying PV and wind power on income inequality in China (Fig. S25).

To clarify the novelty of our study in providing the first estimate of the impact of deploying PV and wind power on inequality in China, we added the following sentences in lines 472-481: *“PV provided income for the poorest rural residents at an individual level⁶⁰ and for 211 pilot counties⁵⁶, but a national cost-benefit analysis for future PV and wind-power plants is lacking. We showed that 28% of residents have income <5000 USD y⁻¹ in Northwest China (Fig. S25), where the people is vulnerable to climate change due to currently high temperatures and frequent drought⁶¹; the revenue of PV and wind power in this region alone, i.e. 510 billion USD y⁻¹ or equivalent to 28% of local gross domestic product (GDP)⁶², will create an opportunity to raise income of the poorest residents and to alleviate the poverty that is likely intensified by climate change⁵.”*

Fig. S25. Impact of deploying the projected photovoltaic (PV) and wind-power plants on income distributions in China. (a, b) Population distributions in different regions of China in the scenario without the projected deployment of PV and wind-power plants during 2023–2060 (a) and in the scenarios deploying PV and wind power under a carbon price of 100 USD $(t CO_2)^{-1}$ (b). The insert shows the distribution of population with the income below the USA’s poverty line of 5000 USD y^{-1} (Jolliffe and Prydz, 2016). (c) Difference in the population density by increasing the carbon price from a scenario without deployment of PV and wind power during 2023–2060 (a) to the scenario deploying PV and wind power under a carbon price of 100 USD $(t CO_2)^{-1}$.

References:

Zhang, H. *et al.* Solar photovoltaic interventions have reduced rural poverty in China. *Nat. Commun.* **11**(1), 1–10 (2020).

Wang, Z., Huang, F., Liu, J., Shuai, J. & Shuai, C. Does solar PV bring a sustainable future to the poor?—an empirical study of anti-poverty policy effects on environmental sustainability in rural China. *Energ. Polic.* **145**, 111723 (2020).

Comment B8

6. ****Beyond wind/solar and beyond China.**** What are the implications of the novel method and new results for other technologies and deployment of wind and solar in other countries? China is a very large and diverse country and I believe many lessons would be relevant to other countries and regions. I think it would be important to know for wider Nature readership. I see several results potentially important for wider readership

1. The already mentioned documentation quantification of rising MACC with large levels of

deployment might be very useful for modellers and for policy makers.

2. The fact that optimised deployment of wind and solar power requires mega plants located in remote regions and backed by interregional transmission and storage

3. The fact that large-scale deployment of wind and solar power can be done profitably to some extent (but requires massive investment and hence stimulus) and requires carbon tax or other subsidies to penetrate to substantial levels.

Response

Thank you very much for these enlightening comments.

We broadened the policy implications of our study by referring to important studies in the literature (Kramer et al., 2009; Ives et al., 2021; Cherp et al., 2021; Keppo et al., 2021; Sasse et al., 2020; Wilson et al., 2013; Victoria et al., 2020; Johansson et al., 2021). To match the wider readership of the Journal, we added a new final paragraph in lines 626-649:

In addition to improving the model of PV and wind using a spatially explicit method, our study provides crucial policy-relevant information on deploying renewable energy beyond PV and wind and for regions beyond China. First, rising MAC with higher levels of deployment even if considering the effect of learning poses a challenge to the widespread view that the costs of renewable energy such as PV and wind keep declining as their deployment grows exponentially to produce net negative costs of energy transition⁸⁰. We showed physical limits to the extent of expanding the size of PV and wind-power plants⁸¹ and supported the need of policy interventions (e.g. levying a carbon tax¹ and supporting grid integration³⁷) to overcome barriers in a large-scale deployment of renewable energy for emerging economies pursuing an alleviation of poverty. Second, our projection of large PV and wind-power plants to achieve a negative MAC provide opportunities to reduce costs of decarbonization in many undeveloped regions in China and also for the Middle East and African countries with vast areas of desert, but they require high initial investments as a stimulus⁷⁸. Working in this direction requires development of high-resolution models³³ combined with a real-time analysis of renewable resource availability⁸² and electrification of non-power sectors¹⁹. Third, our optimisation model brings forward the efforts of emission abatement by installing 80% of the projected PV and wind capacities by 2040, which could accelerate the decline of capital costs and upgrade of electricity systems¹⁷. This path challenges the traditional path under overshoot scenarios where energy transitions will be delayed to the second half of this century⁸³. These results from the first projection of MACC for PV and wind power at plant level with intertemporal dynamics suggest that a better understanding of physical limits of renewable energy and a comprehensive modeling of socio-economic systems are urgently needed to determine the optimal path of global and regional energy transitions and to achieve sustainable development of human society and Earth system.

References:

Keppo, I. *et al.* Exploring the possibility space: taking stock of the diverse capabilities and gaps in integrated assessment models. *Environ. Res. Lett.* **16**, 053006 (2021).

Johansson. D. J. A. The question of overshoot. *Nat. Clim. Chang.* **11**, 1021–1022 (2021).

Sasse, J. P. & Trutnevyte, E. Regional impacts of electricity system transition in Central Europe until 2035. *Nat. Commun.* **11**, 1–14 (2020).

Victoria, M., Zhu, K., Brown, T., Andresen, G. B. & Greiner, M. Early decarbonisation of the European energy system pays off. *Nat. Commun.* **11**, 1–9 (2020).

Ives, M. *et al.* A new perspective on decarbonising the global energy system. URL: https://www.ucl.ac.uk/climate-action-unit/sites/climate_action_unit/files/energy_transition_spm_ch.pdf (2021).

Kramer, G. J. & Haigh, M. No quick switch to low-carbon energy. *Nature* **462**, 568–569 (2009).

Cherp, A., Vinichenko, V., Tosun, J., Gordon, J. & Jewell, J. National growth dynamics of wind and solar power compared to the growth required for global climate targets. *Nat. Energ.* **6**, 742–754 (2021).

Wilson, C. *et al.* Future capacity growth of energy technologies: are scenarios consistent with historical evidence? *Clim. Chang.* **118**, 381–395 (2013).

Comment B9

Limitations and connections with other literature

Though the manuscript uses a sophisticated and rigorous model it should still highlight its limitations based on how it frames and scopes the problem.

1. ****MACC calculations.**** Unless I am mistaken, these depend upon assumptions about the costs and composition (gas vs. coal) of fossil-fuel based power production. How was this modelled and what are the sensitivities?

Response

We agree very much with the Reviewer for examining the sensitivity of MACC to assumptions about the costs and composition (gas vs. coal) of fossil-fuel based power production in the future period.

In our revised paper, we considered the income of replacing fossil fuel based on the prices of coal (0.043 ± 0.007 \$ kWh⁻¹) (Statista, 2021), oil (0.141 ± 0.026 \$ kWh⁻¹) (Yte1, 2022; China National Petroleum Corp, 2012) and gas (0.058 ± 0.008 \$ kWh⁻¹) (National Development and Reform Commission, 2019), derived as the average and standard deviation of prices from 2010 to 2020 in China (Fig. S11a). We estimated the MAC by assuming that PV and wind power will be used to replace oil, gas and coal in the order of their fuel prices to generate the largest profits.

Power production by fossil fuel in China is currently dominated by coal with a contribution of 96% in 2020 (IEA, 2021), but we realize that the fuel composition is uncertain in the future. We projected the consumptions of coal, oil and gas for 2020–2060 by scaling up the consumptions in 2020 with the projected growth of power demand (IEA, 2021) in the central case. To assess the sensitivity of MACC to changes in fuel composition, we performed two sensitivity tests in the new Fig. S11b-d. First, the share of oil in electricity generation will increase from the current level (0.3%) in 2020 to 50% in 2060 by replacing coal, while the share of gas is identical to the central case. Second, the share of gas in electricity generation will increase from the current level (4.1%) in 2020 to 50% in 2060 by replacing coal, while the share of oil is identical to the central case.

Due to a higher price of oil or gas than coal, the MAC to abate 8 Gt CO₂ y⁻¹ by PV or wind power will decrease from 18.2 USD (t CO₂)⁻¹ in the central case to -17.3 USD (t CO₂)⁻¹ if coal is replaced by oil, or to 0.3 USD (t CO₂)⁻¹ if coal is replaced by gas. Meantime, the maximal potential of CO₂ emission abatement by PV and wind will decrease from 12.5 Gt CO₂ y⁻¹ in the central case to 11.6 Gt CO₂ y⁻¹ if coal is replaced by oil, or to 9.8 Gt CO₂ y⁻¹ if coal is replaced by gas, due to a lower emission factor of oil or gas than coal (0.84, 0.72 and

0.46 kg CO₂ kWh⁻¹ for coal, oil and gas, respectively) (Liu et al., 2015, National Bureau of Statistics of China, 2020, Zeiss, 2010). These new results showed how the maximal potential of emission abatement and system costs of PV or wind will depend on the future composition of fossil-fuel based power production in China.

To clarify the sensitivity of MACC to future changes in the composition of fossil-fuel based power production, we added the following paragraph in lines 216-227: “*We assumed that PV and wind power can be used to replace oil, gas and coal in the order of fuel price to generate the highest profits. Power generation by fossil fuel in China is currently dominated by coal⁴¹ with a contribution of 96% in 2020, but the fuel composition is uncertain in the future⁴. The maximal potential to abate CO₂ emissions by PV and wind decreases from 12.5 Gt CO₂ y⁻¹ by maintaining fuel composition at the current level to 11.6 or 9.8 Gt CO₂ y⁻¹ if the share of oil or gas increases from the current level⁴¹ of 0.3% or 4.1% to 50% by replacing coal in China’s power plants, respectively. Interestingly, a transition of coal to oil or gas reduces the maximal potential of emission abatements by PV and wind due to a lower emission factor of oil or gas than coal, but also reduces MAC due to a higher price of oil or gas than coal (Fig. S11). For example, MAC to abate 8 Gt CO₂ y⁻¹ by PV and wind power decreases from 18 USD (t CO₂)⁻¹ when maintaining fuel composition at the current level to -17 or 0.3 USD (t CO₂)⁻¹ if the share of oil or gas reaches 50%, respectively.*”.

To clarify our calculation of MACC based on the costs and composition of fossil-fuel based power production, we added the following paragraph in the **Methods** section in lines 901-916: “*We obtained the price of coal (0.043±0.015 USD kWh⁻¹)⁹⁵, oil (0.141±0.057 USD kWh⁻¹)^{96,97} and gas (0.058±0.016 USD kWh⁻¹)⁹⁸ as the 95% confidence interval during 2010–2020 when they were used to generate electricity with an electricity efficiency of 35, 38 and 45%, respectively⁹⁹ and an emission factor of 0.84, 0.72 and 0.46 kg CO₂ kWh⁻¹, respectively⁹³ (Fig. S11). We assumed that PV and wind power was used to replace oil, gas and coal in the order of fuel price to generate the highest profits. The electricity generation by fossil fuel during 2010–2020 in China was dominated by coal⁴¹ with a contribution of 96% in 2020, but the fuel composition is uncertain in the future. In our central case, the consumptions of coal, oil and gas for 2020–2060 were projected by scaling up their consumptions in 2020 with the projected growth of total electricity demand⁴¹, respectively. We performed two sensitivity tests to assess the impact of fuel composition on MACC of PV and wind power. First, the share of oil in power generation increases from the current level (0.3%) in 2020 to 50% in 2060 by replacing coal when the share of gas is identical to that in the central case. Second, the share of gas in power generation increases from the current level (4.1%) in 2020 to 50% in 2060 by replacing coal when the share of oil is identical to that in the central case.*”.

To show the impact of future changes in the composition of fossil-fuel based power production on MACC, a new **Fig. S11** was added to the **Supporting Information**:

Fig. S11. Impact of future changes in fuel composition on the marginal abatement cost (MAC) for deploying PV and wind power in China. (a) Fluctuation of the prices of coal, oil and gas for electricity generation during 2010–2020. The solid line represents the average price, while the shade indicates the 95% confidence interval. (b) MAC for PV and wind-power plants in 2060 in the central case, where the consumption of coal, oil and gas for 2020–2060 are projected by scaling up the consumption in 2020 with the projected growth rate of total electricity demand in the central case based on the fuel composition in 2020. (c) as (b), but for the sensitivity case, where the share of oil in electricity generation increases from the current level (0.3%) in 2020 to 50% in 2060 by replacing coal, while the share of gas is identical to the central case. (d) as (b), but for the sensitivity case, where the share of gas in electricity generation increases from the current level (4.1%) in 2020 to 50% in 2060 by replacing coal, while the share of oil is identical to the central case.

References:

- China National Petroleum Corp. Domestic fuel oil market annual Analysis report in 2011. URL: <https://view.officeapps.live.com/op/view.aspx?src=http%3A%2F%2Fofoilinfo.cnpc.com.cn%2Fypxx%2Fyjnb%2F201208%2F24404fbc0eed4f2280939eae3c4ea8ad%2Ffiles%2F8920787449ee417997c0147be45fce98.doc&wdOrigin=BROWSELINK> (2012).
- Liu, Z. *et al.* Reduced carbon emission estimates from fossil fuel combustion and cement production in China. *Nature* **524**, 335–338 (2015).
- International Energy Agency (IEA). *An Energy Sector Roadmap to Carbon Neutrality in China*. (International Energy Agency, Paris, France, 2021).

National Bureau of Statistics of China. *China Energy Statistical Yearbook 2020*. (China Statistics Press, Beijing, 2020).

National Development and Reform Commission. Price list of benchmark gate stations for non-residential natural gas in each province. URL: http://www.gov.cn/xinwen/2019-03/29/content_5378081.htm (2019).

Statista. China Qinhuangdao coal spot price from 2003 to 2019. <https://www.statista.com/statistics/383534/asian-coal-marker-price/> (2021).

Yte1. Fuel Oil (180CST) price trend chart in 2022. URL: <http://www.yte1.com/datas/ranliaoyou-pri?end=2022> (2022).

Zeiss, G. Energy Efficiency of Fossil Fuel Power Generation. URL: <https://geospatial.blogs.com/geospatial/2010/01/energy-efficiency-of-fossil-fuel-power-generation.html> (2010).

Comment B10

2. As far as I understand, the paper envisions a large number of very large solar and wind power plants. Many of them are envisioned to be dozens or even hundreds of GW capacity. The plants of these size are unprecedented or rarely observed empirically (for a variety of non-economic reasons). At the same time, limiting plant size significantly reduces the abatement potential of wind and solar power in China (173-175). Furthermore, the model seems to predict much larger plant size than empirically observed (Fig. S8). I would think therefore that the effect of limiting the plant size should be highlighted more prominently, including for MACC and other relevant calculations. The paper might want to mention the non-economic limitations of building such giant plants.

Response

We have carefully examined the technical feasibility of building large PV and wind-power plants. Large PV-power plants have been commissioned in China, such as the 2.2 GW Gonghe PV-power plant initiated in 2020, the 1 GW Yanchi PV power plant initiated in 2016 and the 800 MW Datong PV power plant (PV Magazine, 2021). Large onshore wind-power plants have been commissioned in the west of China, such as the 20 GW Jiuquan onshore wind power plant initiated in 2020 and the 1.5 GW Alta onshore wind power center initiated in 2012 (Power Technology, 2019). Large offshore wind-power plant has been built in Europe, such as the 659 MW Walney Extension wind farm in the Irish Sea (Power Technology, 2019). Although these large power plants are technically feasible, they all require upgraded grid connection, long-distance electricity transmission and large initial investment costs (International Energy Agency, 2015), which are considered when estimating the system costs of deploying PV and wind power in our optimisation model.

In addition, we considered non-economic factors limiting the size of power plants. We believe that a key issue could be the legacy of monopoly control (Farrell, 2016). The economies of size debate for these power plants are not so much about the costs of power generation as it is who retains the power of ownership over electricity systems. If the utility owns large solar and wind projects, it will need to deploy capitals to bolster the transmission and storage systems and earn a return on those investments. In China, the central government has the control over the electricity systems (Lam, 2004) and has the ability to cover the large investments in electricity transmission and energy storage, but the problem is how to distribute the revenue of these investments. We discussed the impact of these non-economic factors in our revised paper.

We performed additional experiments using our optimisation model to further examine the

impact of increasing the maximal size of power plants on MAC, initial investments and the revenue of PV and wind-power plants in the new **Fig. S28**. We found that consideration of these large power plants (>10 GW) has the effect to reduce MAC and to increase the maximal potential of CO₂ emission abatement, but it also generates higher requirements of initial investments for each power plant despite the higher revenue. We agree very much with the Reviewer that these non-economic factors limiting the size of PV/wind power plants, such as ecological impacts (Hernandez et al., 2014), technological monopoly (Schmidt et al., 2009) and political impediments (Sovacool, 2009), should be considered in the projection of large PV and wind-power plants in China.

Therefore, these new results suggested that large PV and wind-power plants, despite their engineering challenges, are feasible and taking them into consideration has the effect to make renewable energy toward carbon neutrality more affordable in China.

To clarify the non-economic factors limiting the size of PV and wind-power plants, we added the following new paragraph in lines 579-599: *“Lastly, our model envisions large PV and wind-power plants, e.g. 856 and 183 plants with installed capacity exceeding 1 GW and 10 GW, respectively. Given commissions of the 20 GW Jiuquan onshore wind power plant in operation since 2020⁷⁰ and the 1 GW Yanchi PV power plant in operation since 2016⁷¹, these large power plants are technically feasible⁷², but they have high requirements of grid connection, electricity transmission and initial investments. Consideration of these power plants has the effect to reduce MAC and increases the potential of emission abatements by PV and wind, but it also increases initial investments for each power plant (**Fig. S28**). Relative to our central case, the maximal potential of PV and wind to abate emissions decreases from 12.5 to 0.5, 2.2 and 6.3 Gt CO₂ y⁻¹ with an increase in the average MAC from -2 to 38.7, 11.6 and -1 USD (t CO₂)⁻¹, respectively, when limiting the size of power plant to 100 MW, 1 GW or 10 GW, respectively; the average initial investments of each power plant decrease from 1.55 to 0.15, 0.43 and 0.95 billion USD, respectively. Many non-economic factors limit the size of power plants, such as ecological impact⁷³, engineering feasibility⁷⁴ and political impediment⁷⁵, but a key issue might be the legacy of monopoly control¹⁶ in China. The debate of the size of power plant largely depends on the ownership of electricity systems, who deploys capitals to bolster transmission and storage systems and earns a return on those investments¹⁶. In China, the central government has the control of electricity systems⁷⁶ and has ability to cover the required investments in system upgrade, but the problem is how to distribute the revenue of investments. These socio-political factors⁷⁶ may slow the pace of building large PV and wind-power plants, affect the prices of power generation by PV and wind and increase overall costs of emission mitigation.”.*

To show the impact of increasing the maximal size of power plants on MAC, initial investments and the revenue of each PV and wind-power plant, a new **Fig. S28** was added to the **Supporting Information**:

Fig. S28. The effect of increasing the maximal size of power plant on the marginal abatement cost (MAC), the initial investments and the revenue of PV and wind-power plants in China. (a) Cumulative electricity generation by PV and wind-power plants in terms of MAC for each power plant. The cumulative number of power plants is plotted in the inserted panel. (b) MAC curves for deploying PV and wind power in 2060 when the capacity of each PV and wind-power plant is limited to 100 MW, 500 MW, 1 GW, 10 GW and 50 GW, respectively. (c) Relationship between the initial investments and the installed capacity of each power plant. (d) Relationship between the revenue under a carbon price of 100 USD (tCO₂)⁻¹ and the installed capacity of each power plant.

References:

- Farrell, J. Is Bigger Best in Renewable Energy? URL: <https://ilsr.org/report-is-bigger-best/> (2016).
- Lam, P. L. Pricing of electricity in China. *Energ.* **29**, 287–300 (2004).
- Schmidt, R. C. & Marschinski, R. A model of technological breakthrough in the renewable energy sector. *Ecol. Econ.* **69**, 435–444 (2009).
- Sovacool, B. K. Rejecting renewables: The socio-technical impediments to renewable electricity in the United States. *Energ. Polic.* **37**, 4500–4513 (2009).
- Hernandez, R. R. *et al.* Environmental impacts of utility-scale solar energy. *Renew. Sustain. Energ. Rev.* **29**, 766–779 (2014).
- PV Magazine. An overview of the world's largest solar power plants. URL: <https://www.pv-magazine.com/2019/06/18/an-overview-of-the-worlds-largest-solar-power-plants/> (2021).
- Power Technology. Top 10 biggest wind farms. URL: <https://www.powermag.com/top-10-biggest-wind-farms/>

<https://www.power-technology.com/analysis/feature-biggest-wind-farms-in-the-world-texas/> (2019).

International Energy Agency. Energy from the Desert: Very Large Scale PV Power Plants for Shifting to Renewable Energy Future. (International Energy Agency, Paris, France, 2015).

Comment B11

3. The manuscript focuses on the end-point of CNZ by 2060 but it does not analyse the rate of deployment required for reaching this point (beyond optimisation with respect to the learning rates). At the same time, the feasible rates of deployment of renewables and other low-carbon technologies have been of concern in prior studies e.g. [3-5]. Are the required rates of deployment comparable to those so far observed in different countries, including in those that devote a lot of resources to energy transitions (e.g. UK, Germany). This may also help to signal the overarching limitation of the manuscript which is focusing on economic aspects of renewables deployment whereas in real life they're also limited by socio-political factors.

Response

We have performed a comparison of the projected increase in the capacities of PV and wind power in China from our study with the projected increase in the other regions of the world from previous studies (Edenhofer et al., 2011; IEA, 2017; Carrara et al., 2020) (see our response to **Comment A17**). We have also compared the projected capacities of PV and wind power in China from our study with the capacities in the regional renewable energy development plans in China according to the energy transition and 14th Five-year Power planning (FYPP) (Global Energy Interconnection Development and Cooperation Organization, 2020) (see our response to **Comment C9**). We found that the projected increase in the capacities of PV and onshore wind power in China is faster than Europe due to a higher fraction of fossil fuel in 2020 for China (71%) (Bloomberg News, 2022) than Europe (37%) (Green News, 2021) (**Fig. S27**). We also found that the projected capacities of China's PV and wind power in our model are very close to the projection by 2030 in FYPP, but the installed capacities are underestimated by FYPP for the future period of 2030–2060 without considering the effect of learning.

To clarify the difference in the rate of increase in the projected capacities of PV and wind power between China and the other regions of the world (Edenhofer et al., 2011; IEA, 2017; Carrara et al., 2020), we added the following paragraph in lines 555-567: *“Learning-by-doing¹⁷ brings abatement efforts forward, where the projected PV and wind capacities will be completed by 80% by 2050. The time of deploying renewable energy depends on appropriate discounting and technological spillovers⁶⁶. Using a discounting rate of 7% y^{-1} by Chen et al.¹⁵ rather than 5% y^{-1} by Duan et al.², the fraction of PV and wind capacities installed by 2050 will decrease from 77 to 75% (Fig. 3a). The effect of global spillovers on the rate of deploying PV and wind in China might be moderate, because the increase in capacities of PV and onshore wind power is faster in China than that projected for the other regions of the world⁶⁷⁻⁶⁹. If we apply the relative increase in global or European PV and wind capacities, the decline of capital costs would be slower than we estimated (**Fig. S27**). The projected increase in offshore wind power is faster in Europe⁶⁷ than for China during 2030–2050, implying a faster decline of capital costs by considering the technological transfer from Europe to China, but the installed capacities of offshore wind are far lower than for PV and onshore wind in our projection.”*

To clarify the difference in the projected capacities of PV and wind power in China between our model and the regional renewable energy development plans (Global Energy Interconnection Development and Cooperation Organization, 2020), we added the following

paragraph in lines 275-292: “We compared the projected capacities of PV, onshore wind and offshore wind by decade between our optimisation model and the regional renewable energy development plans according to China’s energy transition and 14th Five-year Power planning (FYPP)⁵⁰. The projected increase in the capacities of PV and wind power in China is faster in our optimisation model than for FYPP⁵⁰ (Fig. S12). Interestingly, the difference is small by 2030, but increases dramatically from 2030 to 2060, indicating an increasing underestimation of the potential of PV and wind power in the longer term by FYPP⁵⁰. The projected capacities of PV and wind power increase from 1.2 TW in 2020s to 1.7 TW in 2040s by FYPP⁵⁰, compared to an increase from 1.1 TW to 3.6 TW in our optimisation model. This difference is partly due to the fact that the effect of learning¹⁷ is not considered by FYPP⁵⁰. If capacities of PV and wind in our optimisation model are constrained by FYPP⁵⁰, MACC moves leftward and upward relative to our central case, while the maximal potential of emission abatements by PV and wind decrease from 12.6 to 5.8 Gt y⁻¹ in 2060. Spatially, additional PV and wind-power plants in our optimisation model beyond the FYPP projection⁵⁰ are mainly distributed in the Central and North of China, where 57% of power plants are subject to MAC of 0 to 20 USD (t CO₂)⁻¹ (Fig. S12). Although exclusion of plants subject to high MAC makes PV and wind power more affordable in FYPP⁵⁰ than our optimisation model, its projection of PV and wind leads to a higher demand of CCS than our projection to achieve the national target of C neutrality⁶.”.

To compare the rate of projected increase in the capacities of PV and wind power between China and the other regions of the world (Edenhofer et al., 2011; IEA, 2017; Carrara et al., 2020), a new Fig. S27 was added to the Supporting Information:

Fig. S27. Comparison of the projected capacity and decline of capital costs for PV and wind-power plants between China and other regions. (a) Comparison of the projected relative increase in the installed capacity of PV and onshore and offshore wind power plants in China from this study during 2020–2060 and in Europe by Carrara et al. (2020) and the

projected global PV and wind power estimated by the International Energy Agency (IEA) (IEA, 2017) or Edenhofer et al. (2011). The installed capacity in 2020 is normalized to 1 for each estimate. (b) The projected decline of capital costs of PV and onshore and offshore wind-power plants due to learning-by-doing during 2020–2060 based on the projected relative increase in the installed capacity of PV and onshore and offshore wind-power plants. (c) Sensitivity of the marginal abatement cost (MAC) curve for PV and wind power in China to using the projected decline of capital costs based on the relative increase in the installed capacity of PV and wind power in China from this study, based on the relative increase in the installed capacity of PV and wind power in Europe (Carrara et al., 2020) or based on the relative increase in the installed global capacity of PV and wind power from two different estimates (Edenhofer et al., 2011; IEA, 2017).

To show the difference in the projected capacities of PV and wind power in China between our model and the regional renewable energy development plans (Global Energy Interconnection Development and Cooperation Organization, 2020), a new Fig. S12 was added to the **Supporting Information**:

Fig. S12. Comparison of the projected deployment of PV and wind-power plants by 2060 between the optimisation model and the regional renewable energy development plans in China. (a) Difference in the installed capacities of PV, onshore wind and offshore wind-power plants by decade between China's energy transition and 14th Five-year Power planning (Global Energy Interconnection Development and Cooperation Organization, 2020)

and the projection in the central case of our model optimisation. (b) Impact on the marginal abatement cost (MAC) curve by constraining the model optimisation with the projected regional capacities of PV, onshore wind and offshore wind power in the China's energy transition and 14th Five-year Power planning (Global Energy Interconnection Development and Cooperation Organization, 2020) by decade. (c–f) Spatial distributions of PV and wind-power plants in the central case of the model optimisation as a difference to the projection constrained by China's energy transition and 14th Five-year Power planning (Global Energy Interconnection Development and Cooperation Organization, 2020) in terms of the MAC range.

References:

Global Energy Interconnection Development and Cooperation Organization. Research on China's energy transition and 14th Five-year Power planning (2020).

Carrara, S. *et al.* Raw materials demand for wind and solar PV technologies in the transition towards a decarbonised energy system. Luxembourg: Publications Office of the European Union. URL:

https://eitrawmaterials.eu/wp-content/uploads/2020/04/rms_for_wind_and_solar_published_v2.pdf (2020).

Edenhofer, O. *et al.* Renewable energy sources and climate change mitigation: Special report of the intergovernmental panel on climate change. Cambridge University Press. URL: https://www.ipcc.ch/site/assets/uploads/2018/03/SRREN_Full_Report-1.pdf (2011).

International Energy Agency. *Energy technology perspectives 2017: Catalysing Energy Technology Transformations* (2017).

Bloomberg News. China Remains as Reliant as Ever on Fossil Fuels. URL: <https://www.bloomberg.com/news/articles/2022-01-19/two-charts-that-show-china-is-as-reliant-as-ever-on-fossil-fuels#:~:text=China%20remains%20as%20tied%20as,%25%2C%20the%20same%20as%202020> (2022).

Green News. More of Europe's electricity came from renewables than fossil fuels in 2020. URL:

<https://www.euronews.com/green/2021/01/30/renewables-beat-fossil-fuels-in-europe-for-the-first-time-in-2020#:~:text=More%20of%20Europe's%20electricity%20came%20from%20renewables%20than%20fossil%20fuels%20in%202020,-Wind%20and%20solar&text=Renewable%20generated%20more%20electricity%20than,to%20a%20report%20published%20today> (2021).

Comment B12

4. It is not clear how the validation of the model with space observations was done and therefore what are the limitations of such validation.

Response

We are unable to validate our optimisation model that projected the location and capacity of future potential power plants to be built during 2023–2060, so we only evaluated the performance of our model by comparing the identified location and the projected capacity of each power plant with the closest power plant from the state-of-the-art harmonised dataset describing global solar PV and wind turbine installations using OpenStreetMap (Dunnnett *et al.*, 2020). In detail, we obtained the location and projected capacity of each PV or wind-power plant from our optimisation model. We then sought the actual PV or wind-power plant in the OpenStreetMap dataset (Dunnnett *et al.*, 2020) that is the closest to a power plant in our

optimisation model. We calculated the spatial distance between the actual PV or wind-power plant and the predicted power plant. We compared the projected capacity of the power plant normalized by the current area with the actual capacity of the power plant in the OpenStreetMap dataset (Dunnett et al., 2020). This model evaluation assumes that the projected power plant and the closest actual power plant is the same one, but we acknowledged that it is impossible to obtain the actual information for the projected power plant before it is really built in the future.

To clarify the limitation in validating our optimisation model, we added the following discussion in lines 155-165: “*The location and capacity of projected PV and wind-power plants are compared to the observed location and capacity in a harmonised dataset describing PV panel and wind turbine installations using OpenStreetMap⁴⁰ (Fig. S9). We are unable to validate our optimisation model for future power plants, so we sought PV or wind-power plant with actual location and capacity in OpenStreetMap⁴⁰ that is the closest to a power plant in our model. We calculated the distance between the projected and actual power plants, and compared the projected capacity of power plant normalized by the current area with the actual capacity. This model evaluation lends support to building a power plant close to the predicted location at the actual level of capacity. A validation of our model requires detailed data of power plants when they are built in the coming decades.*”.

Reference:

Dunnett, S., Sorichetta, A., Taylor, G. & Eigenbrod, F. Harmonised global datasets of wind and solar farm locations and power. *Sci. Data* 7, 1–12 (2020).

Comment B13

Clarity

I find the figures presented in the manuscript very clear. At the same time I think the flow of text should be improved. In particular, the abstract contains many references but I think it should be more self-contained and explain the background and what are the main results. The rest of the text could be more directed at a wider audience not familiar with special concepts and the wider significance of the problems that the manuscript addresses should be explained. Conclusions should not repeat what the paper has done but rather focus on the implication of the results.

Response

Thank you very much for the positive comment.

We revised the **Abstract** to focus on explaining the general background and presenting our key findings. We also revised the **final paragraph** section to broaden the implications of this study relative to other fields to match the wider readership of the Journal (see our response to **Comment B8**). To meet a wider audience and address the problem at a wider significance, we revised the **Abstract**: “*The pledge toward carbon neutrality in China implies a power demand^{1,2} of 7,000–20,000 TWh y⁻¹ with capacities exceeding 8,000 GW¹ from renewable energy. With currently installed capacities of 530 GW in China, photovoltaic (PV) and wind power have a significant potential to be scaled-up. However, a large-scale deployment depends on the marginal abatement cost (MAC) of mitigating CO₂ emissions, which remains unclear. We developed a spatially explicit method to optimize the size and location of 3,844 utility-scale PV and wind-power plants in China, and the time needed to build each plant during 2023–2060 to minimize system costs. The first MAC curve estimated at plant level showed a negative MAC to abate emissions by 6 Gt CO₂ y⁻¹ with capacities of 4,600 GW in the country, but the MAC will rise at higher levels of deployment even if considering*

the effect of learning. This challenges the widespread view that the costs of renewable energy will keep declining as their deployment grows to produce net negative costs of energy transition. We assessed the sensitivity of MAC curve by considering uncertainties in ultra-high-voltage transmission and storage systems, deployment of PV, onshore or offshore wind power alone, rate of electrification, capacities of other renewable energy, flexible power load, marginal curtailment, effects of learning, capital costs, lifetime of power plants, regional renewable development plans, discounting rate and non-economic factors limiting the size of power plants. Despite high investment costs, deploying PV and wind power at large scale reduces inequalities in the country, with income opportunities for residents in the poorest regions. Our results suggested that a stimulus of investments by one to two trillion US dollars with facilities of electricity transmission and storage could initiate a rapid transition from fossil fuel to PV and wind power in China by 2040, which could be incorporated into a legal, institutional and social-ecological framework that promotes investments in renewable energy in the long term and accelerates decarbonization to achieve national carbon neutrality by 2060.”.

Comment C1

General comment

The authors develop a spatially explicit approach to discuss the future construction and layout of wind and PV in China. The methodology of the study is detailed and transparent, and the robustness of the results is confirmed by extensive sensitivity analysis. However, the authors need to improve the methodology and refine the model to better fit the reality. In addition, the study needs to take more policy considerations into account, especially the recent energy development plans of each province and region, to make it more policy influential and of public interest.

Response

We are pleased to see that our methodology is transparent and our results are robust.

We have further improved our model by adopting the flexible power load with the optimal option of energy storage between pumped hydro and chemical batteries. These improvements allow us to better represent the demand-side responses to a large-scale deployment of PV and wind in China (see our response to **Comment A2**).

We also examined the sensitivity of our results to the lifetime of PV and wind-power plants (see our response to **Comment C4**), discounting rate (see our response to **Comment C5**), using different data of capital costs (see our response to **Comment C6**) and total electricity demand under different rates of electrification (see our response to **Comment C8**).

In addition, we have also compared the projected decadal provincial capacities of PV, onshore wind and offshore wind power between our optimisation model and the regional renewable energy development plans according to China’s energy transition and 14th Five-year Power planning (FYPP) (see our response to **Comment C9**). We found that the projected capacities of China’s PV and wind power in our model are very close to the projection by 2030 in FYPP, but the installed capacities are underestimated by FYPP for the future period of 2030–2060 without considering the effect of learning in FYPP.

We hope that our revised paper can address the Reviewer’s comments by making our model better fit the reality and allowing our paper to be more policy relevant.

Comment C2

Line 67-68

The authors mention that MAC for PV and wind power varies from –US\$50 to US\$500 (tCO₂)⁻¹ in previous estimates (Supplementary Spreadsheet S1). However, many of these literatures provide incremental abatement cost (not MAC) or national/provincial level MAC (eg. “The national MAC is \$69/ton CO₂ in 2007 in China, and Beijing is the highest (\$417/ton CO₂) and Jiangsu is the lowest (\$0.1/ton CO₂)”)

Response

We have carefully examined these data by excluding incremental abatement costs and providing more detailed information for each data from previous studies in the new **Supplementary Spreadsheet S1**.

We added the new **Fig. S1** to compare the estimated power capacity, electricity generation, LCOE and MAC between our study and previous studies (see our response to **Comment B2**). In the new **Fig. S1**, the error bars show the ranges of LCOE and MAC for different provinces

in China. We only used the data with explicit information for LCOE, MAC, power capacity or electricity generation for a comparison in the new **Fig. S1**.

By gathering data in the new **Fig. S1**, we derived the 90% uncertainty of MAC of -22 to 99 USD $(t\ CO_2)^{-1}$ for PV, -16 to 45 USD $(t\ CO_2)^{-1}$ for on-shore wind and -15 to 80 USD $(t\ CO_2)^{-1}$ for off-shore wind, respectively.

To clarify the variance of MAC in previous studies, we revised the sentence in lines 91-94: “*The 90% uncertainty of MAC estimated in China varied from -22 to 99 USD $(t\ CO_2)^{-1}$ for PV, -16 to 45 USD $(t\ CO_2)^{-1}$ for on-shore wind and -15 to 80 USD $(t\ CO_2)^{-1}$ for off-shore wind, respectively (Supplementary Spreadsheet S1 and Fig. S1).*”.

Comment C3

Line 121-122

The authors mention that 14.4 PWh is half of the electricity supply in 2060. This value substantially exceeds the currently available studies related to IAM and energy system models.

Response

We compared our estimate of electricity generation with previous studies in the new **Fig. S1** (see our response to **Comment B2**).

We found that the 14.4 PWh of electricity generation by PV and wind was higher than the top quartile of previous estimates, mainly due to consideration of large PV and wind-power plants in our optimisation model. We have carefully examined the feasibility of deploying these large PV and wind-power plants, assessed their contribution to power generation, and considered the impact of non-economic factors on limiting the size of power plants (see our response to **Comment B10**).

To clarify the difference in electricity generation between our study and previous studies, we added the following sentences in lines 228-245: “*We compared LCOE, MAC, electricity generation and power capacity of PV and wind between our study and previous studies (Fig. S1). Many studies estimated LCOE for PV and wind, but the MAC is lacking in the literature. Our estimate of LCOE (0.04 USD kWh^{-1} for all plants) is closer to the bottom quartile (0.03 USD kWh^{-1}) of previous estimates than to the average (0.07 USD kWh^{-1}), while previous estimates of MAC are higher or close to the values on our MACC. A recent study¹⁴ estimated MAC at 27 USD $(t\ CO_2)^{-1}$ for PV and wind power under an 80% renewable penetration in China, when they assumed that 4 TW of PV and wind capacities would be installed by 2050 without considering the effect of learning and flexible power load; MAC for 4 TW is -6 (90% uncertainty of -21 to 18) USD $(t\ CO_2)^{-1}$ in our estimate. Lu et al.¹⁵, without providing the MAC, suggested an electricity price of PV at 0.03 USD kWh^{-1} for 2060, compared to a LCOE of 0.024 – 0.057 USD kWh^{-1} in our estimate, likely because Lu et al.¹⁵ adopted a high learning rate (42.4% by Lu et al.¹⁵ against 7–20% in our **Table S2**) and Lu et al.¹⁵ did not include the costs of transmission (0.002 – 0.006 USD kWh^{-1} in our estimate). These results suggested that, by optimizing the placement and timing of PV and wind-power plants with flexible power load, transmission and energy storage in China, MAC of PV and wind power could be lower than previously thought, with higher potential of emission abatements.*”.

To compare the estimated power capacity, electricity generation, LCOE and MAC between our study and previous studies, a new **Fig. S1** was added to the **Supporting Information**:

Fig. S1. Comparison of levelized cost of electricity (LCOE) and marginal abatement cost (MAC) for solar photovoltaic (PV) and wind-power plants in China between this study and previous studies. (a, b) A scatter plot of LCOE against electricity generation (a) and installed capacity (b) for PV (circle), onshore wind (square) and offshore wind (pentagram) power in China. Estimates from this study are shown by the crosses. (c, d) A scatter plot of MAC against electricity generation (c) and installed capacity (d) for PV (circle), onshore wind (square) and offshore wind (pentagram) power. The MAC curves are estimated in this study. The size of the symbol indicates the year of the estimate (i.e. the smallest one for the estimate of 2015 and the largest one for the estimate in 2060). The error bars show the range of estimate for different provinces in China. The data in the literature are listed in **Supplementary Spreadsheet S1**, while only the data with explicit information for LCOE, MAC, electricity generation and power capacity are included in this comparison.

Comment C4

Line 130-135

The authors assume the lifetime for PV and wind to be 40 years, much higher than other studies (20-30 years). Sensitivity analysis for the lifetime of PV and wind might need to be presented.

Response

We realize that the lifetime of 40 years adopted in our previous paper is too long for PV and wind-power plants. In our revised paper, we adopted a lifetime of 25 years for PV and

wind-power plants (Victoria et al., 2021) in the central case, and performed additional experiments by examining the sensitivity of MACC to adopting a lifetime of 15 or 35 years (Jordan et al., 2016) in the new Fig. S21. We found that a longer lifetime of power plants has the effect to reduce MAC of PV and wind-power plants.

To clarify the impact of increasing the lifetime of PV and wind-power plants on MAC, we added the following sentence in lines 391-395: “*We adopt a lifetime of 25 years for PV and wind-power plants, but a longer lifetime of power plants (e.g. using crystalline silicon PV modules⁵⁴) has the effect to reduce costs, where an increase in the lifetime of plants from 15 to 35 years reduces the average MAC for PV and wind-power plants from 10 to -7 USD (t CO₂)⁻¹ (Fig. S21).*”.

To show the impact of increasing the lifetime of PV and wind-power plants on MAC, a new Fig. S21 was added to the Supporting Information:

Fig. S21. Sensitivity of the marginal abatement cost (MAC) curve to the lifetime of PV and wind-power plants. The extension of lifetime is a crucial factor to reduce LCOE (Victoria et al., 2021), while it is possible to reach a long lifetime of 25 years by deploying crystalline silicon modules (Jordan et al., 2016).

References:

Victoria, M., Zhu, K., Brown, T., Andresen, G. B. & Greiner, M. Early decarbonisation of the European energy system pays off. *Nat. Commun.* **11**, 1–9 (2020).

Jordan, D.C., Kurtz, S.R., VanSant, K. & Newmiller, J. Compendium of photovoltaic degradation rates. *Prog. Photovolt.: Res. Appl.* **24**, 978–989 (2016).

Comment C5

In addition, although the study includes a sensitivity analysis for the discount rate, a discount rate of 7% may be slightly high for China. Using this discount rate as the central case may underestimate near-term effort requirements.

Response

In our revised paper, we adopted a discounting rate of 5% y⁻¹ (Duan et al., 2021) in the central case. We examined the sensitivity of the timing of deploying PV and wind power to increasing the discounting rate from 3 to 7% y⁻¹ (Lu et al., 2021) (Fig. S20). We found that using a higher discounting rate has the effect to delay the deployment by reducing the fraction of capacities of PV and wind power installed by 2050 from 80 to 75%, because a lower discounting rate places more weight on the future value of electricity generation and thus

brings the efforts of abatement forward.

To clarify the impact of increasing the discounting rate on the timing of deploying PV and wind power, we added the following sentences in lines 388-391: “*The total installed capacities of PV and wind power reach 7.3 TW by 2050 under a discounting rate of 5% y⁻¹, but the timing of abatement depends on the rate of discounting. The fraction of total PV and wind capacities installed by 2050 decreases from 80 to 75% if the discounting rate increases from 3 to 7% y⁻¹ (Fig. S20).*”.

To show the impact of increasing the discounting rate on the deployment of PV and wind power, a new **Fig. S20** was added to the **Supporting Information**:

Fig. S20. Deployment of photovoltaic (PV), onshore wind and offshore wind-power plants by decade under different discounting rates. The installed capacities of PV, onshore wind and offshore wind-power plants by decade are projected under a discounting rate (r_d) of 3 to 7% y^{-1} , which is compared to the central case under a r_d of 5% y^{-1} .

References:

Duan, H. *et al.* Assessing China’s efforts to pursue the 1.5°C warming limit. *Science* **372**, 378–385 (2021).

Lu, X. *et al.* Combined solar power and storage as cost-competitive and grid-compatible supply for China’s future carbon-neutral electricity system. *Proc. Natl. Acad. Sci. U.S.A.* **118**, e21034711118 (2021).

Comment C6

In addition, after comparison, there is a discrepancy between the cost data of the 36th reference cited by authors and the data published by the China Electricity Council, to which the authors may make additional explanations.

Response

The estimate of capital costs (0.73 and 0.88 USD W^{-1} for PV and onshore wind, respectively) used by Lorenczik *et al.* (2020) is higher than the estimate of capital costs (0.23 and 0.76 USD

W^{-1} for PV and onshore wind, respectively) published by the China Electricity Council (2021). However, we do not adopt the data from either of these two references in our central case, because neither of them provides capital costs for each component. We adopted the capital costs by component for PV-power plants using the data published by the China Photovoltaic Industry Alliance (2020) and for onshore wind-power plants using the data published by Liu et al. (2015), which are listed in **Table S8**.

For a comparison, the capital costs in our central case (0.64 and 0.68 USD W^{-1} for PV and onshore wind, respectively) are between the estimates of these two references (Lorenczik et al., 2020; China Electricity Council, 2021). To show the impact of using different data of capital costs on MAC, we performed new experiments to compare the MAC in our central estimate with MAC calculated using the data from either of these two references (Lorenczik et al., 2020; China Electricity Council, 2021) in the new **Fig. S22**.

To account for uncertainties in capital costs, we adopted the range of capital costs from different sources of data (Lorenczik et al., 2020; China Electricity Council, 2021; China Photovoltaic Industry Alliance, 2020; Liu et al., 2015) in our Monte Carlo simulations when estimating uncertainties in MAC in the new **Fig. 1a**.

To clarify the impact of using different data of capital costs on MAC, we added the following sentence in lines 395-398: “*In addition, we examined the sensitivity of MACC to different capital costs. The average MAC for PV and wind-power plants increases from -2 USD ($t\ CO_2$) $^{-1}$ using capital costs by component (Table S8) to 1 USD ($t\ CO_2$) $^{-1}$ if we adopt high capital costs³⁹, or decreases to -6 USD ($t\ CO_2$) $^{-1}$ if we adopt low capital costs⁵⁵ (Fig. S22).*”.

To show the impact of using different data of capital costs on MAC, a new **Fig. S22** was added to the **Supporting Information**:

Fig. S22. The impact of capital costs on the marginal abatement cost (MAC) curves for PV and wind-power plants in 2060. The MAC curves in our central estimate for PV and wind power (blue), PV only (orange) and onshore wind only (green) are compared with the MAC curves calculated by using the data from two references (Lorenczik et al., 2020; China Electricity Council, 2021). The estimate of capital costs (0.73 and 0.88 USD W^{-1} for PV and onshore wind, respectively) by Lorenczik et al. (2020) is higher than the estimate of capital costs (0.23 and 0.76 USD W^{-1} for PV and onshore wind, respectively) published by the China Electricity Council (2021). Because these two references do not provide the cost for each component, we adopt the costs by component for PV-power plants (0.64 USD W^{-1} as the total

capital costs) using the data published by the China Photovoltaic Industry Alliance (2021) and for onshore wind-power plants (0.68 USD W⁻¹ as the total capital costs) using the data by Liu et al. (2015b) to estimate the MAC curves in the central case (solid line).

References:

China Electricity Council. Main indicators of national economy in 2020. URL: <https://www.cec.org.cn/upload/1/editor/1640595481946.pdf> (2021).

China Photovoltaic Industry Alliance. China PV industry development roadmap (2020). URL: <https://www.ccidgroup.com/info/1105/32621.htm> (2021).

Liu, Z., Zhang, W., Zhao, C. & Yuan, J. The economics of wind power in China and policy implications. *Energ.* **8**, 1529–1546 (2015).

Lorenczik, S. et al. *Projected costs of generating electricity-2020 edition*. (Organisation for Economic Co-Operation and Development, Paris, France, 2020).

Comment C7

Line 135-138

The study notes that the average capacity of these 80 plants is 32.5 GW. with current installed capacity of nuclear and thermal plants ranging from 1-4 GW, and with such a large individual plant set-up, the study needs to consider engineering feasibility.

Response

We have carefully examined the feasibility of building these large PV and wind-power plants, assessed their contribution to power generation and considered the potential impact of these non-economic factors on limiting the size of future PV and wind-power plants (see our response to **Comment B10**).

Comment C8

Line 158-159

The main objective of the study is the layout of renewable energy sources, and the authors cite the IEA report on carbon neutrality in China as a value for the electrification rate, and perform sensitivity tests for the absence of electrification in the industrial and transportation sectors. In addition, the authors should refer to other studies that use energy system models to calculate cases that include higher electrification rates or perform sensitivity analysis directly on total electricity demand.

Response

We have revised our model by considering flexible power load on the demand side under different rates of electrification and adopting the optimal option of energy storages between pumped hydro and chemical batteries. These improvements allow us to better represent the demand-side responses to a large-scale deployment of PV and wind power in China (see our response to **Comment A2**).

Following this comment, we performed additional experiments where the rate of electrification varies from 0 to 100% to achieve different total electricity demand in the new **Fig. 2e.f**. Interestingly, we found that, by deploying the flexible power load with transmission and energy storage, the rate of electrification has a small impact on the MAC curve, because the power generation can almost match the power demand and thus total electricity demand is less of a limiting factor to deploy PV and wind power in our central case relative to a

sensitivity case using the fixed power load without transmission and energy storage.

To clarify the impact of total electricity demand under different rates of electrification on the MACs of PV and wind, we added the following sentence in lines 329-334: “*Interestingly, by optimizing power load to match power generation, the impacts of electrification rate (Fig. 2e,f) and power generation by other renewable energy (Fig. S19) on MAC both become small at the left side of MACC, indicating that low MACs to abate emissions by PV and wind are relatively robust with respect to uncertainties in the rate of electrification¹⁴ and deployment of other renewable energy in electricity systems².*”.

To show the impact of total electricity demand under different rates of electrification on MACC, a new Fig. 2e.f was added to Fig. 2:

Fig. 2 | Impacts of hourly power load, ultra-high-voltage (UHV) transmission and energy storage on the efficiency and costs of deploying PV and wind power in China. (a, b) Levelized cost of electricity (LCOE) (black lines) at plant level and power-use efficiency (PUE) for all plants (blue lines) when (a) using fixed power load or (b) optimizing the flexible power load to match the hourly power generation by PV and wind. (c, d) Seasonal variations in power generation and demand when (c) using fixed power load or (d) optimizing the flexible power load with or without UHV and storage when the installed capacities of PV and wind power reach 8 TW under an electrification rate of 58% by 2060⁴¹. (e, f) Changes in marginal abatement cost (MAC) curve when 0%, 20%, 40%, 60%, 80% or 100% of fossil fuel in

non-power sectors is electrified when (e) using fixed power load without UHV or energy storage or (f) optimizing the flexible power load with UHV and storage.

Comment C9

Line 196-197

This figure is the highlight of this paper. However, the information given in this figure is the result considering only optimality, so the layouts vary very much under different capacity levels of wind and PV deployment, which will confuse decisionmakers. Currently, most provinces and regions already have their own renewable energy development plans, but they have not been considered in this study.

Response

We agree very much with the Reviewer for comparing the projected capacities of PV and wind power in China between our optimisation model and the regional renewable energy development plans. We compiled the projected provincial capacities of PV, onshore wind and offshore wind-power plants by decade from the regional renewable energy development plans according to China's energy transition and 14th Five-year Power planning (Global Energy Interconnection Development and Cooperation Organization, 2020). Based on these regional development plans, we performed additional experiments to estimate MACC when the projected provincial capacities of PV and wind power plants in our optimisation model were constrained by the projected provincial capacities of PV, onshore wind and offshore wind-power plants by decade (**Fig. S12**).

We found that the MAC curve would move leftward and upward by constraining our model with the projected capacities in the regional renewable energy development plans, indicating that the potential of PV and wind power has been underestimated by the regional renewable energy development plans (Global Energy Interconnection Development and Cooperation Organization, 2020), leading to overestimation of the costs of deploying PV and wind power. In particular, the difference in the installed capacities is small in the coming decade, but the difference increases dramatically from 2030 to 2060, indicating a larger underestimation of the potential of PV and wind power by regional renewable energy development plans in the long term. This underestimation is likely due to the fact that the effect of learning is not considered in these regional development plans (Global Energy Interconnection Development and Cooperation Organization, 2020), because learning has the effect to accelerate the deployment of renewable energy in the middle of this century (Victoria et al., 2020, 2021). Spatially, the potential PV or wind-power plants in our optimisation model beyond the regional renewable energy development plans are mainly distributed in the Central and North of China, and 57% of these power plants are subject to the MAC between 0 and 20 USD (t CO₂)⁻¹. These results suggested that the regional renewable energy development plans could have underestimated the potential of PV and wind power in China (Global Energy Interconnection Development and Cooperation Organization, 2020), while our optimisation model is useful to provide more detailed information on the placement and timing of PV and wind-power plants in China.

To clarify the difference in the projected capacities of PV and wind power between our optimisation model and the regional renewable energy development plans (Global Energy Interconnection Development and Cooperation Organization, 2020), we added a new section in lines 274-292:

Comparison with the regional renewable energy development plans

We compared the projected capacities of PV, onshore wind and offshore wind by decade

between our optimisation model and the regional renewable energy development plans according to China's energy transition and 14th Five-year Power planning (FYPP)⁵⁰. The projected increase in the capacities of PV and wind power in China is faster in our optimisation model than for FYPP⁵⁰ (**Fig. S12**). Interestingly, the difference is small by 2030, but increases dramatically from 2030 to 2060, indicating an increasing underestimation of the potential of PV and wind power in the longer term by FYPP⁵⁰. The projected capacities of PV and wind power increase from 1.2 TW in 2020s to 1.7 TW in 2040s by FYPP⁵⁰, compared to an increase from 1.1 TW to 3.6 TW in our optimisation model. This difference is partly due to the fact that the effect of learning¹⁷ is not considered by FYPP⁵⁰. If capacities of PV and wind in our optimisation model are constrained by FYPP⁵⁰, MACC moves leftward and upward relative to our central case, while the maximal potential of emission abatements by PV and wind decrease from 12.6 to 5.8 Gt y⁻¹ in 2060. Spatially, additional PV and wind-power plants in our optimisation model beyond the FYPP projection⁵⁰ are mainly distributed in the Central and North of China, where 57% of power plants are subject to MAC of 0 to 20 USD (t CO₂)⁻¹ (**Fig. S12**). Although exclusion of plants subject to high MAC makes PV and wind power more affordable in FYPP⁵⁰ than our optimisation model, its projection of PV and wind leads to a higher demand of CCS than our projection to achieve the national target of C neutrality⁶.

To show the difference in the deployment of PV and wind power between our optimisation model and the regional renewable energy development plans (Global Energy Interconnection Development and Cooperation Organization, 2020), a new **Fig. S12** was added to the **Supporting Information**:

Fig. S12. Comparison of the projected deployment of PV and wind-power plants by 2060 between the optimisation model and the regional renewable energy development plans in China. (a) Difference in the installed capacities of PV, onshore wind and offshore wind-power plants by decade between China's energy transition and 14th Five-year Power planning (Global Energy Interconnection Development and Cooperation Organization, 2020) and the projection in the central case of our model optimisation. (b) Impact on the marginal abatement cost (MAC) curve by constraining the model optimisation with the projected regional capacities of PV, onshore wind and offshore wind power in the China's energy transition and 14th Five-year Power planning (Global Energy Interconnection Development and Cooperation Organization, 2020) by decade. (c–f) Spatial distributions of PV and wind-power plants in the central case of the model optimisation as a difference to the projection constrained by China's energy transition and 14th Five-year Power planning (Global Energy Interconnection Development and Cooperation Organization, 2020) in terms of the MAC range.

References:

Global Energy Interconnection Development and Cooperation Organization. *Research on China's energy transition and 14th Five-year Power planning* (2020).

Victoria, M., Zhu, K., Brown, T., Andresen, G. B. & Greiner, M. Early decarbonisation of the European energy system pays off. *Nat. Commun.* **11**, 1–9 (2020).

Victoria, M. *et al.* Solar photovoltaics is ready to power a sustainable future. *Joule* **5**, 1041–1056 (2021).

Comment C10

Line 233-238

The study uses the current electric load profile to characterize the future load profile, ignoring very critical factors such as temporal heterogeneity of the added load, demand-side response, and V2G. And these factors can have a very significant impact on the integration of energy storage and renewable energy.

Response

We have carefully revised our model by considering flexible power load on the demand side under different rates of electrification and adopting the optimal option of energy storages between pumped hydro and chemical batteries. These improvements allow us to better represent the demand-side responses to a large-scale deployment of PV and wind power in China (see our response to **Comment A2**). We hoped that these improvements allow our model to better fit the reality by representing the integration of energy storage with deployment of large-scale renewable energy.

Comment C11

Line 250-253

In the study, the average size of power plants reached 14 GW and 12 GW at both 2030 and 2040 time points, and both were distributed only in northwest and northern China (Fig. 1). However, this is quite different from the existing wind farm and PV plant sizes and future development plans. Does the study not consider the actual size and location distribution of existing PV plants or wind farms?

Response

We have examined the impact of increasing the maximal size of each PV and wind-power plant on the calculation of the MAC curves in our optimisation model (see our response to **Comment B10 of Reviewer #2**). We have also compared the timing and placement of the projected new PV and wind-power plants between our optimisation model and previous studies (see our response to **Comment B5**). By comparing the deployment of PV and wind-power plants between our optimisation model and the regional renewable energy development plans (Global Energy Interconnection Development and Cooperation Organization, 2020), we found that the difference in the installed capacities is small in the coming decade, but this difference increases dramatically from 2030 to 2060, indicating that the potential of PV and wind power has been underestimated in the long term by the regional renewable energy development plans without considering the effect of learning (see our response to **Comment C9**). Spatially, the potential PV or wind-power plants in our optimisation model beyond the regional renewable energy development plans are mainly distributed in the Central and North of China, and 57% of these power plants are subject to the MAC between 0 and 20 USD (t CO₂)⁻¹.

We hope that these additional analyses by gathering new data allow our paper to be more policy-relevant.

Reference:

Global Energy Interconnection Development and Cooperation Organization. *Research on China's energy transition and 14th Five-year Power planning* (2020).

Comment C12

Line 269-272

In Figure 3d, except for the scenario where coal generation does not rise for the electricity price of \$0.09, the rebound in coal generation in 2060 after a steady decline until 2050 is very counterintuitive and inconsistent with realistic policies that the authors need to explain. At the same time, the authors assume a constant amount of nuclear, hydro, hydrogen and bioenergy generation in the future, which is also a counterfactual assumption.

Response

We realize that it is unreasonable to estimate the electricity generation based on the exogenous price of electricity (see our response to **Comment A27**). To avoid the exogenous assumption on the prices of electricity, we replaced the three scenarios based on the exogenous prices of electricity with the “no-policy” and “1.5°C-limiting” scenarios based on the projected power generation by oil, gas, bioenergy, nuclear and hydropower derived as the average of three models (SWITCH, IPAC and GCAM_TU) from a recent multi-model study (Duan et al., 2021).

In the new **Fig. 3d**, the rebound of coal generation in 2060 is explained by a rapid increase in the electricity demand due to a rapid electrification of non-power sectors, while the increase in the power generation by renewable energy after 2050 is likely insufficient to meet the increasing electricity demand. It indicates that CCS would be needed to achieve carbon neutrality, but our estimate of PV and wind power reduced the demand of CCS relative to a previous study prescribing MACC from the other regions of the world for China (Duan et al., 2021).

We realize that it is unreasonable to assume that the electricity generation by other renewable energy beyond PV and wind are constant under different scenarios in the future (see our response to **Comment A28**). In our revised paper, we adopted the projection of electricity generation by oil, gas, bioenergy, nuclear and hydropower during 2030–2060 under the

“1.5°C-limiting” scenario to match the national target of carbon neutrality, which is compared to the projection under the “no-policy” scenario (Duan et al., 2021). It allows the electricity generation by other renewable energy to be different during 2030–2060, which can be compared with our projection of electricity generation by PV and wind based on the prices of carbon given in the “no-policy” and “1.5°C-limiting” scenarios (Duan et al., 2021).

To explain the rebound of coal in 2060 and the implications of our higher estimates of PV and wind power than a multi-model study (Duan et al., 2021), we added the following sentences in lines 377-387: *“Under a rapid electrification of non-power sectors, coal power may rebound during 2051–2060, because the projected increase in renewable energy² after 2050 is likely unable to meet the rising total electricity demand⁴¹. In this case, CCS would be needed to achieve C neutrality⁶. As our projected capacities for PV and wind are higher than most of previous estimates (Fig. S1), update of PV and wind power in IAMs² with our estimate reduces the demand of CCS to achieve C neutrality in 2060 from 18.9 to 12.9 PWh y⁻¹ in the “no-policy” scenario and from 10.9 to 2.7 PWh y⁻¹ in the “1.5°C-limiting” scenario, thereby reducing the geological constraints on C neutrality in China⁴⁷.”*

To compare the different paths of deploying renewable energy under the “no-policy” and “1.5°C-limiting” scenarios, a new Fig. 3 was revised:

Fig. 3 | Optimal path to achieve C neutrality by building PV and wind-power plants in China by 2060. (a) Dependence of LCOE of the projected PV and wind-power plants built by 2060 on the share of electricity generation by PV and wind-power plants built in 2020s in total electricity demand in 2060. The optimal path minimizes LCOE of the projected power plants in 2060 by randomly varying the time each plant is built by decade (the shaded area) under a discounting rate (r_d) of 5% y^{-1} in the central case² or 7% y^{-1} in the sensitivity case¹⁵ (red). The insert shows the share of electricity generation by PV and onshore wind and offshore wind-power plants built by decade in total electricity generation in 2060 under the optimal path. (b, c) The (b) time and (c) LCOE of building each PV or wind-power plant in the optimal path. (d) Composition of power generation by decade. Power generation by oil, gas, bioenergy, nuclear and hydropower is derived as the average of three models (SWITCH, IPAC and GCAM_TU) under the “no-policy” and “1.5°C-limiting” scenarios from a multi-model study², while power generation by PV and wind is derived from MACC in this

study under the two scenarios based on the prices of carbon in the models², respectively. Assuming that coal meets the remaining power demand, the demand of CCS (red dot) is estimated to meet the national carbon neutrality. The estimate of PV and wind power in a previous study² and the resultant demand of CCS are shown by the hatched bar and the blue dot, respectively. (e, f) Spatial distributions of (e) PV and (f) wind-power plants constructed by decade under the optimal path. The size and color of the symbol denote the installed capacity and the time to build each power plant. The background shows the global horizontal irradiance (GHI) and wind-power density (WPD).

Reference:

Duan, H. et al. Assessing China's efforts to pursue the 1.5° C warming limit. *Science* **372**, 378–385 (2021).

Comment C13

Line 346-349

It is possible that the value of the \$1 trillion investment by 2040 may be relatively low due to the discount rate setting and investment cost assumptions.

Response

We performed additional experiments to examine the change in initial investments by 2040 required for profitable PV and wind-power plants when increasing the discounting rate from 3 to 7% y^{-1} and using different data of capital costs (Lorenczik et al., 2020; China Electricity Council, 2021). The estimated initial investments were compared with those in our central case, which adopted the data of capital costs by component for PV power using the data published by the China Photovoltaic Industry Alliance (2020) and for onshore wind-power plants using the data compiled by Liu et al. (2012) (see our response to **Comment C6**) under a discounting rate of 5% y^{-1} (see our response to **Comment C5**).

We found that the initial investments for profitable PV and wind-power plants with $MAC < 0$ depend on the discounting rate and capital costs. A lower discounting rate places more weight on the future value of electricity generation and brings the abatement efforts forward (see our response to **Comment C5**). Interestingly, the effect of using a lower discounting rate in bringing abatement efforts forward is more profound when using low capital costs from the data published by China Electricity Council (2021) than using high capital costs by Lorenczik et al. (2020) or capital costs in our central case, because the lower capital costs make PV and wind power more affordable in the near term (**Fig. S26**). As a stimulus required for profitable PV and wind-power plants, the initial investments by 2040 would increase to 2.6 trillion USD using capital costs from the data published by China Electricity Council (2021), higher than the 1.6 trillion USD using the data by Lorenczik et al. (2020) or the 1.8 trillion USD using the data in our central case.

To clarify the sensitivity of initial investments to the discounting rate and capital costs, we added the following sentences in lines 547-554: “*A lower discounting rate brings efforts of abatement forward by placing more weight on the value of future power generation (Fig. S20). This effect becomes more profound when adopting lower capital costs from the data published by China Electricity Council⁵⁵ than the data by Lorenczik et al.³⁹, because lower capital costs make PV and wind power more affordable in the near term (Fig. S26). Initial investments by 2040 required as a stimulus to deploy profitable PV and wind power will reach 1,740 billion USD when using low capital costs⁵⁵ by increasing the number of profitable PV and wind-power plants, compared to 1,407 billion USD by using higher capital costs³⁹.”.*

To show the impact of the discounting rate and capital costs on the initial investments by 2040

as a stimulus for PV and wind-power plants, a new **Fig. S26** was added to the **Supporting Information**:

Fig. S26. The impact of capital costs and the discounting rate on the initial investments for profitable PV and wind-power plants by 2040. The initial investments by 2040 required for PV and wind-power plants with $MAC < 0$ are estimated when varying the discounting rate from 3 to 7% y^{-1} and using different capital costs (Lorenczik et al., 2020; China Electricity Council, 2021; China Photovoltaic Industry Alliance, 2021; Liu et al., 2015b). The capital costs (0.73 and 0.88 USD W^{-1} for PV and onshore wind, respectively) by Lorenczik et al. (2020) (orange line) are higher than the capital costs (0.23 and 0.76 USD W^{-1} for PV and onshore wind, respectively) published by the China Electricity Council (2021) (gray line) or the capital cost by component for PV-power plants (0.64 USD W^{-1} as the total capital costs) using the data published by the China Photovoltaic Industry Alliance (2021) and for onshore wind-power plants (0.68 USD W^{-1} as the total capital costs) using the data by Liu et al. (2015b) to estimate the MAC curves in the central case (green line).

References:

China Electricity Council. *Main indicators of national economy in 2020*. URL: <https://www.cec.org.cn/upload/1/editor/1640595481946.pdf> (2021).

China Photovoltaic Industry Alliance. *China PV industry development roadmap (2020)*. URL: <https://www.ccidgroup.com/info/1105/32621.htm> (2021).

Liu, Z., Zhang, W., Zhao, C. & Yuan, J. The economics of wind power in China and policy implications. *Energ.* **8**, 1529–1546 (2015).

Lorenczik, S. et al. *Projected costs of generating electricity-2020 edition*. (Organisation for Economic Co-Operation and Development, Paris, France, 2020).

Comment C14

Line 473-475

It is slightly conservative that electric energy storage facilities only have a five-year lifetime. From the supplementary information, it is clear that the authors assumed the lifetime of 8,000 cycles, i.e., more than four complete charge and discharge cycles per day, which may not

correspond to the actual battery operation.

Response

We realize that the lifetime of five years is too short for chemical batteries. In our revised paper, we adopted the optimal option of energy storage between mechanical storage (pumped hydro) with a lifetime of 50 years and a round-trip efficiency of 70% (Chen et al., 2021) or chemical storage (batteries) with a shorter lifetime of 15 years but a higher round-trip efficiency of 85% (Cole and Frazier, 2019).

To give the properties of mechanical or chemical storages, a new **Table S9** was added to the **Supporting Information**:

Table S9. Parameters of the pumped-hydro storage and chemical storages.

Type		Pumped-hydro storage	Chemical batteries	Reference(s)
Lifetime (years)		50	15	Cole and Frazier, 2019; Chen et al., 2021
Round-trip efficiency (%)		70%	85%	Cole and Frazier, 2019; Chen et al., 2021
Energy loss rate (%)		0%	1%	Xiong and Singh, 2015
Per unit throughput (kWh kWh⁻¹)		N/A	6000	Chen et al., 2021
Minimum/Maximum residual energy rate (%)		100%/0%	100%/0%	Chen et al., 2021
Energy-specific cost (2019 USD kWh⁻¹)	2020	100	345	Cole and Frazier, 2019; Chen et al., 2021
	2030	100	198	
	2040	100	174	
	2050	100	149	
	2060	100	124	
Power-specific cost (2019 USD kW⁻¹)	2020	1200	595.73	Cole and Frazier, 2019; Hiesl et al., 2020
	2030	1200	374.45	
	2040	1200	327.22	
	2050	1200	280.78	
	2060	1200	234.34	
Optional cost (2019 USD kWh⁻¹)		0.0015	0.0015	Zhang et al., 2016

References:

Chen, X. *et al.* Pathway toward carbon-neutral electrical systems in China by mid-century with negative CO₂ abatement costs informed by high-resolution modeling. *Joule* **5**, 2715–2741 (2021).

Cole, W. J. & Frazier, A. Cost projections for utility-scale battery storage. Report number: NREL/TP-6A20-73222. (National Renewable Energy Lab, Golden, CO, United States, 2019).

Comment C15

Line 525-539

It is unreasonable to calculate the power demand from thermal power by 2060 with the assumption of power sector sharing 80% of national total carbon emissions and as high as 5500 MtCO₂y⁻¹ carbon sinks. The emission factor of 0.827 kg CO₂ kWh⁻¹ for thermal power (including gas-fired power plants) is too high. In addition, why no CCS including BECCS considered for the year 2060 to achieve carbon neutrality?

Response

We realize that it is unreasonable to assume that the power sector takes up a share of 80% of national total CO₂ emissions in 2060. In our revised paper, we estimated the demand of CCS needed to achieve the national carbon neutrality, which depends on terrestrial C sink, rate of electrification, total energy demand and future power generation by PV, wind and other low-carbon energy. We adopted the power generation by oil, gas, bioenergy, nuclear and hydropower as the average of three models (SWITCH, IPAC and GCAM_TU) under the “no-policy” and “1.5°C-limiting” scenarios from a multi-model study (Duan *et al.*, 2021). Coal was used to meet the remaining electricity demand beyond oil, gas, nuclear and renewable energy including PV and wind power. The power generation by PV and wind was estimated from the MACC in our study under the “no-policy” and “1.5°C-limiting” scenarios based on the average prices of carbon in the IAMs (Duan *et al.*, 2021). This calculation of CCS demand allows us to avoid making assumptions on the share of the power sector in total CO₂ emissions.

We realize that terrestrial C sink of 5500 Mt CO₂ y⁻¹ by Wang *et al.* (2020) is too large. In our revised paper, we adopted the estimated terrestrial C sink of 1870 Mt CO₂ y⁻¹ from a bottom-up study (Jiang, 2016), which is recommended by Wang *et al.* (2022).

We also realize that using an emission factor of fossil fuel (0.827 kg CO₂ kWh⁻¹) is unreasonable. In our revised paper, we adopted the CO₂-emission factor for coal (0.84 kg CO₂ kWh⁻¹), oil (0.72 kg CO₂ kWh⁻¹) and gas (0.46 kg CO₂ kWh⁻¹) (Liu *et al.*, 2015, National Bureau of Statistics of China, 2020, Zeiss, 2010). We estimated MAC by assuming that PV and wind power will be used to replace oil, gas and coal in the order of their prices to generate the highest profits.

We estimated the demand of CCS rather than BECCS, because CCS might be deployed when retrofitting coal, gas, oil or biomass-fired power plants, while biomass might be used to produce cellulosic ethanol and Fischer-Tropsch biofuels without CCS (Fajardy *et al.*, 2019).

To clarify our method of estimating the demand of CCS to achieve the national carbon neutrality, we added the following paragraph in the **Method summary** section in lines 174-184: “*Given uncertainties in the geological constraints on CCS⁴⁷, it is interesting to know the demand of CCS needed to achieve China’s C neutrality, which depends on terrestrial C sink (1.87 Gt CO₂ y⁻¹)⁴⁸, rate of electrification (58%)⁴¹, total electricity demand⁴¹ and future power generation by PV, wind and other low-carbon energy². We adopted total power generation by oil, gas, bioenergy, nuclear and hydropower as the average of three IAMs (SWITCH, IPAC and GCAM_TU) under the “no-policy” and “1.5°C-limiting” scenarios from*

a multi-model study². The power generation by PV and wind was estimated according to MACC from this study under the “no-policy” and “1.5°C-limiting” scenarios based on the average prices of carbon in the IAMs². Assuming that coal provides the remaining electricity, we estimated total demand of CCS in electricity systems to meet the national target of C neutrality⁶.”.

To clarify the application of the emission factor by fuel, we added the following sentence in lines 901-905: “We obtained the price of coal (0.043 ± 0.015 USD kWh⁻¹)⁹⁵, oil (0.141 ± 0.057 USD kWh⁻¹)^{96,97} and gas (0.058 ± 0.016 USD kWh⁻¹)⁹⁸ as the 95% confidence interval during 2010–2020 when they were used to generate electricity with an electricity efficiency of 35, 38 and 45%, respectively⁹⁹ and an emission factor of 0.84, 0.72 and 0.46 kg CO₂ kWh⁻¹, respectively⁹³ (Fig. S11).”.

Reference:

Duan, H. et al. Assessing China’s efforts to pursue the 1.5° C warming limit. *Science* **372**, 378–385 (2021).

Jiang, F. et al. A comprehensive estimate of recent carbon sinks in China using both top-down and bottom-up approaches. *Sci. Rep.* **6**, 22130 (2016).

Wang, J. et al. Large Chinese land carbon sink estimated from atmospheric carbon dioxide data. *Nature* **586**, 720–723 (2020).

Wang, Y. et al. The size of the land carbon sink in China. *Nature* **603**, E7–E9 (2022).

Fajardy, M., Koeberle, A., MacDowell, N., & Fantuzzi, A. BECCS deployment: a reality check. *Grantham Institute briefing paper* **28**. URL: <https://www.imperial.ac.uk/media/imperial-college/grantham-institute/public/publications/briefing-papers/BECCS-deployment---a-reality-check.pdf> (2019).

Reviewer Reports on the First Revision:

Referees' comments:

Referee #2 (Remarks to the Author):

As I commented in the previous review, the manuscript main result is constructing carbon emission MACs for solar and wind power in China under different scenarios. I have already commented on the importance of the topic and the excellent execution of the study. In response to the Reviewers' comments, the authors of the manuscript have further improved the technical soundness of their manuscript by additional sensitivity tests and model improvements. The paper uses appropriate data, methods and statistical analysis.

My main comments in the previous review was that the main area of improvement for this manuscript would be to better highlight its novelty and implications as well as its limitations and how these should be addressed in future research. By and large, the manuscript has answered these questions by providing detailed comparisons with other studies (where such studies existed) for every point that I raised. These comparisons are however mostly in the range of providing more accurate estimates rather than in radically different result.

I would still encourage the authors to work on their abstract, framing and conclusions to highlight the significance of their study for a broad audience interested not only in wind and solar but in climate mitigation in general and also frame some of their results in more contrasting terms to other studies.

I also believe the paper can still do a better job for communicating the challenges associated with implementing the modelled scenario in a real life. The authors did a great job by comparing their scenarios with a range in other modeling studies and with China's existing Five-Year-Plans (FYPs). However, the way this comparison is presented should be further improved

1. When the study is compared with other studies (Edenhorer 2011, etc.), it should be explained which (if any) of these studies are forecasts or projections based on realistic trends and which are normative scenarios based on emission and other targets. There is some utility of comparing normative scenarios to each other, but there is more value, in my view, in comparing scenarios with forecasts, plans (e.g. FYPs), and historic trends. Also it would be useful to compare the growth of renewables in studies not in relational (taking the first year as 100%) but in absolute terms (possibly normalised to the size of electricity system or to population).

2. When the study is compared with the FYPs it is an oversimplification to say that FYPs underestimate wind and solar potential. Perhaps they provide different (more nuanced?) estimates based on different assumptions (e.g. on land availability, maximum size of the power plants etc)? Furthermore, it is asserted that the effect of learning is not considered in the FYPs, but does ref [50] really contain this argument.

3. It would be useful to compare the rates of construction of solar and wind power plants envisioned in the optimized model with how much was constructed in China (and perhaps in Europe or in Germany - normalised) in, say, the last 5 years and indicate how much the construction rates have to

be ramped up to be in line with the scenarios.

Referee #4 (Remarks to the Author):

This paper presented the most comprehensive to-date analysis of utility-scale solar and wind (including offshore wind) deployment strategy to achieve carbon neutrality in China, using a spatial and temporal explicit marginal abatement costs approach. The authors also spend enormous efforts to bring pieces together and address previous reviewers' comments. The results offer a new benchmark for policymakers when making renewable development plans, even though those plans are often not based on optimizations.

Below are a few comments and questions for the authors to consider, some might not be relevant or may not be feasible given model limitations or data availability.

The authors acknowledge that "a spatially explicit method has been applied to represent energy systems in Europe[19-21] and USA[22,23]", does this research adds China as a case study, or does it provide insights beyond filling a key region gap? The author might want to highlight those contributions.

The paper is very ambitious to try to bring every piece together from resources assessment, system optimization, and environmental and social-economic impacts, this is on one side good for a paper, and on the other side, risky for losing the main messages. Readers might be overwhelmed by the scope of the study and the authors should give a high-level summary storyline on the key messages and findings of the study.

Questions on the model: the study calculated the MAC of each plant and optimize the time of installation based on the MAC, the paper also optimize the revenue (might double counting by adding and replacing fossil fuel and selling renewable electricity), are these two optimizations combined in the optimization decision or one after another?

I understand the paper focuses on utility-scale solar, onshore, and offshore wind; however, it is not clear how the paper assumes the role of nuclear, CCS, pumped hydro, hydrogen, and other technologies.

It is not clear if the paper model the capacity expansion every 1/5/10 years or simply optimizes one-year in 2060? It seems the model assumes no new UHV transmission lines will be built. Not sure how the model deals with transmission cost if this is the case as those already built are sunk costs. It will help if the paper can show the hourly load, and how the consideration of storage, transport, etc., changes the load.

The paper has done a great job of presenting and visualizing the sensitivity of key parameters to the results. However, the system is hardly impacted by one single parameter, land cost, investment costs, discounting rate, variable load, transmission, and carbon costs could change simultaneously, not sure how the paper address such cases.

Other minor notes:

In SI:

Ln 154: Needs better justification why the GE2.5MW is used, is it the most popular wind turbine used in China?

Ln221: Do you assume no news UHV lines will be built after 2025? Can the model simulate transmission capacity expansion?

Ln273: Are costs endogenous with an installed capacity based on learning rates? Pay attention to the learning rates cited are mostly international studies, and rates in China are usually higher than international rates.

Ln578: It seems offshore only gets built after 2040? Does it because of the assumption of costs? Incorporating offshore wind plans might help to address this issue.

Author Rebuttals to First Revision:

Referee #2

Comment B1

As I commented in the previous review, the manuscript main result is constructing carbon emission MACs for solar and wind power in China under different scenarios. I have already commented on the importance of the topic and the excellent execution of the study. In response to the Reviewers' comments, the authors of the manuscript have further improved the technical soundness of their manuscript by additional sensitivity tests and model improvements. The paper uses appropriate data, methods and statistical analysis.

Response

We are happy to receive these positive comments. No specific response is required.

Comment B2

My main comments in the previous review were that the main area of improvement for this manuscript would be to better highlight its novelty and implications as well as its limitations and how these should be addressed in future research. By and large, the manuscript has answered these questions by providing detailed comparisons with other studies (where such studies existed) for every point that I raised. These comparisons are however mostly in the range of providing more accurate estimates rather than in radically different result.

Response

Following the Reviewer's suggestion, we have revised our paper by focusing on these results that are radically different from previous studies, on **how to accelerate the growth of PV and wind power to meet the power demand under C neutrality by using a spatially explicit approach**. By improving the optimisation model for PV and wind power, we address **the gap between on the one hand, the low capacity of power generation based on previous forecasts of renewable energy or historical trends (5–9.5 PWh y⁻¹), and on the other hand, the high demand for PV and wind-power generation when meeting the target of C neutrality (10–15 PWh y⁻¹), which could provide general insights into climate mitigation**.

The purpose of our paper is now clarified at the beginning of the **Abstract**: *“Pledges to achieve carbon (C) neutrality in China represent the need to scale up photovoltaic (PV) and wind-power generation from 1 to 10–15 PWh y⁻¹ by 2060 (refs.¹⁻⁵). Following the historical rate of renewable installation¹, energy modeling⁶ or forecasts that inform the China's 14th Five-year Energy Development (CFED) plans⁷, however, only indicated an increase in PV and wind-power generation to 5–9.5 PWh y⁻¹ by 2060. Here we propose to increase the potential of PV and wind power by optimizing the location, capacity and construction time of 3,844 utility-scale PV and wind-power plants, which are coordinated with ultra-high-voltage transmission and energy storage by accounting for the flexibility of power loads and intertemporal dynamics of learning.”*, and highlighted at the beginning of Introduction in line 72-82: *“Under the projected growth of various renewables in economic models, achieving C neutrality requires scaling up China's PV and wind-power generation from 1 PWh y⁻¹ in 2020 to 10–15 PWh y⁻¹ in 2060 (e.g. refs.¹⁻⁵). The projected PV and wind-power generation in 2060, however, will only increase to 5 PWh y⁻¹ if we extrapolate an annual rate of growth by 100 TWh y⁻² during 2010–2020 (ref.¹), to 9 PWh y⁻¹ if we assume an annual rate of growth by 200 TWh y⁻² based on the forecasts in the governmental plans of China's 14th Five-year Energy Development (CFED)⁷, or to 9.5 PWh y⁻¹ based on a projection from a regional comprehensive modeling of renewable energy⁶. Despite the rapid growth of PV and wind power in China in the last decade¹, the rate may slow in the coming decades due to a*

reduction of governmental subsidies²⁵, a lack of new transmission infrastructure⁶ and restrictions for protecting agricultural, industrial and urban lands²⁶.”.

The key findings of our study are summarized in the **Abstract**: *“Our optimisation model ramps up annual investments in PV and wind power from 72 billion US dollars (USD) in 2021 to 250 billion USD during 2021–2060 to accelerate the transition from fossil fuels to PV and wind power. In contrast to the CFED path⁷, our optimal path increases the generation of PV and wind power from 9 to 15 PWh y⁻¹ in 2060, with a reduction of the average marginal abatement cost from 98 (ref.¹) to 6 (-1 to 12 as 90% uncertainty based on our Monte Carlo simulations) USD (t CO₂)⁻¹ by optimizing the power systems to minimize costs. The development of PV and wind power in China will reduce income inequality by creating new income opportunities for residents in the poorest regions. Our results reinforce the importance of trend-breaking investments for upgrading power systems to accelerate the development of PV and wind power at affordable costs to achieve C neutrality in China.”.*

The Title of our paper is revised as: *“Achieving carbon neutrality in China by 2060 requires trend-breaking investments in photovoltaic and wind power”.*

Our paper is re-framed to focus on this storyline. Please see our detailed responses to all your comments below.

Comment B3

I would still encourage the authors to work on their abstract, framing and conclusions to highlight the significance of their study for a broad audience interested not only in wind and solar but in climate mitigation in general and also frame some of their results in more contrasting terms to other studies.

Response

Our paper aims at providing general insights for a broad audience interested **not only in wind and solar power but also in climate mitigation**, and **not only for China but also for many other countries with rich PV and wind resources but limited growth of PV and wind power in the last decade**.

To provide general insights for a broad audience, we highlight the scope of our paper at the beginning of our paper in line 45-62: *“High ambitions from all nations are needed to achieve the goal of 2°C warming in the Paris Agreement⁸⁻¹⁰, which requires to shift massive investments from fossil fuels to low-carbon energy by meeting the target of carbon (C) neutrality¹¹. Accelerating the penetration of renewables in power generation is a key pillar for global and regional strategies of energy transitions^{2,12,13}. Global decarbonization of power systems, however, may not proceed as fast as it should be in the coming decades, because the financial support of fossil fuels in 51 countries, representing 85% of the global energy supply, almost doubled in 2021 and will likely continue to increase for years¹⁴. The penetration of renewable energy cannot accelerate unless the balance between technological advances and countervailing forces is radically shifted by overcoming barriers such as technical feasibility, costs and land limitation¹⁵. The latest evidence indicated that the world is probably on track toward 2.8°C warming under the current policies or 2.6°C warming based on the 2030 Nationally Determined Contributions (NDCs) due to limited progress in phasing out fossil fuels¹⁶. The cover decision of the 27th United Nations Climate Change Conference, known as the Sharm el-Sheikh Implementation Plan, highlighted that a global transformation to low-carbon economies required annual investments of at least 4–6 trillion USD during 2030–2050 (ref.¹⁷). However, the distribution of these funds toward renewable energy development remains unclear^{6,17}, which requires advanced grid modeling approaches to determine the timing of investments by optimally scheduling the operation of energy*

systems with spatial details and coordinating infrastructure⁸”.

To provide general insights into climate mitigation, we summarize our key findings in the last paragraph in line 415-445: *“Our approach improves the optimisation of power systems^{2,6,30,31,35,36} by applying a spatially explicit method in China. Moreover, our results provide insights into climate mitigation for other countries with rich resources of PV and wind power, where growth of them has been limited in the last decade¹⁵. First, we find that the MAC will increase when scaling up PV and wind-power generation to meet the target of C neutrality, even by considering the benefits of learning^{38,48}. This behavior of MAC challenges a widespread view that the costs of renewable energy will decrease as its deployment increases to reach even net negative costs of energy transitions⁶⁷. Our study clarifies the impact of physical limitations (e.g. demand for land and planning infrastructure) when expanding the capacity of renewable energy⁴⁷, demonstrates the importance of policy interventions (e.g. supporting grid integration⁶ and upgrading power systems at the demand side⁴²), and highlights the need to ramp up initial investments to overcome techno-economic barriers. China is among a few countries with an accelerated growth of PV and wind power in the last decade¹⁵, but such growth may decelerate due to the constraints of electricity transmission and energy storage and the restrictions of land availability. Overlooking these limiting factors will lead to an overestimate of the potential of climate mitigation by PV and wind power (e.g. refs^{2,12,13}). Second, our projection of PV and wind power in Northwest and North China provides opportunities to reduce the costs of decarbonization in undeveloped regions with vast areas of desert and marginal land, but both areas will require high initial investments for upgrading the power systems. Working in this direction requires the development of high-resolution models of energy systems⁶⁸, which can be combined with a real-time analysis of the availability of renewable energy^{23,27,66} and a dynamic simulation of regional financial systems. Third, our optimisation model highlights the importance of continuous investments in the coming decades, which will be essential to accelerate the decline of module prices and upgrade of the power systems. Our optimal path entails costs of developing PV and wind power higher than current levels⁴⁶, but lower costs in the 2040s and 2050s than the national plans for renewable energy development^{1,7}. Over-reliance on future ambitious mitigation is a dangerous distraction to defer mitigation efforts today⁶⁹, thus increasing the risk of exceeding climatic tipping points⁷⁰. This study improves our understanding of the techno-economic limitations in developing key renewable energy in China, clarifies the importance of trend-breaking investments in the coming decades for climate mitigation, and addresses the potential and costs to achieve C neutrality in the long term.”.*

We have improved the comparison in our paper by focusing on results that are radically different from previous studies. Please see our detailed responses to your comments below.

Comment B4

I also believe the paper can still do a better job for communicating the challenges associated with implementing the modelled scenario in a real life. The authors did a great job by comparing their scenarios with a range in other modeling studies and with China’s existing Five-Year-Plans (FYPs). However, the way this comparison is presented should be further improved.

Response

Following the Reviewer’s suggestion, we focus on comparing the potential and costs of PV and wind-power generation between our optimal path and the “CFED” path adopting the rate of installing renewables in China’s 14th Five-year Energy Development (CFED) plans (Center for Security and Emerging Technology, 2021), and comparing the projected marginal

abatement cost (MAC) for PV and wind power between our optimal path and a previous estimate under C neutrality in 2060 (Goldman Sachs Research, 2021).

We show the differences in the temporal variation of investments in PV and wind power (Fig. 1e,f), the potential and costs of climate mitigation by developing PV and wind power (Fig. 2a), and the effects of ramping up investments in PV and wind power on the total costs of achieving C neutrality (Fig. 3) between our optimal path and the CFED path. By using a spatially explicit approach, our study **highlights unrecognized aspects of renewable energy** that power systems could be optimized to reduce the costs of PV and wind power and increase the capacity of PV and wind power based on the optimizing procedures considered in our model (Fig. 2b,c). Please see our detailed responses to your comments below.

References:

Center for Security and Emerging Technology. Translation: Outline of the People's Republic of China 14th Five-Year Plan for National Economic and Social Development and Long-Range Objectives for 2035. Available at: <https://cset.georgetown.edu/publication/china-14th-five-year-plan/> (2021).

Goldman Sachs Research. Carbonomics: China net zero—The clean tech revolution. Available at: <https://wwwqa.goldmansachs.com/insights/pages/gs-research/carbonomics-china-netzero/report.pdf> (2021).

Comment B5

1. When the study is compared with other studies (Edenhorer 2011, etc.), it should be explained which (if any) of these studies are forecasts or projections based on realistic trends and which are normative scenarios based on emission and other targets. There is some utility of comparing normative scenarios to each other, but there is more value, in my view, in comparing scenarios with forecasts, plans (e.g. FYPs), and historic trends. Also it would be useful to compare the growth of renewables in studies not in relational (taking the first year as 100%) but in absolute terms (possibly normalised to the size of electricity system or to population).

Response

Following the Reviewer's suggestion, we distinguish the projections of PV and wind power based on historical trends, forecast data, and a previous modeling of energy systems using the absolute terms (given by the annual PV and wind-power generation). The differences among these projections are now clarified at the beginning of our paper in line 72-82: "*Under the projected growth of various renewables in economic models, achieving C neutrality requires scaling up China's PV and wind-power generation from 1 PWh y^{-1} in 2020 to 10–15 PWh y^{-1} in 2060 (e.g. refs.¹⁻⁵). The projected PV and wind-power generation in 2060, however, will only increase to 5 PWh y^{-1} if we extrapolate an annual rate of growth by 100 TWh y^{-2} during 2010–2020 (ref.¹), to 9 PWh y^{-1} if we assume an annual rate of growth by 200 TWh y^{-2} based on the forecasts in the governmental plans of China's 14th Five-year Energy Development (CFED)⁷, or to 9.5 PWh y^{-1} based on a projection from a regional comprehensive modeling of renewable energy⁶. Despite the rapid growth of PV and wind power in China in the last decade¹, the rate may slow in the coming decades due to a reduction of governmental subsidies²⁵, a lack of new transmission infrastructure⁶ and restrictions for protecting agricultural, industrial and urban lands²⁶.*"

The PV and wind-power generation based on the forecast data mentioned above by maintaining the rate of installing renewables in China's 14th Five-year Energy Development

(CFED) plans (9 PWh y^{-1}) is close to the estimate from a normative scenario based on a comprehensive modeling of regional energy systems (9.5 PWh y^{-1}), but higher than what is achieved based on historical trends (5 PWh y^{-1}). Following the Reviewer's suggestion, we focus on comparing the temporal variation of investments in PV and wind power (Fig. 1e,f), the potential and costs of climate mitigation by developing PV and wind power (Fig. 2a), and the effects of ramping up investments in PV and wind power on the total costs of achieving C neutrality (Fig. 3) between our optimal path and the CFED path.

Our results are radically different from the CFED path by 1) projecting higher generation of PV and wind power through optimizing the power systems, 2) projecting lower costs of PV and wind power in the 2040s and 2050s by accelerating the price declines of PV and wind power, and 3) producing larger benefits of PV and wind power generation to poverty alleviation by reducing the total costs. These new aspects recognized in our study are now highlighted in the Abstract: “*Our optimisation model ramps up annual investments in PV and wind power from 72 billion US dollars (USD) in 2021 to 250 billion USD during 2021–2060 to accelerate the transition from fossil fuels to PV and wind power. In contrast to the CFED path⁷, our optimal path increases the generation of PV and wind power from 9 to 15 PWh y^{-1} in 2060, with a reduction of the average marginal abatement cost from 98 (ref.¹) to 6 (-1 to 12 as 90% uncertainty based on our Monte Carlo simulations) USD (t CO₂)⁻¹ by optimizing the power systems to minimize costs. The development of PV and wind power in China will reduce income inequality by creating new income opportunities for residents in the poorest regions. Our results reinforce the importance of trend-breaking investments for upgrading power systems to accelerate the development of PV and wind power at affordable costs to achieve C neutrality in China.*”.

The new Fig. 1 shows the larger potential of PV and wind-power generation in our optimal path than the CFED path:

Fig. 1 | Optimisation of the placement, capacity and construction time of utility-scale photovoltaic (PV) and wind-power plants during 2021–2060 in China. (a, b) Maps of PV (a) and wind (b) power plants built by decade in the optimal path. The background shows global horizontal irradiance (GHI) and wind-power density (WPD). (c) Seasonal and diurnal variations in the generation and demand of power under an electrification rate of 58% for non-power sectors in 2060. The shading represents the PV and wind-power generation without considering curtailments. Relative to a baseline scenario limiting the capacity of power plants to 10 GW, we design five experiments by sequentially increasing the limit of power capacity from 10 to 100 GW (Case A), building new UHV lines after 2025 (Case B), storing energy (Case C), improving the electrification of non-power sectors (Case D), considering the flexibility of hourly power loads (Case E), and optimizing the dynamics of learning (optimal path). (d) Power-use efficiency defined as the fraction of the generated power consumed by end-users. (e) Influences of increasing the capacity of PV and wind-power generation in the 2020s on the levelized cost of electricity (LCOE) of all new PV

and wind-power plants built during 2021–2060. The optimal path minimizes LCOE by varying the construction time of each power plant by decade (the shaded area) under a discounting rate of 5% y^{-1} . The insert shows the annual costs by decade. We consider a “CFED” path by adopting the rate of installing renewables in China’s 14th Five-year Energy Development (CFED)⁷ and the projected costs of PV and wind power¹. (f) Dependencies of the annual costs of developing PV and wind power during 2021–2060 on the demand for PV and wind-power generation in 2060 under different scenarios.

The new Fig. 2 shows the lower costs of climate mitigation by PV and wind power in our optimal path than the CFED path:

Fig. 2 | Potential and costs of climate mitigation by PV and wind power in China. (a) Marginal abatement cost (MAC) for coordinating PV and wind-power generation with ultra-high-voltage (UHV) transmission, energy storage, electrification, flexibility of power loads, and learning dynamics under a discounting rate of 5% y⁻¹ in 2060. We design five experiments by sequentially increasing the limit of power capacity from 10 to 100 GW (Case A), building new UHV lines after 2025 (Case B), developing energy storage (Case C), improving the electrification of non-power sectors (Case D), considering the flexibility of power loads (Case E), and optimizing the dynamics of learning (optimal path) relative to a baseline scenario limiting the capacity of power plant to 10 GW. We consider a “CFED” path maintaining the rate of installing renewables in China’s 14th Five-year Energy Development (CFED)⁷ and the projected costs of PV and wind power¹. Arrows represent the MACs for hydropower, hydrogen energy, carbon capture utilization and storage (CCUS), and direct air carbon capture and storage (DACCS)¹. (b) Impacts of optimizing procedures on the potential of CO₂ emissions abatement based on the range of MAC. (c) Violin plots of the average MAC when building new PV and wind-power plants to meet the power demand in 2060. We perform sensitivity experiments by applying the international learning rates (Table S1) (I), adopting low⁴⁹ (II) or high³⁷ (III) capital costs, ignoring construction costs of new UHV lines (IV), adopting a high discounting rate (7% y⁻¹)⁶ (V), and assuming a short lifetime (20 years) of power plants³⁸ (VI) relative to the optimal path. (d) Composition of the total costs when increasing PV and wind-power generation from 5 to 10 PWh y⁻¹ in 2060.

The new Fig. 3 shows the different impacts of ramping up investments on the total costs of PV, wind and CCS under C neutrality between our optimal path and the CFED path:

Fig. 3 | Different paths to achieve C neutrality in China by 2060. (a) Composition of the generated power by decade. Total power demand and the projected generation of power by oil, gas, bioenergy, nuclear and hydropower are prescribed from the “1.5°C-limiting” scenario in a multi-model study². The projected PV and wind-power generation is derived from the optimal or CFED⁷ path. Assuming that coal meets the remaining power demand, we estimate the demand for CCS installed with either fossil fuels or biomass when achieving C neutrality in 2060. (b) Dependence of the annual costs of PV, wind and CCS under C neutrality in 2060 on the costs of transmitting electricity by UHV lines during 2021–2060 in the optimal path from this study. (c) Dependence of the annual costs of PV, wind and CCS under C neutrality in 2060 on the costs of PV and wind power during 2021–2060 in the optimal (dashed lines) or CFED⁷ (solid lines) path. In (b), the total capacity of PV and wind power built during 2021–2060 in the optimal path from this study depends on the capacity of electricity transmission, while the total length of new UHV lines is indicated by the color of the circles. Differently, the total capacity of PV and wind power in 2060 depends on the rate of growth of PV and wind-power generation during 2021–2060 in the CFED⁷ path, which is indicated by the color of the circles in (c). We predict the costs of CCS based on the marginal abatement cost of CCS¹ and the demand for CCS to achieve C neutrality in 2060 under different scenarios.

The new Fig. 4d shows the impact of developing PV and wind power on the Gini coefficient in China, which is compared among different scenarios using our model:

Fig. 4 | Impacts of developing PV and wind power on alleviating poverty in China. (a) Revenue from PV and wind-power generation in 2060 under different carbon prices. (b) Change in the distribution of per capita income when the carbon price increases from 0 to 100 USD (t CO₂)⁻¹. (c) Change in the income of the 10% poorest people in 2060 due to the development of PV and wind power during 2021–2060 when adopting a carbon price of 100 USD (t CO₂)⁻¹. (d) Change in the Gini coefficient when the carbon price increases from 0 to 100 USD (t CO₂)⁻¹. The shaded area represents 90% uncertainty in Monte Carlo simulations. (e) Flow of finances embodied in the transmission of electricity generated by new PV and wind-power plants built during 2021–2060 under a carbon price of 100 USD (t CO₂)⁻¹. The inserts show the changes in per capita income by region when the carbon price increases from 0 to 100 USD (t CO₂)⁻¹ in 2060.

Comment B6

2. When the study is compared with the FYPs, it is an oversimplification to say that FYPs underestimate wind and solar potential. Perhaps they provide different (more nuanced?) estimates based on different assumptions (e.g. on land availability, maximum size of the power plants etc)? Furthermore, it is asserted that the effect of learning is not considered in the FYPs, but does ref [50] really contain this argument.

Response

Following **Comment B4**, we now focus on comparing our optimal path with the CFED path adopting the rate of installing renewables in China's 14th Five-year Energy Development (CFED) plans (Center for Security and Emerging Technology, 2021) and the projected marginal abatement cost (MAC) for PV and wind power under C neutrality in 2060 (Goldman Sachs Research, 2021).

We agree with the Reviewer that the difference in results is caused by different assumptions in different models. However, we do not find information on how these previous studies (Goldman Sachs Research, 2021; Center for Security and Emerging Technology, 2021) modeled the intertemporal dynamics of learning and their original codes are not available to

us, so we cannot comment on the effects of learning in their estimates. When I was reviewing another recent study (Chen et al., 2021), I had confirmed with the authors that learning is not explicitly modeled in that study. We have released our code and data used to estimate MACs in our study on a public website (<https://github.com/rongwang-fudan/PV-Wind-China>), so it should allow other researchers to fully repeat our modeling procedures.

Although it is a challenge to fully compare two models which are developed by different groups using different methods, we explain the differences between our optimisation model and these forecast models (Center for Security and Emerging Technology, 2021; Goldman Sachs Research, 2021) by **carefully analyzing the impact of each assumption (e.g. learning) made in our optimisation model**. To do this, we design five experiments by sequentially increasing the limit of power capacity from 10 to 100 GW (Case A), constructing new UHV lines after 2025 (Case B), developing the optimal energy storage systems (Case C), improving electrification of non-power sectors (Case D), considering the flexibility of power load (Case E), and optimizing the time of building each power plant (optimal path). We show the impact of each optimizing procedure on the generation of PV and wind power and the required investments (Fig. 1f), on the potential and costs of climate mitigation by PV and wind power (Fig. 2b,c), and on the Gini coefficient under different carbon costs (Fig. 4d).

We highlight the differences in the ramping up of investments in PV and wind from different studies in line 155-162: *“Annual costs in our optimal path are similar in the 2020s and 2030s to a “CFED” path adopting the absolute rate of installing renewables in the national energy development plans⁷ and the projected costs of PV and wind power¹, but are much lower in the 2040s and 2050s. The annual costs of PV and wind power in our optimisation model in the period 2031–2050 (220 billion USD y⁻¹) are **much lower than a previous estimate⁶ (320 billion USD y⁻¹) based on a regional comprehensive modeling of energy systems when achieving 80% penetration of renewables as this study⁶ did not explicitly account for the effects of learning dynamics.**”*

We highlight the impact of each assumption (e.g. learning) made in our optimisation model in line 163-179: *“In contrast to the CFED path^{1,7}, our optimisation model predicts lower costs to accelerate the growth of PV and wind power (Fig. 1e). We analyzed the contribution of each optimization procedure to the potential and costs of PV and wind power by comparing the scenario in our optimal path with a baseline scenario limiting the capacity of PV and wind-power plants to 10 GW without transmitting electricity and storing energy. We designed five experiments by sequentially increasing the limit of power capacity from 10 to 100 GW (Case A), allowing the construction of new UHV lines after 2025 (Case B), adding energy storage capacities (Case C), improving the electrification of non-power sectors (Case D), and considering the flexibility of hourly power loads (Case E). Case E would become equivalent to our optimal path if the intertemporal dynamics of learning is further optimized. In contrast to the baseline scenario, our optimal path expands the potential of PV and wind-power generation by increasing the limit of power capacity (+1.1 PWh y⁻¹ for generating power and +126 billion USD y⁻¹ for annual costs), building new UHV lines (+2.2 PWh y⁻¹ and +36 billion USD y⁻¹), developing energy storage (+6.4 PWh y⁻¹ and +114 billion USD y⁻¹), improving electrification (+0 PWh y⁻¹ but -58 billion USD y⁻¹), considering the flexibility of hourly power loads (+0 PWh y⁻¹ but -30 billion USD y⁻¹), and optimizing the intertemporal dynamics of learning (+0 PWh y⁻¹ but -115 billion USD y⁻¹) (Fig. 1f).”*

References:

Center for Security and Emerging Technology. Translation: Outline of the People’s Republic of China 14th Five-Year Plan for National Economic and Social Development and Long-Range Objectives for 2035. Available at:

<https://cset.georgetown.edu/publication/china-14th-five-year-plan/> (2021).

Chen, X. *et al.* Pathway toward carbon-neutral electrical systems in China by mid-century with negative CO₂ abatement costs informed by high-resolution modeling. *Joule* **5**, 2715–2741 (2021).

Goldman Sachs Research. Carbonomics: China net zero—The clean tech revolution. Available at: <https://wwwqa.goldmansachs.com/insights/pages/gs-research/carbonomics-china-netzero/report.pdf> (2021).

Comment B7

3. It would be useful to compare the rates of construction of solar and wind power plants envisioned in the optimized model with how much was constructed in China (and perhaps in Europe or in Germany - normalised) in, say, the last 5 years and indicate how much the construction rates have to be ramped up to be in line with the scenarios.

Response

We have compared the projected growth of PV and wind power in our optimal path with the rate of expansion in the last decade from 2011 to 2020 (rather than five years to represent a decadal trend) in China and Europe. We notice that the relative rate of growth in these two regions was high enough in the last decade, but **the absolute rate of growth should be increased by ramping up investments to achieve C neutrality in China by 2060** (see our response to **Comment B5** above).

The needs to increase the absolute rate of growth of PV and wind power are now highlighted in line 311-336: *“We found that a shift of PV and wind power projected from the CFED forecast⁷ to our optimal path reduces the demand for CCS in the generation of power from 8.9 to 2.8 PWh y⁻¹ in 2060 (Fig. 3a). Our optimal path increases PV and wind-power generation from 1.0 to 15.3 PWh y⁻¹ during 2021–2060 at relative rates of 10.7, 7.4, 5.6 and 4.7% y⁻¹ in the 2020s, 2030s, 2040s and 2050s, respectively. Annual PV and wind-power generation during 2011–2020 had expanded from 0.22 to 1.0 PWh y⁻¹ at a relative rate of 18% y⁻¹ in China¹, and from 0.21 to 0.55 PWh y⁻¹ at a relative rate of 11% y⁻¹ in Europe⁶². Despite these high relative rates of growth, we highlight the importance of ramping up investments over time (Fig. 1e) and upgrading power systems (Fig. 2b) to accelerate the absolute growth of PV and wind power and the phasing out of fossil fuels in China. Previous studies of modeling of renewable energy^{2,6,28-31,35,36} did not simultaneously optimize the spatial deployment and timing of renewable energy development. For example, the growth of capacity of PV and wind power does not depend on investments in electricity transmission in the CFED path⁷ (Fig. 3c). In contrast, our model considers that the growth of capacity of PV and wind power during 2021–2060 depends on investments in expanding the capacity of existing UHV transmission lines (Fig. 3b). Recognizing such dependency will help us to represent the techno-economic barriers in developing PV and wind power in China. For example, when annual costs of PV and wind power during 2021–2060 increase from 0 to 60 billion USD y⁻¹, the ratio of the saved total costs of PV, wind and CCS under C neutrality to the investments in PV and wind power is 2.7 in our optimal path, lower than the value of 6.1 if we followed the CFED path⁷. This ratio increases to 5.5 in our optimal path, but decreases to 0.4 in the CFED path⁷, when further increasing annual costs from 60 to 250 billion USD y⁻¹. Our optimisation model suggests that current investments in PV and wind power (72 billion USD y⁻¹ in 2021)⁴⁶ should be ramped up in the coming decades to accelerate the growth of PV and wind power in China to reduce the total costs of achieving C neutrality by 2060.”.*

Comment D1

This paper presented the most comprehensive to-date analysis of utility-scale solar and wind (including offshore wind) deployment strategy to achieve carbon neutrality in China, using a spatial and temporal explicit marginal abatement costs approach. The authors also spend enormous efforts to bring pieces together and address previous reviewers' comments. The results offer a new benchmark for policymakers when making renewable development plans, even though those plans are often not based on optimizations.

Response

We are happy by receiving these positive comments. No specific response is required.

Comment D2

Below are a few comments and questions for the authors to consider, some might not be relevant or may not be feasible given model limitations or data availability.

Response

Thank you very much for these high-level comments and detailed suggestions. We tried our best to address them by re-framing our paper carefully and answering your questions below.

Comment D3

The authors acknowledge that “a spatially explicit method has been applied to represent energy systems in Europe[19-21] and USA[22,23]”, does this research adds China as a case study, or does it provide insights beyond filling a key region gap? The author might want to highlight those contributions.

Response

Our revised paper aims at providing general insights in climate mitigation beyond filling a key regional gap.

We clarify the purpose of our paper at the beginning of the Introduction in line 45-62: “*High ambitions from all nations are needed to achieve the goal of 2°C warming in the Paris Agreement⁸⁻¹⁰, which requires to shift massive investments from fossil fuels to low-carbon energy by meeting the target of carbon (C) neutrality¹¹. Accelerating the penetration of renewables in power generation is a key pillar for global and regional strategies of energy transitions^{2,12,13}. Global decarbonization of power systems, however, may not proceed as fast as it should be in the coming decades, because the financial support of fossil fuels in 51 countries, representing 85% of the global energy supply, almost doubled in 2021 and will likely continue to increase for years¹⁴. The penetration of renewable energy cannot accelerate unless the balance between technological advances and countervailing forces is radically shifted by overcoming barriers such as technical feasibility, costs and land limitation¹⁵. The latest evidence indicated that the world is probably on track toward 2.8°C warming under the current policies or 2.6°C warming based on the 2030 Nationally Determined Contributions (NDCs) due to limited progress in phasing out fossil fuels¹⁶. The cover decision of the 27th United Nations Climate Change Conference, known as the Sharm el-Sheikh Implementation Plan, highlighted that a global transformation to low-carbon economies required annual investments of at least 4–6 trillion USD during 2030–2050 (ref.¹⁷). However, the distribution of these funds toward renewable energy development remains unclear^{6,17}, which requires advanced grid modeling approaches to determine the timing of investments by optimally scheduling the operation of energy systems with spatial details and coordinating*

*infrastructure*⁸”.

We highlight the implications of our results for a broad audience in the last paragraph of our paper in line 415-445: “*Our approach improves the optimisation of power systems^{2,6,30,31,35,36} by applying a spatially explicit method in China. Moreover, our results provide insights into climate mitigation for other countries with rich resources of PV and wind power, where growth of them has been limited in the last decade¹⁵. First, we find that the MAC will increase when scaling up PV and wind-power generation to meet the target of C neutrality, even by considering the benefits of learning^{38,48}. This behavior of MAC challenges a widespread view that the costs of renewable energy will decrease as its deployment increases to reach even net negative costs of energy transitions⁶⁷. Our study clarifies the impact of physical limitations (e.g. demand for land and planning infrastructure) when expanding the capacity of renewable energy⁴⁷, demonstrates the importance of policy interventions (e.g. supporting grid integration⁶ and upgrading power systems at the demand side⁴²), and highlights the need to ramp up initial investments to overcome techno-economic barriers. China is among a few countries with an accelerated growth of PV and wind power in the last decade¹⁵, but such growth may decelerate due to the constraints of electricity transmission and energy storage and the restrictions of land availability. Overlooking these limiting factors will lead to an overestimate of the potential of climate mitigation by PV and wind power (e.g. refs^{2,12,13}). Second, our projection of PV and wind power in Northwest and North China provides opportunities to reduce the costs of decarbonization in undeveloped regions with vast areas of desert and marginal land, but both areas will require high initial investments for upgrading the power systems. Working in this direction requires the development of high-resolution models of energy systems⁶⁸, which can be combined with a real-time analysis of the availability of renewable energy^{23,27,66} and a dynamic simulation of regional financial systems. Third, our optimisation model highlights the importance of continuous investments in the coming decades, which will be essential to accelerate the decline of module prices and upgrade of the power systems. Our optimal path entails costs of developing PV and wind power higher than current levels⁴⁶, but lower costs in the 2040s and 2050s than the national plans for renewable energy development^{1,7}. Over-reliance on future ambitious mitigation is a dangerous distraction to defer mitigation efforts today⁶⁹, thus increasing the risk of exceeding climatic tipping points⁷⁰. This study improves our understanding of the techno-economic limitations in developing key renewable energy in China, clarifies the importance of trend-breaking investments in the coming decades for climate mitigation, and addresses the potential and costs to achieve C neutrality in the long term.*”.

We highlight the key finding of our paper in the **Abstract**: “*Our optimisation model ramps up annual investments in PV and wind power from 72 billion US dollars (USD) in 2021 to 250 billion USD during 2021–2060 to accelerate the transition from fossil fuels to PV and wind power. In contrast to the CFED path⁷, our optimal path increases the generation of PV and wind power from 9 to 15 PWh y⁻¹ in 2060, with a reduction of the average marginal abatement cost from 98 (ref.¹) to 6 (-1 to 12 as 90% uncertainty based on our Monte Carlo simulations) USD (t CO₂)⁻¹ by optimizing the power systems to minimize costs. The development of PV and wind power in China will reduce income inequality by creating new income opportunities for residents in the poorest regions. Our results reinforce the importance of trend-breaking investments for upgrading power systems to accelerate the development of PV and wind power at affordable costs to achieve C neutrality in China.*”.

Comment D4

The paper is very ambitious to try to bring every piece together from resources assessment, system optimization, and environmental and social-economic impacts, this is on one side good for a paper, and on the other side, risky for losing the main messages. Readers might be

overwhelmed by the scope of the study and the authors should give a high-level summary storyline on the key messages and findings of the study.

Response

Following the Reviewer's suggestion, our revised paper now focus on one key storyline, on **how to accelerate the growth of PV and wind power to meet the power demand under C neutrality by using a spatially explicit approach**. By improving the optimisation model for PV and wind power, we address **the gap between on the one hand, the low capacity of power generation based on previous forecasts of renewable energy or historical trends (5–9.5 PWh y^{-1}), and on the other hand, the high demand for PV and wind-power generation when meeting the target of C neutrality (10–15 PWh y^{-1}), which could provide general insights into climate mitigation**.

To focus on this key storyline, we have carefully re-framed our paper by revising the Title, Abstract, Main Figures and Main Text (please see our response to **Comment B5** above).

The key storyline of our paper is now highlighted in the **Abstract**: *“Pledges to achieve carbon (C) neutrality in China represent the need to scale up photovoltaic (PV) and wind-power generation from 1 to 10–15 PWh y^{-1} by 2060 (refs.¹⁻⁵). Following the historical rate of renewable installation¹, energy modeling⁶ or forecasts that inform the China's 14th Five-year Energy Development (CFED) plans⁷, however, only indicated an increase in PV and wind-power generation to 5–9.5 PWh y^{-1} by 2060. Here we propose to increase the potential of PV and wind power by optimizing the location, capacity and construction time of 3,844 utility-scale PV and wind-power plants, which are coordinated with ultra-high-voltage transmission and energy storage by accounting for the flexibility of power loads and intertemporal dynamics of learning. Our optimisation model ramps up annual investments in PV and wind power from 72 billion US dollars (USD) in 2021 to 250 billion USD during 2021–2060 to accelerate the transition from fossil fuels to PV and wind power. In contrast to the CFED path⁷, our optimal path increases the generation of PV and wind power from 9 to 15 PWh y^{-1} in 2060, with a reduction of the average marginal abatement cost from 98 (ref.¹) to 6 (-1 to 12 as 90% uncertainty based on our Monte Carlo simulations) USD (t CO₂)⁻¹ by optimizing the power systems to minimize costs. The development of PV and wind power in China will reduce income inequality by creating new income opportunities for residents in the poorest regions. Our results reinforce the importance of trend-breaking investments for upgrading power systems to accelerate the development of PV and wind power at affordable costs to achieve C neutrality in China.”*

The Title of our paper is revised as: *“Achieving carbon neutrality in China by 2060 requires trend-breaking investments in photovoltaic and wind power”*.

Comment D5

Questions on the model: the study calculated the MAC of each plant and optimize the time of installation based on the MAC, the paper also optimize the revenue (might double counting by adding and replacing fossil fuel and selling renewable electricity), are these two optimizations combined in the optimization decision or one after another?

Response

There is only one optimisation in our model that is to optimize variables including the order of building power plants (ϵ), the capacity of each power plant based on the number of pixels installing PV panels or wind turbines (n_x), the time to build each power plant (t_x) and the option of energy storage (s_x). This point is explained in the Section of **Optimisation of the placement, capacity and timing of power plants** in **Methods**. Our optimisation is

constrained by the target of annual abatement of CO₂ emissions, which is explained in line 600-602: “*Solving the cost-minimization problem in Eq. 1 was constrained by the target of the annual abatement of CO₂ emissions by substituting fossil fuels when a new PV or wind-power plant ϵ was built (F_ϵ).*”. Our optimisation model returns the MAC for each PV and wind-power plant built during 2021–2060 and the LCOE of all new PV and wind-power plants.

When calculating the MAC of a new PV or wind-power plant based on the LCOE, we account for the income of replacing fossil fuels, which is explained in line 609-615: “*We derived the marginal abatement cost (MAC_ϵ) for a new PV or wind-power plant ϵ based on the abated CO₂ emissions: $MAC_\epsilon = \frac{LCOE_\epsilon \cdot E_\epsilon - LCOE_{\epsilon-1} \cdot E_{\epsilon-1} \cdot \varrho \cdot (E_\epsilon - E_{\epsilon-1})}{F_\epsilon - F_{\epsilon-1}}$ (10), where ϱ is the price of coal, oil or gas. We obtained the prices of coal (0.043 ± 0.015 USD kWh⁻¹ as the 95% confidence interval)⁸⁴, oil (0.141 ± 0.057 USD kWh⁻¹)^{85,86}, and gas (0.058 ± 0.016 USD kWh⁻¹)⁸⁷ in China as the averages during 2010–2020, when they are considered to generate electricity with an efficiency of 35, 38 and 45%, respectively⁸⁸.*”.

We estimate the revenue from developing PV and wind-power plants by accounting for the income of replacing fossil fuels based on the LCOE for all new plants and the saved carbon costs based on a prescribed carbon price. **We discard the PV and wind-power plants with MAC above the prescribed carbon price, which is not done in a new step of optimisation.** This is now explained in line 678-684: “*Given a carbon price (ζ), we considered power plants with MACs below this carbon price and estimated the revenue (R_ϵ) from power generation when building a new PV or wind-power plant (ϵ): $R_\epsilon = \varrho \cdot E_\epsilon + \zeta \cdot F_\epsilon - LCOE_\epsilon \cdot E_\epsilon$ (11), where ϱ is the price of coal, oil or gas in China that is substituted by PV or wind power, F_ϵ is total abatement of CO₂ emissions, E_ϵ is total PV and wind-power generation, and $LCOE_\epsilon$ is the LCOE for the projected PV and wind-power plants after building plant ϵ .*”.

Because our calculation of revenue is based on the LCOE of all new PV and wind-power plants rather than the MACs of them, the income of replacing fossil fuel is not double-counted in our estimate.

Comment D6

I understand the paper focuses on utility-scale solar, onshore, and offshore wind; however, it is not clear how the paper assumes the role of nuclear, CCS, pumped hydro, hydrogen, and other technologies.

Response

Our paper focus on addressing the potential of utility-scale PV and wind power in China. To show the contribution of PV and wind power to achieving C neutrality, we have considered the contribution of other technologies such as oil, gas, bioenergy, nuclear, hydropower and CCS. The feasibility of large-scale CCS is widely debated, so we focus on estimating the demand for CCS installed with fossil fuels (Duan et al., 2021) or biomass (Xu et al., 2022) to achieve China’s C neutrality **under different scenarios of PV and wind-power generation** by considering 1) the terrestrial C sink, 2) the projected rate of electrification (58%), 3) the total power demand, and 4) **the projected power generation by renewables other than PV and wind under a “1.5°C-limiting” scenario from a previous multi-model study (Duan et al., 2021) (see our new Fig. 3a).**

This method is explained in line 302-313: “*Most scenarios for climate mitigation meeting the target of 2°C warming considered by the Intergovernmental Panel on Climate Change (IPCC)^{8,51} rely on retrofitting existing plants with large-scale CCS to offset CO₂ emissions from fossil fuels, but their credibility has been debated for many reasons such as economic*

costs¹, geological constraints⁶⁰, and biomass limitations²³. We analyzed the trade-offs between developing PV and wind power and installing CCS with fossil fuels² or bioenergy²³ under C neutrality in 2060 by considering the terrestrial C sinks⁶¹, electrification of non-power sectors (58%)⁴⁴, and the projected generation of power by renewables other than PV and wind² (Fig. 3). A higher demand for CCS probably indicates a larger challenge of achieving C neutrality due to the higher costs of deploying CCS than many other decarbonizing technologies¹ and the unclear limits of C storage in China²⁷. We found that a shift of PV and wind power projected from the CFED forecast⁷ to our optimal path reduces the demand for CCS in the generation of power from 8.9 to 2.8 PWh y⁻¹ in 2060 (Fig. 3a).”.

In addition, we have addressed the advantage of PV and wind power relative to other renewables by comparing the MAC between PV and wind power in our estimate and other renewables based on a previous energy modeling (Goldman Sachs Research, 2021) in Fig. 2a, which is explained in line 207-210 “By achieving a lower MAC than other renewables, PV and wind is competitive for the annual power generation of 2.7 PWh y⁻¹ against hydropower, of 12.1 PWh y⁻¹ against nuclear energy, of 14.8 PWh y⁻¹ against C capture utilization and storage, and of 15.3 PWh y⁻¹ against hydrogen or direct air CCS (ref.¹).”.

References:

Duan, H. *et al.* Assessing China’s efforts to pursue the 1.5° C warming limit. *Science* **372**, 378–385 (2021).

Goldman Sachs Research. Carbonomics: China net zero—The clean tech revolution. Available at: <https://wwwqa.goldmansachs.com/insights/pages/gs-research/carbonomics-china-netzero/report.pdf> (2021).

Comment D7

It is not clear if the paper model the capacity expansion every 1/5/10 years or simply optimizes one-year in 2060? It seems the model assumes no new UHV transmission lines will be built. Not sure how the model deals with transmission cost if this is the case as those already built are sunk costs. It will help if the paper can show the hourly load, and how the consideration of storage, transmission, etc., changes the load.

Response

Our model optimizes the growth of PV and wind power capacity every 10 years during 2021–2060 by accounting for the intertemporal dynamics of learning. We make this choice since it takes 10–20 years for new technologies to be widely applied (Popp, 2002). This assumption is now explained in line 505-506: “We optimized the increase in power capacity every 10 years during 2021–2060, because it takes 10 to 20 years for new technologies to be widely applied⁷⁴.”.

For the UHV transmission, we consider 130 UHV lines built by 2025 and 817 UHV lines commissioned to be built during 2025–2060, which is now explained in line 213-217 in SI: “We considered that electricity will be transported among regions in a national grid network of ultra-high-voltage (UHV) transmission using 130 lines built by 2025 and 817 lines commissioned during 2025–2060 based on national energy development plans (Center for Security and Emerging Technology, 2021) (see the detailed information on these UHV lines in the **Supplementary Spreadsheet S1**).”.

We account for the costs of expanding current UHV transmissions lines after 2025 in our central case, while we show the impact of ignoring these costs in Fig. 2c.

Following the Reviewer’s suggestion, we show the impact of limiting the capacity of power plants, building new UHV lines, developing new energy storage systems, electrification of non-power sectors, considering flexibility of power load, and optimizing the intertemporal dynamics of learning on the hourly power load in a new Fig. 1c:

Fig. 1... (c) Seasonal and diurnal variations in the generation and demand of power under an electrification rate of 58% for non-power sectors in 2060. The shading represents the PV and wind-power generation without considering curtailments. Relative to a baseline scenario limiting the capacity of power plants to 10 GW, we design five experiments by sequentially increasing the limit of power capacity from 10 to 100 GW (Case A), building new UHV lines after 2025 (Case B), storing energy (Case C), improving the electrification of non-power sectors (Case D), considering the flexibility of hourly power loads (Case E), and optimizing the dynamics of learning (optimal path).

Reference:

Popp, D. Induced innovation and energy prices. *Ame. Econ. Rev.* **92**, 160–180 (2002).

Comment D8

The paper has done a great job of presenting and visualizing the **sensitivity** of key parameters to the results. However, the system is hardly impacted by one single parameter, investment costs, discounting rate, variable load, transmission, and carbon costs could change simultaneously, not sure how the paper address such cases.

Response

While it is a challenge to deliver the key message and present the sensitivities to all parameters at the same time, we follow three strategies to address them in our revised paper.

First, we consider the impact of six parameters that are model-relevant, including the rates of learning, the rate of discounting, capital costs, costs of building new UHV lines and the lifetime of power plants (Fig. 2c) and the carbon costs (Fig. 4a,d). We show the impact of varying the discounting rate from 3 to 7% y^{-1} , increasing the lifetime of power plants from 15 to 35 years and using different capital costs on the average MAC (Fig. S7). We adopt the recommended discounting rate of 5% y^{-1} for China from a multi-model study (Duan et al., 2021) in our central case, while we show the impact of using a high discounting rate (7% y^{-1}) from a previous study on MAC (Chen et al., 2021) (Fig. 2c). Similarly, we adopt the recommended lifetime of 25 years (Victoria et al., 2021) for PV and wind-power plants in our central case, while we show the impact of using a short lifetime of 20 years on MAC (Victoria et al., 2021) (Fig. 2c). We adopt the central estimate of capital costs by components, which is explained in Fig. S7: “In our central case (green line), we adopt the capital costs by

component using the data published by the China Photovoltaic Industry Alliance (2021) for PV-power plants (0.64 USD W^{-1} as the total capital costs) and using the data by Liu et al. (2015b) for onshore wind-power plants (0.68 USD W^{-1} as the total capital costs), because neither of the two estimates above (Lorenczik et al., 2020; China Electricity Council, 2021) provides the capital costs by component.”, while we show the impact of using another two estimates on MAC (Fig. 2c). We design two sensitivity tests by adopting the international learning rates or ignoring the costs of building new UHV lines after 2025 to show their impacts on the average MAC (Fig. 2c). Lastly, the carbon costs affect the revenue from PV and wind power by replacing fossil fuels, so we show the impact of applying different carbon prices on the income distribution and the Gini coefficient (Fig. 4a,d).

Second, we consider the impact of factors that have been optimized in our optimisation model, including the limit to the capacity of power plants, the hourly power load profile, transmission, energy storage, electrification, and intertemporal dynamics of learning. To show their impacts on the potential and costs of PV and wind-power generation, we design five experiments by sequentially increasing the limit of power capacity from 10 to 100 GW (Case A), constructing new UHV lines after 2025 (Case B), developing energy storages (Case C), improving electrification of non-power sectors (Case D), considering the flexibility of power load (Case E), and optimizing the time of building each power plant (optimal path). We show the impact of these assumptions on the power loads and the potential of PV and wind-power generation (Fig. 1c,d,f), on the potential and costs of climate mitigation by PV and wind power (Fig. 2), and on the Gini coefficient (Fig. 4d).

Third, we combine the uncertainties in our model parameters in Monte Carlo simulations when estimating the uncertainty in MAC (Fig. 2c) and the Gini coefficient under different carbon costs (Fig. 4d) to provide implications for climate mitigation and poverty alleviation. This part is now fully explained in line 688-699 in a section **Uncertainty analyses** in **Methods**: “We estimated the uncertainties in MAC and the Gini coefficient by running an ensemble of Monte Carlo simulations 40,000 times⁹⁶. We randomly varied the parameters in these simulations, including: **i**) the variability of PV power generation ($\pm 5\%$) over a suitable pixel (W_{ijy}) due to the impact of aerosol deposition on PV panels⁹⁷ and the variability of wind-power generation ($\pm 2\%$) over a suitable pixel (W_{ijy}) due to the impact of climate change on wind resources⁶⁶, **ii**) the rate of growth of power demand by province during 2020–2060 ($\pm 1\%$)⁷, **iii**) the parameters used in the calculation of initial investment costs based on the range of capital costs ($\pm 10\%$) from previous estimates^{79, 80}, **iv**) the historical rates of learning for different cost components measured in China (Table S1), and **v**) the parameters used for calculating the costs of UHV transmission and energy storage from different studies (Table S9). Lastly, we adopted the medians of MAC and the Gini coefficient to represent our best estimates, while we used the 90% uncertainties and interquartile ranges to represent their uncertainties.”.

The **Fig. S7** shows the impacts of varying the discounting rate, the lifetime of power plants and capital costs on the average MAC of all new PV and wind-power plants:

Fig. S7. Impacts of the discounting rate and lifetime of power plants on the average marginal abatement cost (MAC) for building all new PV and wind power-plants during 2021–2060 under different capital costs. We estimate the average MAC when increasing the discounting rate from 3 to 7% y⁻¹ (a) or increasing the lifetime of power plants from 15 to 35 years (b) under different capital costs (Lorenz et al., 2020; China Electricity Council, 2021; China Photovoltaic Industry Alliance, 2021; Liu et al., 2015b). The capital costs (0.73 and 0.88 USD W⁻¹ for PV and onshore wind, respectively) published by Lorenz et al. (2020) (orange line) are higher than the data (0.23 and 0.76 USD W⁻¹ for PV and onshore wind, respectively) published by the China Electricity Council (2021) (gray line). In our central case (green line), we adopt the capital costs by component using the data published by the China Photovoltaic Industry Alliance (2021) for PV-power plants (0.64 USD W⁻¹ as the total capital costs) and using the data by Liu et al. (2015b) for onshore wind-power plants (0.68 USD W⁻¹ as the total capital costs), because neither of the two estimates above (Lorenz et al., 2020; China Electricity Council, 2021) provides the capital costs by component.

The Fig. 2c shows the impacts of policy-dependent parameters as well as the parameters optimized in our model on the MACs of PV and wind power under different scenarios:

Fig. 2 |... We design five experiments by sequentially increasing the limit of power capacity from 10 to 100 GW (Case A), building new UHV lines after 2025 (Case B), developing energy storage (Case C), improving the electrification of non-power sectors (Case D), considering the flexibility of power loads (Case E), and optimizing the dynamics of learning (optimal path) relative to a baseline scenario limiting the capacity of power plant to 10 GW. ... (c) Violin plots of the average MAC when building new PV and wind-power plants to meet the power demand in 2060. We perform sensitivity experiments by applying the international learning rates (Table S1) (I), adopting low⁴⁹ (II) or high³⁷ (III) capital costs, ignoring construction costs of new UHV lines (IV), adopting a high discounting rate (7% y⁻¹)⁶ (V), and assuming a short lifetime (20 years) of power plants³⁸ (VI) relative to the optimal path.

The Fig. 4d shows the uncertainty in the calculated Gini coefficient under different carbon prices:

Fig. 4 | Impacts of developing PV and wind power on alleviating poverty in China. (a) Revenue from PV and wind-power generation in 2060 under different carbon prices. (b) Change in the distribution of per capita income when the carbon price increases from 0 to 100 USD (t CO₂)⁻¹. (c) Change in the income of the 10% poorest people in 2060 due to the development of PV and wind power during 2021–2060 when adopting a carbon price of 100 USD (t CO₂)⁻¹. (d) Change in the Gini coefficient when the carbon price increases from 0 to 100 USD (t CO₂)⁻¹. The shaded area represents 90% uncertainty in Monte Carlo simulations. (e) Flow of finances embodied in the transmission of electricity generated by new PV and wind-power plants built during 2021–2060 under a carbon price of 100 USD (t CO₂)⁻¹. The inserts show the changes in per capita income by region when the carbon price increases from 0 to 100 USD (t CO₂)⁻¹ in 2060.

References:

Duan, H. *et al.* Assessing China’s efforts to pursue the 1.5° C warming limit. *Science* **372**, 378–385 (2021).

Chen, X. *et al.* Pathway toward carbon-neutral electrical systems in China by mid-century with negative CO₂ abatement costs informed by high-resolution modeling. *Joule* **5**, 2715–2741 (2021).

Victoria, M. *et al.* Solar photovoltaics is ready to power a sustainable future. *Joule* **6**, 1041–1056 (2021).

Comment D9

Other minor notes: In SI: Ln 154: Needs better justification why the GE2.5MW is used, is it the most popular wind turbine used in China?

Response

According to the China Wind Energy Association (2020), 2–2.5 MW is China’s mainstream model for installed onshore-wind turbines in 2018 with an average capacity of 2.18 MW (Fig.

R1), while 5–10 MW is the mainstream model for offshore-wind turbines in China. This is now explained in line 484-489: “*Onshore wind turbines with the capacity of 2–2.5 MW and offshore wind turbines with the capacity of 5–10 MW are considered as the main models currently used in China⁷², so we considered models for onshore (General Electric 2.5 MW) and offshore (Vestas 8.0 MW) wind-power plants (Table S5) at a hub height of 100 m above the ground to convert air kinetic energy to electricity based on the recommended power generation curve⁷³ (Fig. S10).*”.

Fig. R1. Composition of on-shore wind turbines by capacity in China during 2008–2018. This graph is taken from a report published by China Wind Energy Association (2020).

Reference:

China Wind Energy Association. Report for the grid parity of China’s wind power (in Chinese). <https://finance.sina.com.cn/stock/stockzmt/2020-05-14/doc-iirczymk1651936.shtml> (2020).

Comment D10

Ln221: Do you assume no new UHV lines will be built after 2025? Can the model simulate transmission capacity expansion?

Response

We have considered 130 UHV lines built before 2025 and 817 UHV lines commissioned to be built during 2025–2060 (Center for Security and Emerging Technology, 2021). This assumption is now clarified in line 213-217 in SI: “*We considered that electricity will be transported among regions in a national grid network of ultra-high-voltage (UHV) transmission using 130 lines built by 2025 and 817 lines commissioned during 2025–2060 based on national energy development plans (Center for Security and Emerging Technology, 2021) (see the detailed information on these UHV lines in the Supplementary Spreadsheet SI).*”.

Our model simulates the expansion of UHV transmission capacity endogenously. This is now highlighted in line 323-328 in the Main Text: “*For example, the growth of capacity of PV and wind power does not depend on investments in electricity transmission in the CFED path⁷ (Fig. 3c). In contrast, our model considers that the growth of capacity of PV and wind power during 2021–2060 depends on investments in expanding the capacity of existing UHV transmission lines (Fig. 3b). Recognizing such dependency will help us to represent the techno-economic barriers in developing PV and wind power in China.*”.

Following the Reviewer’s suggestion, we added a new Fig. 3b,c to show the impact of expanding the capacity of UHV transmission on the total costs of PV, wind and CCS:

Fig. 3 | Different paths to achieve C neutrality in China by 2060. (a) ... (b) Dependence of the annual costs of PV, wind and CCS under C neutrality in 2060 on the costs of transmitting electricity by UHV lines during 2021–2060 in the optimal path from this study. (c) Dependence of the annual costs of PV, wind and CCS under C neutrality in 2060 on the costs of PV and wind power during 2021–2060 in the optimal (dashed lines) or CFED⁷ (solid lines) path. In (b), the total capacity of PV and wind power built during 2021–2060 in the optimal path from this study depends on the capacity of electricity transmission, while the total length of new UHV lines is indicated by the color of the circles. Differently, the total capacity of PV and wind power in 2060 depends on the rate of growth of PV and wind-power generation during 2021–2060 in the CFED⁷ path, which is indicated by the color of the circles in (c). We predict the costs of CCS based on the marginal abatement cost of CCS¹ and the demand for CCS to achieve C neutrality in 2060 under different scenarios.

Reference:

Center for Security and Emerging Technology. Translation: Outline of the People’s Republic of China 14th Five-Year Plan for National Economic and Social Development and Long-Range Objectives for 2035. Available at: <https://cset.georgetown.edu/publication/china-14th-five-year-plan/> (2021).

Comment D11

Ln273: Are costs endogenous with an installed capacity based on learning rates? Pay attention to the learning rates cited are mostly international studies, and rates in China are usually higher than international rates.

Response

Our model considers the costs of building a new PV or wind-power plant endogenously as a function of the capacity of all PV or wind-power plants that are built before this plant based on the historical rates of learning in China (detailed in **S8 Intertemporal dynamics of learning** in **SI Methods**). This point is now explained in line 141-145 in the main text: “*Similar to a previous study⁴⁵, we estimated that the rates of learning measured in China during 2000–2020 (standard deviations of 32.4±8.6% for PV modules and 11.8±2.7% for wind turbines) are higher than the international rates of learning from other regions during 1975–2020 (20.8±9.4% for PV and 8.5±2.8% for wind turbines) (Table S1), indicating that the unit costs of power generation will continue to decrease as the installed capacity increases.*”.

Our revised model adopts the historical rates of learning measured for China in our central case, but we show the impact of using the international rates of learning on MACs (Fig. 2c).

This is now explained in line 238-242: “*Adopting the international rates of learning for PV and wind (Table S1), high capital costs³⁷, a short lifetime of power plants (20 years)³⁸ or a high discounting rate (7% y⁻¹)⁶ increases MAC relative to our central estimate, while overlooking the construction costs of new UHV lines or using low capital costs⁴⁹ leads to lower MACs.*”.

Comment D12

Ln578: It seems offshore only gets built after 2040? Does it because of the assumption of costs? Incorporating offshore wind plans might help to address this issue.

Response

Our previous model treated offshore and onshore-wind power plants together when optimizing the time of construction based on learning, which leads to construction of offshore-wind power after 2040 due to the higher initial investment costs. We realize that these two technologies are distinct in their applications (Enevoldsen, 2016), so our revised model optimizes the construction of these two types of power plants separately, which results in construction of offshore-wind power plants since 2020 (see a new Fig. 1e).

However, we do not incorporate the national energy development plans for offshore-wind power in our model, because these plans could underestimate the potential of renewable energy in China, which has been highlighted as a finding in our paper (please see our response to **Comment B5**).

Fig. 1 | ... (e) Influences of increasing the capacity of PV and wind-power generation in the 2020s on the levelized cost of electricity (LCOE) of all new PV and wind-power plants built during 2021–2060. The optimal path minimizes LCOE by varying the construction time of each power plant by decade (the shaded area) under a discounting rate of 5% y⁻¹. The insert shows the annual costs by decade. We consider a “CFED” path by adopting the rate of installing renewables in China’s 14th Five-year Energy Development (CFED)⁷ and the projected costs of PV and wind power¹.

Reference:

Enevoldsen, P. & Valentine, S. V. Do onshore and offshore wind farm development patterns differ? *Energ. Sustain. Dev.* **35**, 41–51 (2016).

Reviewer Reports on the Second Revision:

Referees' comments:

Referee #2 (Remarks to the Author):

Summary of the key results

The manuscript models scenarios of expansion of large-scale wind and solar power in China to support the goal of carbon neutrality by 2060. It presents an advanced model with high degree of spatial and temporal granularity and a large number of factors taken into account. It shows that with massive and dedicated investment and careful optimisation of wind and solar power location marginal abatement costs for carbon emissions can be significantly reduced compared to the current government plans and prior studies.

Originality and significance:

I reviewed previous submissions of this study and I commented on the importance of the topic and the excellent execution of the study. In this version the study has even better highlighted its difference with previous studies so I have no doubt about its originality.

In terms of manuscript significance for a wide Nature readership, I still have concerns that it speaks to a relatively well-defined community of energy modellers and planners focus on China's electricity decarbonisation. This topic is of pivotal importance for the world as a whole (given China's share in global emissions and leadership in renewables) and yet I wonder whether the findings can be made more relevant for even broader audience. Perhaps even stating that the modelling approach may be applicable beyond China and what it would mean to bring down MACs globally would be useful. It would also be useful to explain in plain language accessible to a broad audience what we need to do to bring the costs down (see also on communication below).

Data & methodology: validity of approach, quality of data, quality of presentation, appropriate use statistics and treatment of uncertainties

Although I cannot check all results in the manuscript it seems to be very professionally executed with numerous sensitivity analyses to address potential uncertainties.

Conclusions: robustness, validity, reliability

The conclusions are robust and valid

Suggested improvements: experiments, data for possible revision

In line 314-320, the manuscript compares the year-on-year growth rates in the proposed scenarios versus in Europe and China historically. This is a useful comparison, but for mature technologies, comparing the normalised absolute growth rate (in % of the electricity supply) may be more

appropriate, especially as similar statistics has recently been collected from a number of countries.

References: appropriate credit to previous work?

The manuscript appropriately references previous work

Clarity and context: lucidity of abstract/summary, appropriateness of abstract, introduction and conclusions

The manuscript conveys extremely complex context and is very dense. For publication in Nature, the text would need to be more accessible (this also relates to stressing the study's significance).

Referee #4 (Remarks to the Author):

The authors have in general addressed my questions and comments. The paper is now more focused on the main message and the manuscript is also enhanced.

I only have a few small concerns about the explained assumptions, but I do not expect they will change the main findings of the paper or hurt the novelty of the work. The paper should note those limitations and clarify the interpretation of the results.

The paper assumes the costs/prices of fossil fuels do not change during the study period.

"We obtained the prices of coal (0.043 ± 0.015 USD kWh⁻¹ as the 95% confidence interval)⁸⁴, oil (0.141 ± 0.057 USD kWh⁻¹)^{85,86}, and gas (0.058 ± 0.016 USD kWh⁻¹)⁸⁷ in China as the averages during 2010–2020, when they are considered to generate electricity with an efficiency of 35, 38 and 45%, respectively⁸⁸"

The paper calculates the revenue as in (11) which assumes carbon price and fossil fuel price, while in a real setting, revenue is more likely seen as electricity price minus LCOE.

The model assumes 10 years interval which is acceptable, given renewables (solar and wind) have been evolving very fast 10 years might not be ideal or sufficient for a high-resolution study like this one.

Transmission lines, ideally, should be included in the optimization if possible as it is a major investment decision and transmission access has a huge impact on where and when to build renewables which is the major conclusion of the paper.

While checking the UHV lines in Supplementary Spreadsheet S1, it shows 128 lines will be built from Huidong to Wanan from 2025 to 2060, one such line has a 12GW capacity, that's 1536GW transmission capacity in total from this one linkage. Does this sound realistic? There are some other lineages that have similar assumptions.

Author Rebuttals to Second Revision:

Referee #2

Summary of the key results

The manuscript models scenarios of expansion of large-scale wind and solar power in China to support the goal of carbon neutrality by 2060. It presents an advanced model with high degree of spatial and temporal granularity and a large number of factors taken into account. It shows that with massive and dedicated investment and careful optimisation of wind and solar power location marginal abatement costs for carbon emissions can be significantly reduced compared to the current government plans and prior studies.

Originality and significance:

I reviewed previous submissions of this study and I commented on the importance of the topic and the excellent execution of the study. In this version the study has even better highlighted its difference with previous studies so I have no doubt about its originality.

In terms of manuscript significance for a wide Nature readership, I still have concerns that it speaks to a relatively well-defined community of energy modellers and planners focus on China's electricity decarbonisation. This topic is of pivotal importance for the world as a whole (given China's share in global emissions and leadership in renewables) and yet I wonder whether the findings can be made more relevant for even broader audience. Perhaps even stating that the modelling approach may be applicable beyond China and what it would mean to bring down MACs globally would be useful. It would also be useful to explain in plain language accessible to a broad audience what we need to do to bring the costs down (see also on communication below).

Following the Reviewer's suggestion, we revised the final paragraph to highlight the significance of our study in a plain language: "*Our approach improves the optimisation of PV and wind-power systems^{2,6,25-27} by applying a spatially explicit method, providing insights into the path of mitigation for countries beyond China¹⁴. First, we highlight that investments must increase for scaling up PV and wind power toward zero-C economies, even by considering the benefits of technological improvements^{29,30}. In contrast to the prediction that the costs of decarbonizing the energy systems by renewables will decline dramatically in the coming decade⁴⁵, our study highlights the impact of physical limitations (e.g. demand for land and infrastructure)³⁴, demonstrates the importance of policy interventions (e.g. supporting grid integration⁶ and optimizing power loads²⁸), and clarifies the need of ramping up initial investments to overcome techno-economic barriers (e.g. building large PV and wind-power plants³⁹). Second, our projection of PV and wind power provides opportunities to increase revenue for less developed regions with vast areas of desert and marginal lands, which has*

implications for a cost-effective use of PV and wind power in the semi-arid regions (e.g. the Middle East and Africa)³⁵. Third, optimization of the power systems using our method (e.g., identification of locations suitable for PV and wind power, adaptation of energy-consuming facilities to match the supply of renewables, and development of network for electricity transmission) for countries under development⁸ can potentially bring down global decarbonization costs and reduce the socio-political resistance. This study improves our understanding of techno-economic-territorial limitations in developing renewables, reinforces the importance of ramping up investments for accelerating energy transition toward PV and wind, and informs the optimal path to achieve C neutrality in the long term.”.

Data & methodology: validity of approach, quality of data, quality of presentation, appropriate use statistics and treatment of uncertainties.

Although I cannot check all results in the manuscript it seems to be very professionally executed with numerous sensitivity analyses to address potential uncertainties.

Conclusions: robustness, validity, reliability

The conclusions are robust and valid.

Suggested improvements: experiments, data for possible revision

In line 314-320, the manuscript compares the year-on-year growth rates in the proposed scenarios versus in Europe and China historically. This is a useful comparison, but for mature technologies, comparing the normalised absolute growth rate (in % of the electricity supply) may be more appropriate, especially as similar statistics has recently been collected from a number of countries.

Following the Reviewer’s suggestion, we revised the sentences in lines 350-355 to provide the growth rate of PV and wind power as a percentage of total power demand: “*Our optimal path indicates that the share of PV and wind in total power supply increases from 12 to 59% during 2021–2060 at an annual rate of 1.8, 1.4, 1.0 and 0.7% in the 2020s, 2030s, 2040s and 2050s, respectively, which requires accelerating growth this decade relative to an annual rate of 0.7% in France, 0.9% in Japan, 1% in China and the USA, or 1.7% in Germany in 2010s⁴¹.”.*

References: appropriate credit to previous work?

The manuscript appropriately references previous work.

We have added two new references to highlight the demand for renewable energy in China (*Xia et al., 2022*) and the importance of using a spatially explicit method (*Klaaßen et al., 2023*). We are happy to know if any other references should be cited in our paper.

References:

Klaaßen, L., & Steffen, B. Meta-analysis on necessary investment shifts to reach net zero

pathways in Europe. *Nat. Clim. Chang.* **13**, 1–9 (2023).

Xia, N. Four research teams powering China's net-zero energy goal. *Nature* **603**, S41 (2022).

Clarity and context: lucidity of abstract/summary, appropriateness of abstract, introduction and conclusions

The manuscript conveys extremely complex context and is very dense. For publication in *Nature*, the text would need to be more accessible (this also relates to stressing the study's significance).

We have improved the text of our paper to make it more accessible and stressed the study's significance (see the response to your first comment please).

Referee #4

The authors have in general addressed my questions and comments. The paper is now more focused on the main message and the manuscript is also enhanced.

I only have a few small concerns about the explained assumptions, but I do not expect they will change the main findings of the paper or hurt the novelty of the work. The paper should note those limitations and clarify the interpretation of the results.

The paper assumes the costs/prices of fossil fuels do not change during the study period. “We obtained the prices of coal (0.043 ± 0.015 USD kWh⁻¹ as the 95% confidence interval)⁸⁴, oil (0.141 ± 0.057 USD kWh⁻¹)^{85,86}, and gas (0.058 ± 0.016 USD kWh⁻¹)⁸⁷ in China as the averages during 2010–2020, when they are considered to generate electricity with an efficiency of 35, 38 and 45%, respectively⁸⁸”. The paper calculates the revenue as in (11) which assumes carbon price and fossil fuel price, while in a real setting, revenue is more likely seen as electricity price minus LCOE.

Our study aims at clarifying the impact of climate policy (carbon prices) on the revenue of developing PV and wind power. We agree with the Reviewer that revenue should be derived from electricity price minus LCOE, but we consider that the electricity price depends on carbon prices and the prices of fossil fuel. We do not have a reliable method to predict the fossil fuel prices in 2060, so we ran Monte Carlo simulations. In each Monte Carlo simulation, we randomly drew fossil fuel prices from the normal distributions, the average and standard deviation of which are estimated from the fossil fuel prices from 2010 to 2020 (please see our **Supplementary Fig. 6**).

To clarify our method, we added the following sentences in lines 1005-1015: “*Revenue of PV and wind-power generation could be derived from the electricity price minus LCOE, where the electricity price can be influenced by many socio-political factors^{28,50}. We focus on analyzing the impact of carbon price as a proxy for climate policies on the revenue of PV and wind power, so we consider that the electricity price depends on the fossil fuel prices and carbon price. The fossil fuel prices might increase in the future owing to the scarcity of fossil fuels⁷², which could increase the revenue of replacing fossil fuels with renewables including PV and wind power. To estimate the fossil fuel prices in 2060, we randomly draw the prices from normal distributions, of which the average and standard deviations are estimated from the prices of coal and oil from 2010 to 2020 and the prices of natural gas from 2013 to 2020 (Supplementary Fig. 6). Predicting the impact of the energy scarcity on the fossil fuel prices and thus the revenue of PV and wind power deserves further study.*”.

The model assumes 10 years interval which is acceptable, given renewables (solar and wind)

have been evolving very fast 10 years might not be ideal or sufficient for a high-resolution study like this one.

By performing a sensitivity experiment at an interval of 5 years, we confirmed that the installed PV and wind-power capacity and total costs are moderately changed (please see our **Supplementary Fig. 12**).

To clarify the impact of the interval of optimization, we added the following sentences in lines 815-820: “*Given the variation of renewable energy within a decade, we performed a sensitivity experiment by optimizing the model at an interval of 5 years, where the installed PV and wind-power capacity and total costs both change moderately (Supplementary Fig. 12). Nevertheless, simulating the penetration of renewable energy within a decade will be useful to improve the optimization model.*”.

We also added a new **Fig. S12** in the Supporting Information:

Fig. S12. Impacts of the interval of optimization on the installed capacity and costs of new PV and wind-power plants built during 2020–2060. The total installed capacity (a) and annual costs (b) of all new PV and wind-power plants built during 2020–2060 are estimated by our optimization model run at a temporal interval of 5 years (bar) or 10 years (pentagrams).

Transmission lines, ideally, should be included in the optimization if possible as it is a major investment decision and transmission access has a huge impact on where and when to build renewables which is the major conclusion of the paper.

We have considered the impact of transmission access on where and when to build new PV and wind power plants in our model. When optimizing the construction of 3,844 PV and wind-power plants, we have considered the costs of building new UHV transmission lines when a new power plant is built, which has influences on the LCOE of this plant and the construction time of individual power plants. After optimizing the construction time of each power plant, we considered that the new UHV line needed for this plant will be constructed at the same time. To clarify the impact of transmission access on where and when to build new PV and wind power

plants considered in our model, we added the following sentences in lines 831-837: “*In our optimisation model of the construction of 3,844 PV and wind-power plants, we have considered the costs of building new UHV transmission lines when a new power plant is built, which influences the LCOE of this plant and the construction time of all new plants, so we have considered the impact of transmission access on where and when to build new PV and wind-power plants. After optimizing the construction time of each new power plant, we considered that the new UHV line needed for this plant will be constructed at the same time.*”.

We revised the **Fig. S1** in the Supporting Information to clarify that the UHV costs had influenced the total costs of building each power plant and the LCOE, which determined the construction time of individual power plants:

Fig. S1. Procedures of optimizing the placement, capacity and timing of new PV and wind-

power plants in China. We coordinate PV and wind power generation with electricity transmission and storage in the power systems. We use the levelized cost of electricity (LCOE) to indicate the grid parity of PV and wind power generation. We seek the best strategy of installing PV panels or wind turbines in each county for achieving the largest power generation. We take the number of pixels installing PV panels or wind turbines and the time of building each PV or wind-power plant by decade as a decision variable in our optimisation model for the minimization of LCOE of all PV and wind-power plants.

While checking the UHV lines in Supplementary Spreadsheet S1, it shows 128 lines will be built from Huaidong to Wanan from 2025 to 2060, one such line has a 12GW capacity, that's 1536 GW transmission capacity in total from this one linkage. Does this sound realistic? There are some other lineages that have similar assumptions.

Sorry for this confusion in our original manuscript. When our model projected 128 lines built from Huaidong to Wanan, we were predicting that at least a total transmission capacity of 1536 GW is required for transmitting electricity from the region centered in Huaidong to the region centered in Wanan. We did not have explicit information for all UHV transmission lines, so we assumed that the UHV lines projected by the central government could be used as a proxy for other lines between major regions in the country. Although this assumption might lead to some bias in the cost estimates due to the lack of detailed information for the future UHV lines, it helped us to determine the demand for electricity transmission between regions.

To clarify our assumption on new UHV lines, we added the following sentences in lines 838-851: *“As a caveat of this study, we do not have explicit information for all transmission lines, so we assumed that the UHV lines projected by the central government can be used as a proxy for UHV lines between major regions in the country. This assumption is useful to estimate the demand for electricity transmission between regions, but it could lead to bias in the cost estimation due to the lack of detailed information for all UHV lines. For example, the projection of 128 UHV lines with a capacity of 12 GW each from Huaidong to Wanan in our model indicates that at least a total transmission capacity of 1536 GW is required for transmitting electricity from the region centered in Huaidong to the region centered in Wanan, but the ultimate UHV lines built between these two regions might be different from our prediction. This limitation should be addressed when detailed information for all UHV lines are available. When considering the transmission of electricity from a county, we search for the substation of UHV lines which is closest to this county, and then we can estimate the cost of electricity transmission from this county to the transmission substation and the cost of electricity transmission using one of the UHV lines.”.*

Reviewer Reports on the Third Revision:

Referees' comments:

Referee #2 (Remarks to the Author):

I have reviewed the previous versions of the manuscript before and assessed its novelty and significance, which were both very high. In response to my most recent comments the authors explained wider significance of their results for a broader scientific audience. In addition they have also improved the accessibility and quality of the text and graphic material which makes the manuscript even better.

I have a couple of minor comments.

First, the authors now compare the annual rates of additions of wind and solar power to the decadal rates in France, Japan, China and the US (lines 213-217), indicating that the proposed addition rates are challenging but feasible. A similar comparison can be made to a more systematic assessment of maximum growth rates (for solar and wind separately) in all countries made by ref. 13 thus providing one more evidence of potential feasibility of the proposed scenarios (but also its ambition).

Second, the text in lines 224-227 is unclear. Which costs are compared to which under which scenarios? This may require a simple re-writing.

Referee #4 (Remarks to the Author):

The authors have addressed my comments, and I do not have any further questions.

On the caveats of UHV transmissions, the authors could discuss the physical, technical, and social economic constraints of building such a large transmission capacity, even if it might be an optimized results.

P.S. Wanan should be Wan'nan. It makes sense for the authors to check the text, data, and model thoroughly.

Author Rebuttals to Third Revision:

Referee #2

I have reviewed the previous versions of the manuscript before and assessed its novelty and significance, which were both very high. In response to my most recent comments the authors explained wider significance of their results for a broader scientific audience. In addition they have also improved the accessibility and quality of the text and graphic material which makes the manuscript even better.

I have a couple of minor comments.

First, the authors now compare the annual rates of additions of wind and solar power to the decadal rates in France, Japan, China and the US (lines 213-217), indicating that the proposed addition rates are challenging but feasible. A similar comparison can be made to a more systematic assessment of maximum growth rates (for solar and wind separately) in all countries made by ref. 13 thus providing one more evidence of potential feasibility of the proposed scenarios (but also its ambition).

We re-wrote this sentence to compare the projected rates for PV and wind separately with the maximum growth rates in six countries where the growth of PV or wind is table or stalling from Fig. 3 of ref. 13 in lines 186–190: *“Although the projected annual growth rates for wind (1%) and PV (0.8%) in China during the 2020s are comparable to the maximal annual rates of 1% in Spain, 0.9% in Turkey, and 0.6% in the USA and New Zealand for wind, or 1.1% in Japan and 1% in Germany for PV¹³, the expansion of these technologies may present greater challenges in China because of her larger absolute power demand¹.”*

Second, the text in lines 224-227 is unclear. Which costs are compared to which under which scenarios? This may require a simple re-writing.

Thank you for the good suggestion. We re-wrote this sentence to clarify which costs (i.e. cost reduction for PV, wind, and CCS) are compared to which (i.e. the increase in costs for PV and wind power) under which scenarios (i.e. the optimal path toward C neutrality in 2060) in lines 195–199: *“An example of the findings show that with the increase in PV and wind investment from \$0 to \$60 billion y^{-1} over the period 2021–2060, the ratio of cost reduction for PV, wind, and CCS to the increase in costs for PV and wind power is 2.7:1 in the optimal path toward C neutrality in 2060, which is lower than the ratio of 6.1:1 in CFED plans⁷.”*

Referee #4

The authors have addressed my comments, and I do not have any further questions.

On the caveats of UHV transmissions, the authors could discuss the physical, technical, and social economic constraints of building such a large transmission capacity, even if it might be an optimized results.

Thank you for the good suggestion. We agree with the Reviewer that, though our results are obtained from our optimization model, and the physical, technical and social economic constraints of building such a large transmission capacity should be considered in the construction. To highlight the constraints of building such a large transmission capacity, we added two sentences in lines 594–599: *“Although this study projected the construction of a large transmission capacity to optimize power systems, it is important to account for the physical, technical, and economic constraints. These include the demand for advanced polymer matrix composites that can operate under a voltage of >1000 kV, the construction of UHV lines over challenging terrains, the maintenance of these lines, and ensuring the security of electricity transmission under extreme weather conditions.”*.

P.S. Wanan should be Wan'nan. It makes sense for the authors to check the text, data, and model thoroughly.

Thank you very much for your good suggestion. We have taken this opportunity to check through the text, data and model again to ensure reliability of our results. We follow three rules to give the name of all UHV stations. First, we adopted the direct translation to be understood easily. For example, we used “Beijingdong” rather than “East of Beijing”. Second, we adopted the symbol ' to separate the name if this is confusion about the name when vowels of “a, o, e” are following the consonants. Third, we distinguished “u” from “ü” when they are following the syllable of “n” and “l”. The Supplementary Spreadsheet file is updated accordingly.